



# A global anthropogenic emission inventory of atmospheric pollutants from sector- and fuel-specific sources (1970-2017): An application of the Community Emissions Data System (CEDS)

Erin E. McDuffie[1,2], Steven J. Smith[3], Patrick O'Rourke[3], Kushal Tibrewal[4], Chandra Venkataraman[4],

Eloise A. Marais[5], Bo Zheng[6], Monica Crippa[7], Michael Brauer[8,9], Randall V. Martin[2,1]

[1]Department of Physics and Atmospheric Science, Dalhousie University, Halifax, NS, Canada
[2]Department of Energy, Environmental, and Chemical Engineering, Washington University in St. Louis, St. Louis, MO, USA
[3]Joint Global Change Research Institute, Pacific Northwest National Laboratory, College Park, MD, USA
[4]Department of Chemical Engineering, Indian Institute of Technology Bombay, Mumbai, Maharashtra, India
[5]School of Physics and Astronomy, University of Leicester, Leicester, UK
[6]State Key Joint Laboratory of Environment Simulation and Pollution Control, School of Environment, Tsinghua University, Beijing 100084, People's Republic of China
[7]European Commission, Joint Research Centre (JRC), Via E. Fermi 2749 (T.P. 123), 21027 Ispra, Varese, Italy
[8]School of Population and Public Health, University of British Columbia, Vancouver, BC, Canada
[9]Institute for Health Metrics and Evaluation, University of Washington, Seattle, WA, USA

*Correspondence to*: Erin McDuffie (erin.mcduffie@dal.ca)





**Abstract.** Global anthropogenic emission inventories remain vital for understanding the fate and transport of atmospheric pollution, as well as the resulting impacts on the environment, human health, and society. Rapid changes in today's society require that these inventories provide contemporary estimates of multiple atmospheric pollutants with both source sector and fuel-type information to understand and effectively mitigate future impacts. To fill this need, we have updated the open-source

Community Emissions Data System (CEDS) (Hoesly et al., 2019) to develop a new global emission inventory, CEDS$_{GBD-MAPS}$. This inventory includes emissions of seven key atmospheric pollutants ($NO_x$, CO, $SO_2$, $NH_3$, NMVOCs, BC, OC) over the time period from 1970 – 2017 and reports annual country-total emissions as a function of 11 anthropogenic sectors (agriculture, energy generation, industrial processes, transportation (on-road and non-road), residential, commercial, and other sectors (RCO), waste, solvent use, and international-shipping) and four fuel categories (total coal, solid biofuel, and the sum of liquid

fuels and natural gas combustion, plus remaining process-level emissions). The CEDS$_{GBD-MAPS}$ inventory additionally includes global gridded (0.5°×0.5°) emission fluxes with monthly time resolution for each compound, sector, and fuel-type to facilitate their use in earth system models. CEDS$_{GBD-MAPS}$ utilizes updated activity data, updates to the core CEDS default calibration procedure, and modifications to the final procedures for emissions gridding and aggregation to retain sector and fuel-specific information. Relative to the previous CEDS data released for CMIP6 (Hoesly et al., 2018), these updates extend the emission

estimates from 2014 to 2017 and improve the overall agreement between CEDS and two widely used global bottom-up emission inventories. The CEDS$_{GBD-MAPS}$ inventory provides the most contemporary global emission estimates to-date for these key atmospheric pollutants and is the first to provide global estimates for these species as a function of multiple fuel-types across multiple source sectors. Dominant sources of global $NO_x$ and $SO_2$ emissions in 2017 include the combustion of oil, gas, and coal in the energy and industry sectors, as well as on-road transportation and international shipping for $NO_x$.

Dominant sources of global CO emissions in 2017 include on-road transportation and residential biofuel combustion. Dominant global sources of carbonaceous aerosol in 2017 include residential biofuel combustion, on-road transportation (BC only), as well as emissions from waste. Global emissions of $NO_x$, $SO_2$, CO, BC, and OC all peak in 2012 or earlier, with more recent emission reductions driven by large changes in emissions from China, North America, and Europe. In contrast, global emissions of $NH_3$ and NMVOCs continuously increase between 1970 and 2017, with agriculture serving as a major source of

global $NH_3$ emissions and solvent use, energy, residential, and the on-road transport sectors as major sources of global NMVOCs. Due to similar development methods and underlying datasets, the CEDS$_{GBD-MAPS}$ emissions are expected to have consistent sources of uncertainty as other bottom-up inventories, including uncertainties in the underlying activity data and sector- and region-specific emission factors. The CEDS$_{GBD-MAPS}$ source code is publicly available online through GitHub: https://github.com/emcduffie/CEDS/tree/CEDS_GBD-MAPS. The CEDS$_{GBD-MAPS}$ emission inventory dataset (both annual

country-total and global gridded files) is publicly available and registered under: https://doi.org/10.5281/zenodo.3754964 (McDuffie et al., 2020c).



# 1 Introduction

Human activities emit a complex mixture of chemical compounds into the atmosphere, impacting air quality, the environment, and population health. For instance, direct emissions of nitric oxide (NO) rapidly oxidize to form nitrogen dioxide ($NO_2$) and

can lead to net ozone ($O_3$) production in the presence of sunlight and oxidized volatile organic compounds (VOCs) (e.g., Chameides, 1978;Crutzen, 1970). In addition, direct emissions of organic and black carbon-containing particles (OC, BC), as well as secondary reactions involving gaseous sulfur dioxide ($SO_2$), NO, ammonia ($NH_3$), and VOCs can lead to atmospheric fine particulate matter less than 2.5 µm in diameter ($PM_{2.5}$) (e.g., Mozurkewich, 1993;Jimenez et al., 2009;Saxena and Seigneur, 1987;Brock et al., 2002). $PM_{2.5}$ concentrations were estimated to account for nearly 3 million deaths worldwide in

2017 (Stanaway et al., 2018), while surface $O_3$ concentrations were associated with nearly 500,000 deaths in 2017 (Stanaway et al., 2018) and significant global crop losses, valued at \$11 billion ($USD_{2000}$) in 2000 (Avnery et al., 2011;Ainsworth, 2017). In addition, atmospheric $O_3$ and aerosol both impact Earth's radiative budget (e.g., Bond et al., 2013;Haywood and Boucher, 2000;US EPA, 2018). Other pollutants, including carbon monoxide (CO), $NO_2$, and $SO_2$ are also directly hazardous to human health (US EPA, 2018), while $NO_2$ and $SO_2$ can additionally contribute to acid rain (Saxena and Seigneur, 1987;US EPA,

2018) and indirectly impact human health via their contributions to secondary $PM_{2.5}$ formation. In addition, $NH_3$ deposition and nitrification can also cause nutrient imbalances and eutrophication in terrestrial and marine ecosystems (e.g., Behera et al., 2013;Stevens et al., 2004). While these reactive gases and aerosol have both anthropogenic and natural sources, dominant global sources of $NO_x$ (= NO + $NO_2$), $SO_2$, CO, and VOCs include fuel transformation and use in the energy sector, industrial activities, and on-road and off-road transportation (Hoesly et al., 2018). Global $NH_3$ emissions are predominantly from

agricultural activities such as animal husbandry and fertilizer application (e.g., Behera et al., 2013) and OC and BC have large contributions from incomplete or uncontrolled combustion in residential and commercial settings (e.g., Bond et al., 2013). Emissions of these compounds and the distribution of their chemical products vary spatially and temporally, with atmospheric lifetimes that allow for their transport across political boundaries, continuously driving changes in the composition of the global atmosphere.

Global emission inventories of these major atmospheric pollutants, with both sectoral, and fuel-type information are paramount for 1) understanding the range of emission impacts on the environment and human health and 2) for developing effective strategies for pollution mitigation. For example, general circulation/climate (GCM) and chemical transport models (CTM) use spatially gridded emission inventories as inputs to solve for atmospheric transport and chemical processes, and to predict the evolution of atmospheric constituents over space and time. By perturbing emission sources or historical emission

trends, such models can quantify the impact of emissions on the environment, economy, and human health (e.g., Mauzerall et al., 2005;Lelieveld et al., 2019;IPCC, 2013;Liang et al., 2018;Lacey and Henze, 2015), provide mitigation-relevant information for polluted regions (e.g., GBD MAPS Working Group, 2016, 2018;RAQC, 2019;Lacey et al., 2017), and anchor future projections (e.g., Shindell and Smith, 2019;Venkataraman et al., 2018;Gidden et al., 2019;Mickley et al., 2004).



Three global emission inventories have been widely used for these purposes, including the Emissions Database for
Global Atmospheric Research (EDGAR) from the European Commission Joint Research Centre (Crippa et al., 2018), the
ECLIPSE v5a (Evaluating the Climate and Air Quality Impacts of Short-Lived Pollutants) inventory from the Greenhouse Gas
– Air Pollution Interactions and Synergies (GAINS) model at the International Institute for Applied Systems Analysis (IIASA)
(Amann et al., 2011;Klimont et al., 2017), and the CEDS (v2016-07-26) inventory from the newly developed Community
Emissions Data System (CEDS), from the Joint Global Change Research Institute at the Pacific Northwest National Laboratory
and University of Maryland (Hoesly et al., 2018). All three inventories are derived using a bottom-up approach where
emissions are estimated using reported activity data (e.g., amount of fuel consumed) and source- and regionally- (where
available) specific emission factors (mass of emitted pollutant per mass of fuel consumed) for each emitted compound. All
three inventories are similar in that they use this bottom-up approach to provide historical, source-specific gridded emission
estimates of major atmospheric pollutants ($NO_x$ (as $NO_2$) $SO_2$, CO, NMVOCs, $NH_3$, BC, and OC). Table 1 provides a
comparison of the key features between these inventories, which provide emissions from multiple source sectors over the
collective time period from 1750-2014. In contrast to EDGAR and GAINS, the CEDS system implements a mosaic approach
that uses activity and emission input data from other sources such as EDGAR, GAINS, and regional/national-level inventories
in order to produce global emissions that are both historically consistent and reflective of contemporary country-level estimates
(Hoesly et al., 2018). The CEDS source code has also been publicly released (https://github.com/JGCRI/CEDS/tree/master),
increasing both the reproducibility and public accessibility to quality emission estimates of global and national-level air
pollutants.

Due to the long development times of global bottom-up inventories, current versions of the EDGAR, ECLIPSE, and
CEDS inventories are limited in their ability to capture emission trends over recent years (Table 1), particularly the last 6 – 10
years in regions undergoing rapid change such as China, North America, Europe, India, and Africa. For example, China
implemented the Action Plan on the Prevention and Control of Air Pollution in 2013, which has targeted specific emission
sectors, fuels, and species and resulted in reductions of ambient $PM_{2.5}$ concentrations by up to 40% in metropolitan regions
between 2013 and 2017 (reviewed in Zheng et al., 2018). Similarly, over the past 10-20 years in the US and Europe, the
reduction of coal-fired power plants and phase-in of stricter vehicle emission standards have resulted in emission reductions
of $SO_2$ and $NO_x$ across these regions (Krotkov et al., 2016;Duncan et al., 2013;Castellanos and Boersma, 2012;de Gouw et al.,
2014). Over this same time period, however, oil and gas production in key regions in the US has more than tripled between
2007 and 2017 (EIA, 2020). In addition, the absence of widespread regulations targeting $NH_3$ from agricultural practices has
led to continuous increases in global $NH_3$ emissions (Behera et al., 2013). Global energy consumption also increased by an
average of 1.5% each year between 2008 and 2018 (BP, 2019) and the global consumption of coal increased for the first time
in 2017 since its peak in 2013 (BP, 2019). Many of these energy changes have been attributed to the growth of energy
generation in rapidly growing regions, such as India (BP, 2019). Africa is also experiencing rapid growth, with increasing
emissions from diffuse and inefficient combustion sources, which may not be accurately accounted for in current global
inventories (Marais and Wiedinmyer, 2016). Therefore, to capture recent trends around the globe, quantify the resulting



economic, health, and environmental impacts, and mitigate future burdens, computational models require emission inventories
with regionally accurate estimates, global coverage, and the most up-to-date information possible. Though global bottom-up
inventories can lag in time due to data collection and reporting requirements, the incorporation of smaller regional inventories
provides the opportunity to improve the timeliness and regional accuracy of global estimates.

To further increase the policy-relevance of such data, it is also important that global emission inventories not only
provide contemporary estimates, but report emissions as a function of detailed source sector and fuel type. For example, the
recent air quality policies in China have included emission reductions targeting coal-fired power plants within the larger energy
generation sector (e.g., Zheng et al., 2018). Decisions to implement such policies require accurate predictions of the air quality
benefits, which in turn depend on simulations that use accurate estimates of contemporary sector- and fuel-specific emissions.
While the EDGAR, ECLIPSE, and CEDS inventories all provide varying degrees of sectoral information (Table 1), there are
no global inventories to-date that provide public datasets of multiple atmospheric pollutants with both detailed source sector
and fuel-type information. Crippa et al. (2019) do describe estimates of biofuel use from the residential sector in Europe using
emissions from the EDGARv4.3.2 inventory (EC-JRC, 2018), but do not report global estimates or regional emissions from
other fuel-types. Similarly, Hoesly et al. (2018) describe fuel-specific activity data and emission factors used to develop the
global CEDSv2016-07-26 inventory, but do not publicly report final global emissions as a function of fuel-type. In contrast, a
limited number of regional inventories have provided both fuel- and sector-specific emissions. These inventories, for example,
have been applied to earth system models to attribute the mortality associated with outdoor air pollution to dominant sources
of ambient $PM_{2.5}$ mass, such as residential biofuel combustion in India and coal combustion in China (GBD MAPS Working
Group, 2018, 2016). As countries undergo rapid changes that impact fluxes of their emitted pollutants, including population,
emission capture technologies, and the mix of fuels used, fuel and source-specific estimates are vital for capturing these
contemporary changes and understanding the air quality impacts across multiple scales.

As part of the Global Burden of Disease - Major Air Pollution Sources (GBD-MAPS) project, which aims to quantify
the    disease    burden    associated    with    dominant    country-specific    sources    of    ambient    $PM_{2.5}$    mass
(https://sites.wustl.edu/acag/datasets/gbd-maps/), we have updated and utilized the CEDS open source emissions system to
produce a new global anthropogenic emission inventory ($CEDS_{GBD-MAPS}$). $CEDS_{GBD-MAPS}$ includes country-level and global
gridded (0.5°×0.5°) emissions of seven major atmospheric pollutants ($NO_x$ (as $NO_2$), CO, $NH_3$, $SO_2$, NMVOCs, BC, OC) as a
function of 11 detailed emission source sectors (agriculture, energy generation, industry, on-road transportation, non-road/off-
road transportation, residential energy combustion, commercial combustion, other combustion, solvent use, waste, and
international shipping) and four fuel groups (emissions from the combustion of total coal, solid biofuel, liquid fuels and natural
gas, plus all remaining process-level emissions) for the time period between 1970 – 2017. Similar to the prior CEDS inventory
released for CMIP6 (Hoesly et al., 2018), $CEDS_{GBD-MAPS}$ provides surface level emissions from all sectors, including fertilized
soils, but does not include emissions from open burning. In the first two sections we provide an overview of the $CEDS_{GBD-}$
$_{MAPS}$ system and describe the updates that have allowed for the extension to the year 2017 and the added fuel-type information.
These include updates to the underlying activity data and input emission inventories used for default estimates and calibration



procedures (including the use of two new inventories from Africa and India), the added calibration of default BC and OC emissions, as well as the use of updated spatial gridding proxies, and adjustments to the final gridding and aggregation steps that retain detailed sub-sector and fuel-type information. The third section presents global CEDS$_{GBD-MAPS}$ emissions in 2017

and discusses historical trends as a function of compound, sector, fuel-type, and world region. The final section provides an evaluation of the global CEDS$_{GBD-MAPS}$ emissions against other global inventories, as well as a discussion of the magnitude and sources of uncertainty associated with the CEDS$_{GBD-MAPS}$ products.

## 2 Methods

The December 23, 2019 full release of the Community Emissions Data System (Hoesly et al., 2019) provides the core system

framework for the development of the contemporary CEDS$_{GBD-MAPS}$ inventory. As detailed in Hoesly et al. (2018), a similar version of the CEDS system was used to produce the first CEDSv2016-07-26 inventory (hereafter called CEDS$_{Hoesly}$) (CEDS, 2017a, b), which provides global gridded (0.5°×0.5°) emissions of atmospheric reactive gases (NO$_x$ (as NO$_2$), SO$_2$, NH$_3$, NMVOCs, CO), carbonaceous aerosol (BC, OC), and greenhouse gases (CO$_2$, CH$_4$) from eight anthropogenic sectors (Agriculture (AGR), Transportation (TRA), Energy (ENE), Industry (IND), Residential, Commercial, Other (RCO), Solvents

(SLV), Waste (WST), International Shipping (SHP)) over the time period from 1750 - 2014. Here we provide a brief overview of the Community Emissions Data System with detailed descriptions of the major updates that have been implemented to produce the new CEDS$_{GBD-MAPS}$ inventory. This inventory has been extended to provide emissions from 1970 – 2017 for reactive gases and carbonaceous aerosol (NO$_x$, SO$_2$, NMVOCs, NH$_3$, CO, BC, OC) with increased fuel and sectoral information relative to the CEDS$_{Hoesly}$ inventory (Sect. 2.2.-2.3). Updates primarily include the use of updated input datasets

(Sect. 2.1), new and updated global and regional calibration inventories (Sect. 2.2), added calibration of default BC and OC emissions (Sect. 2.3), and the disaggregation of emissions into contributions from additional source sectors and multiple fuel-types (Sect. 2.4).

### 2.1. Overview of CEDS$_{GBD-MAPS}$ System

The CEDS system has five key procedural steps, illustrated in Fig. 1. After the collection of input data in Step 0, Step

1 calculates default global emission estimates (Em) for each chemical compound using a bottom-up approach shown in Eq. (1). In Eq. (1), emissions are calculated using relevant activity (A) and emission factor (EF) data for each country (c) and year (y), as a function of 52 detailed working sectors (s) and nine fuel-types (f) (Table 2). CEDS conducts these calculations for two types of emission categories: 1) fuel combustion sources (e.g., electricity production, industrial machinery, on-road transportation, etc.) and 2) process sources (e.g., metal production, chemical industry, manure management, etc.). We note that

the distinction between these source categories is reflective of both sector definition and CEDS methodology, as described further in Sect. S2.1. This results in some working sectors that include emissions from combustion, such as waste incineration and fugitive petroleum and gas emissions, to be characterized in the CEDS system as process-level sources (further details in





Sect. S2.1). In contrast to CEDS combustion source emissions, which are calculated in Eq. (1) as a function of 8 fuel types, emissions from CEDS process-level sources are combined into a single 'process' category, as described in Sect. 2.4. Table 2

provides a complete list of CEDS_GBD-MAPS working sectors and fuel-types, as well as source category distinctions.

$$\mathrm{Em}_{\mathrm{species}}^{\mathrm{country,\ sector,\ fuel,\ year}} = \mathrm{A}^{\mathrm{c,\ s,\ f,\ y}} \times \mathrm{EF}_{\mathrm{species}}^{\mathrm{c,\ s,\ f,\ y}} \qquad (1)$$

For emissions from CEDS combustion sources, annual activity drivers in Eq. (1) primarily include country-, fuel-, and sector-specific energy consumption data from the International Energy Agency (IEA, 2019). Sector- and compound-specific emission factors are typically derived from energy use and total emissions reported from other inventories, including

from the GAINS model (Klimont et al., 2017;IIASA, 2014;Amann et al., 2015), Speciated Pollutant Emission Wizard (SPEW) (Bond et al., 2007), and the US National Emissions Inventory (NEI) (NEI, 2013). In contrast, default emissions (Em) for CEDS process sources are directly taken from other inventories, including from the EDGAR v4.3.2 global emission inventory (EC-JRC, 2018;Crippa et al., 2018). "Implied emission factors" are then calculated for these process sources in Eq. (1) using global population data (UN, 2019, 2018) or pulp and paper consumption (FAOSTAT, 2015) as the primary activity drivers. For years

without available emissions, default estimates for CEDS process sources are calculated in Eq. (1) from a linear interpolation of the "implied emission factors" and available activity data (A) for that year. Supplemental Sect. S2.1 and S2.2 provide additional details regarding the input datasets for activity drivers and emission factors used for both CEDS combustion and process source categories.

While CEDS Step 1 is designed to provide a complete set of historical emission estimates, CEDS Step 2 calibrates

(or scales) these total default emission estimates to existing, authoritative global, regional, and national-level inventories. As described in Hoesly et al. (2018), CEDS uses a "mosaic" scaling approach to retain detailed fuel- and sector-specific information, while maintaining consistent methodology over time. The first step in this calibration procedure is to derive a time series of scaling factors (SF) for each calibration inventory using Eq. (2), calculated as a function of chemical compound, country, sector, and fuel-type (where available). Due to persistent differences and uncertainties in the underlying activity data

and sectoral definitions in each calibration inventory, CEDS emissions are calibrated to total emissions within aggregate scaling sectors (and fuels, where applicable). These aggregate scaling groups are defined for each calibration inventory and are chosen to be broad in order to improve the overlap between CEDS emission estimates and those reported in other inventories. For example, the sum of CEDS emissions from working sectors 1A4a_Commercial-institutional, 1A4b_Residential, and 1A4c_agriculture-forestry-fishing are calibrated to the aggregate 1A4_energy-for-buildings sector in

the EDGAR v4.3.2 inventory. Sections 2.2 and Supplemental Sect. S2.3 provide further details about this calibration procedure and the calibration inventories used to develop the 1970 – 2017 CEDS_GBD-MAPS inventory.

$$\mathrm{SF}_{\mathrm{species}}^{\mathrm{c,\ s,\ f,\ y}} = \frac{\text{calibration inventory } \mathrm{Em}_{\mathrm{species}}^{\mathrm{c,\ s,\ f,\ y}}}{\text{default CEDS } \mathrm{Em}_{\mathrm{species}}^{\mathrm{c,\ s,\ f,\ y}}} \qquad (2)$$





After SFs are calculated in Eq. (2), the second step in the calibration procedure is to extend these SFs forward and backward in time to fill years with missing data. For these time periods, the nearest available SF is applied. If a particular

sector or compound is not present in a calibration inventory, default CEDS estimates are not scaled. For BC and OC emissions, the default procedure in the CEDSv2019-12-23 system was to retain all default BC and OC emission estimates due to limited availability of historical BC and OC emissions. In the CEDS_GBD-MAPS inventory, these species are now calibrated to available regional and national-level inventories (further details in Sect. 2.2). For all other species, the CEDS_GBD-MAPS system uses a sequential scaling methodology where total default emissions for each country are first calibrated to available global

inventories (primarily EDGAR v4.3.2) and calibrated second to regional and national-level inventories, many of which have been updated in this work (Sect. 2.2 and Table 3). This process results in final CEDS_GBD-MAPS emissions that reflect the inventory last used to calibrate the emissions for that country (Fig. 2). Sections 2.2 and S2.3 describe further details and updates to this calibration procedure.

CEDS Step 3 extends the calibrated emission estimates from 1970 back in time to 1750. This process is necessary as

reported emission estimates and energy data are not typically reported with the same level of sectoral and fuel-type detail prior to 1970. Hoesly et al. (2018) provides a detailed description of this historical extension procedure, which is used to derive pre-1970 emissions in the CEDS_Hoesly inventory. The new CEDS_GBD-MAPS inventory only reports more contemporary emissions after 1970 and therefore, does not utilize this historical extension.

CEDS Step 4 aggregates the calibrated country-level CEDS_GBD-MAPS emissions into 17 intermediate gridding sectors

(defined in Table 2). In the CEDSv2019-12-23 system, Step 4 additionally aggregated sectoral emissions from all fuel-types. In contrast, the CEDS_GBD-MAPS system, retains sectoral emissions from the combustion of total coal (hard coal + coal coke + brown coal), solid biofuel, and the sum of liquid oil (light oil + heavy oil + diesel oil) and natural gas, as well as all CEDS process-level emissions (Table 2). Sections 2.4 and 4.2.3 describe the CEDS_GBD-MAPS fuel-specific emissions in further detail.

Lastly, CEDS Step 5 uses normalized spatial distribution proxies to allocate annual country-level emission estimates

on to a 0.5°×0.5° global grid. Annual emissions from the 17 intermediate gridding sectors and four fuel groups are first distributed spatially using compound-, sector-, and year-specific spatial proxies, primarily from the gridded EDGAR v4.3.2 inventory. Supplemental Table S6 provides a complete list of sector-specific gridding proxies, with additional details specific to the CEDS_GBD-MAPS system in Sect. S2.5 and about the general CEDS gridding procedure in Feng et al. (2020). Second, gridded emission fluxes (units: kg m$^{-2}$ s$^{-1}$) are aggregated into 11 final sectors (Table 2) and distributed over 12 months using

sectoral and spatially explicit monthly fractions from the ECLIPSE project (IIASA, 2015) and EDGAR inventory (international shipping only). Relative to CEDSv2019-12-23, the new CEDS_GBD-MAPS inventory retains detailed sub-sector emissions from the aggregate RCO (now RCO-Residential, RC-Commercial, and RCO-Other) and TRA (now On-Road and Non-Road) sectors, as well as separate sectoral emissions from process sources, as well as combustion sources that utilize coal, solid biofuel, and the sum of liquid fuels and natural gas. Table 2 contains a complete breakdown of the definitions of CEDS

working, intermediate gridding, and final sectors. Gridded total NMVOCs are additionally disaggregated into 25 VOC classes following sector- and country-specific VOC speciation maps from the RETRO project (HTAP2, 2013), which are different



from those used in the recent EDGARv4.3.2 inventory (Huang et al., 2017). Similar to the gridding procedure, the same VOC speciation and monthly distributions are applied to sectoral emissions associated with each fuel category.

Final products from the CEDS$_{GBD-MAPS}$ system include total annual emissions from 1970 - 2017 for each country, as well as annual global gridded ($0.5° \times 0.5°$) emission fluxes, both as a function of 11 final source sectors and four fuel-categories (total coal, solid biofuel, liquid fuel + natural gas, and remaining process sources). Section 5 provides additional details on the dataset availability and file formats.

**2.2 Default Emission Calibration Procedure – CEDS$_{GBD-MAPS}$ Update Details**

As described above, default emission estimates for each compound are calibrated (or scaled) in CEDS Step 2 to existing
authoritative inventories as a function of emission sector and fuel type (where available). In the calibration procedure, annual emissions and EFs for each country are first calibrated to available global inventories, then to available regional and national-level inventories, assuming that the latter use local knowledge to derive more accurate regional estimates. Final CEDS$_{GBD-MAPS}$ emission totals for each country therefore reflect the inventory last used to calibrate each compound and sector. Many of these inventories are updated annually and where available, have been updated in this work relative to the CEDSv2019-12-23 system
(Table 3). For example, global CEDS$_{GBD-MAPS}$ emissions are first calibrated to EDGAR v4.3.2 country-level emissions that extend to 2012 for NO$_x$, total NMVOCs, CO, and NH$_3$. CEDS$_{GBD-MAPS}$ emissions from European countries are then calibrated to available EMEP (European Monitoring and Evaluation Programme) (EMEP, 2019) and UNFCCC (United Nations Framework Convention on Climate Change) (UNFCCC, 2019) inventories that extend to 2017, while CO, NMVOCs, NO$_x$ and SO$_2$ emissions from the US, Canada, and Australia are also calibrated to emissions that extend to 2017 from the US NEI
(US EPA, 2019), Canadian APEI (Air Pollutant Emissions Inventory) (ECCC, 2019), and Australian NPI (National Pollutant Inventory) (ADE, 2019), respectively. In addition, emissions of all 7 compounds from China are calibrated to emissions for 2008, 2010, and 2012 from Li et al. (2017c), followed by subsequent calibration to emissions between 2010 and 2017 from Zheng et al. (2018). Relative to the CEDSv2019-23-13 system, regional inventories have also been added to calibrate CEDS$_{GBD-MAPS}$ emissions from India and Africa as described below. Updates to additional regional calibration inventories,
including South Korea, Japan, and other European and Asian countries are not available relative to those used in the CEDSv2019-12-23 system. Table 3 provides a complete list of the inventories used to calibrate CEDS$_{GBD-MAPS}$ default emissions, with additional details in Sect. S2.3.

    Relative to the CEDSv2019-12-23 system, the CEDS$_{GBD-MAPS}$ system adds calibration inventories for two rapidly changing regions, Africa and India. First, CEDS$_{GBD-MAPS}$ emissions from Africa for select sectors are now scaled to the Diffuse
and Inefficient Combustion Emissions in Africa (DICE-Africa) inventory from Marais and Wiedinmyer (2016). This inventory provides gridded ($0.1° \times 0.1°$) emissions for NO$_x$ (= NO + NO$_2$), SO$_2$, 25 speciated VOCs, NH$_3$, CO, BC, and OC for 2006 and 2013 for select anthropogenic sectors and fuels. In this work, default CEDS emissions are calibrated to total DICE-Africa emissions from each country and later re-gridded in CEDS Step 5 using source-specific spatial proxies described in Sect. 2.1. Following the CEDSv2019-12-23 calibration procedure (Supplemental Sect. S2.3), a set of aggregate scaling sectors and fuels





are defined to ensure that CEDS<sub>GBD-MAPS</sub> emissions are calibrated to emissions from consistent sectors and fuel types within the DICE-Africa inventory (Table S3). Briefly, CEDS<sub>GBD-MAPS</sub> 1A3b_Road and 1A4b_Residential emissions are calibrated to DICE-Africa emissions from diesel and gasoline powered cars and motorcycles, as well as biomass and oil combustion associated with residential charcoal, crop residue, fuelwood, and kerosene use. The DICE-Africa inventory also includes emission estimates from gas flares across Africa and ad-hoc oil refining in the Niger Delta, fuelwood use for charcoal

production and other commercial enterprises, and gas and diesel use in residential generators. Marais and Wiedinmyer (2016) state that these particular sources are missing or not adequately captured in existing global inventories. Therefore, depending on the source sector and inventory details, they recommend that these emissions be added to existing global inventories for formal industry and on-grid energy production in Africa (DICE-Africa, 2016). Due to uncertainties in the representation of these sectors in the default CEDS Africa emissions, these sources are not included in the calibration process here. Default

CEDS<sub>GBD-MAPS</sub> emissions from the 1B2_fugitive_pert_gas (gas flaring) sector (derived from the ECLIPSE and EDGAR inventories) are larger than DICE-Africa gas flaring emissions in 2013, suggesting that this source may be accurately represented in the default CEDS<sub>GBD-MAPS</sub> estimates. As described in Sect. S2.3.2, however, residential generator and fuelwood use for charcoal production and other commercial activities are not explicitly represented in CEDS and will be accounted for only to the extent that these sources are included in the underlying IEA activity data and EDGAR process emission estimates.

In the event that the DICE-Africa emissions from these sources are missing in the default CEDS estimates, total 2013 CEDS<sub>GBD-MAPS</sub> emissions from Africa for each compound may be underestimated by up to 11% (Sect. S2.3, Table S4). These values range from 0.7% for $SO_2$ to 11% for CO (Table S4) and all fall within the range of uncertainties typically reported from regional bottom-up inventories (>20%, Sect. 4.2.2). Final emissions from additional sectors or species in CEDS that are not included in the DICE-Africa inventory are set to CEDS<sub>GBD-MAPS</sub> default values.

Second, emissions from India for select sectors are now scaled to the Speciated Multi-pollutant Generator Inventory described by Venkataraman et al. (2018) (hereafter called SMoG-India). This inventory includes gridded emissions (0.25° × 0.25°) of $NO_x$ (as $NO_2$), $SO_2$, total NMVOCs, CO, BC, and OC for the year 2015 from select anthropogenic sectors and fuels (SMoG-India, 2019). Similar to DICE-Africa emissions, the final spatial distribution in the SMoG-India and CEDS<sub>GBD-MAPS</sub> inventories will differ as country-level emissions are calibrated to country totals and spatially re-allocated using CEDS proxies

in Step 5. SMoG-India emissions for each compound are available for 17 sectors and nine fuel types (coal, fuel oil, diesel, gasoline, kerosene, naptha, gas, biomass, process/fugitive). Similar to the DICE-Africa inventory, aggregate calibration groups have been defined to calibrate consistent sectors and fuels between inventories, as described in Sect. S2.3. Briefly default CEDS<sub>GBD-MAPS</sub> emissions for 1A4c_Agriculture-forestry-fishing sector are calibrated to the sum of SMoG-India emissions for agricultural pumps and tractors, 1A4b_Residential emissions are calibrated to the sum of SMoG-India emissions from

residential lighting, cooking, diesel generator use, space and water heating, 1A1a electricity and heat generation sectors are calibrated to SMoG-India thermal power plant emissions, 1A3b road and rail sectors are calibrated to the respective SMoG-India road and rail emissions, and CEDS<sub>GBD-MAPS</sub> industrial working sectors are allocated and calibrated to four SMoG-India industrial sectors: light industry (e.g., mining and chemical production), heavy industry (e.g., iron and steel production),



informal industry (e.g., food production), and brick production. Calculated scaling factors for these sectors are held constant
before and after 2015. CEDS$_{GBD-MAPS}$ emissions do not include contributions from open burning and are not calibrated to
SMoG-India open burning emissions. In cases where SMoG-India emissions are not reported (e.g., power generation from oil
combustion), default CEDS$_{GBD-MAPS}$ emissions are retained. Sect. S2.3.3 provides additional details.

To examine the changes in CEDS$_{GBD-MAPS}$ emissions associated with the incorporation of the SMoG-India and DICE-
Africa calibration inventories, as well as the updated underlying input datasets, Fig. 3 compares the total and sectoral
distribution of CEDS$_{GBD-MAPS}$ and CEDS$_{Hoesly}$ emissions for these two regions in 2014 (year with latest overlapping data). For
the Africa comparison, the left plot in Fig. 3 shows that total NO$_x$, BC, and OC emissions are generally lower in the CEDS$_{GBD-}$
$_{MAPS}$ inventory than in CEDS$_{Hoesly}$. Lower NO$_x$ and OC emissions are largely associated with smaller contributions from on-
road transport and residential combustion, respectively, while lower BC emissions are associated with both lower residential
and on-road transport contributions. Lower emissions of NO$_x$ from the transport sector result from the lower EF used for diesel
vehicles in the DICE-Africa inventory, which reflects the use of inefficient combustion sources (Marais et al., 2019). Compared
to GAINS (2010) and EDGAR v4.3.2 (2012), on-road emissions from Africa in CEDS$_{GBD-MAPS}$ are up to 2.5 Tg lower for NO$_x$,
but within 0.1 Tg for BC. In contrast, due to larger EFs in DICE-Africa, on-road emissions of CO and OC are up to 14.8 and
0.3 Tg higher, respectively, in CEDS$_{GBD-MAPS}$. In addition, Africa emissions from the residential/commercial sectors are
generally lower in CEDS$_{GBD-MAPS}$ than in CEDS$_{Hoesly}$ due to both lower biofuel consumption and a lower assumed EF in the
DICE-Africa inventory (Marais and Wiedinmyer, 2016). These residential BC and OC emission estimates are also lower than
those from GAINS (Klimont et al., 2017). The difference in biofuel consumption is due to different data sources. The DICE-
Africa inventory uses residential wood fuel consumption estimates from the UN while CEDS$_{Hoesly}$ uses data from the IEA.
Both of these sources consist largely of estimates for African countries because there is little country-reported biofuel
consumption data available. The estimation methodologies for both the UN and IEA estimates are not well documented, which
adds to the uncertainty in these values (Sect. 4.2). Total CEDS$_{GBD-MAPS}$ emissions of NMVOCs are larger, primarily due to
increased contributions from solvent use and the energy sector in the EDGAR v4.3.2 inventory, while total emissions of CO,
SO$_2$, and NH$_3$ are relatively consistent between the two CEDS versions.

For the India comparison, the right panel of Fig. 3 shows that total emissions of NO$_x$, CO, SO$_2$, NMVOCs, and OC
are lower in CEDS$_{GBD-MAPS}$. Relative reductions in NO$_x$ emissions are largely associated with on-road transport. Calibrated
CEDS$_{GBD-MAPS}$ transport emissions are 5 Tg smaller than NO$_x$ emissions in CEDS$_{Hoesly}$, largely as a result of lower fuel
consumption levels for gas, diesel, and CNG on-road vehicles used to develop SMoG-India estimates (Sadavarte and
Venkataraman, 2014). Compared to EDGAR and GAINS inventories, NO$_x$ transport emissions are also lower in CEDS$_{GBD-}$
$_{MAPS}$. Causes of other reductions are mixed. For example, lower emissions of SO$_2$ and NMVOCs are largely associated with
the energy sector, while reductions in the industry sector contribute to reduced CO emissions.

To further examine the CEDS$_{GBD-MAPS}$ inventory in these regions, Fig. 4 compares final CEDS$_{GBD-MAPS}$ and CEDS$_{Hoesly}$
emissions for India and Africa to total emissions from two widely used global inventories: GAINS (ECLIPSE v5a) and
EDGAR (v4.3.2). First, Fig. 4 shows the percent difference between the CEDS$_{GBD-MAPS}$ inventory and the GAINS and EDGAR



inventories on the y-axis, against the percent difference between the CEDS$_{Hoesly}$ inventory and GAINS and EDGAR emissions on the x-axis. Percent differences are calculated from total emissions from Africa (left) and India (right) for the year 2012 for

the comparison with EDGAR and for 2010 for the comparison to GAINS (most recent years with overlapping data). The green shaded areas indicate regions where the updated CEDS$_{GBD-MAPS}$ inventory has improved agreement with EDGAR or GAINS relative to the CEDS$_{Hoesly}$ inventory. This comparison shows that the additional calibration of CEDS$_{GBD-MAPS}$ emissions to the SMoG-India inventory generally improves agreement with both the EDGAR and GAINS inventories relative to CEDS$_{Hoesly}$ for all species except black carbon (BC). Calibration to the DICE-Africa inventory generally improves CEDS$_{GBD-MAPS}$

agreement with the EDGAR inventory but not with GAINS (except for OC). Further comparisons to these two inventories are discussed in Sect. 4. While uncertainties in emissions from these inventories are expected to be at least 20% for each compound (discussed in Sect. 3.3), this comparison provides an illustration of the changes between the two CEDS versions relative to two widely used global inventories.

## 2.3 Default BC & OC Calibration – CEDS$_{GBD-MAPS}$ Update Details

Relative to the CEDSv2019-12-23 system, the second largest change to the CEDS$_{GBD-MAPS}$ system is the added calibration of BC and OC emissions in CEDS Step 2. In the v2019-12-23 system, OC and BC were not scaled due to a lack of historical BC and OC emission estimates in regional and global inventories. Due to the focus of the CEDS$_{GBD-MAPS}$ inventory on more recent years, these two compounds are now scaled to available regional and country-level estimates (Table 3), following the same calibration procedure described above for the reactive gases. Unlike the reactive gases, however, BC and OC emissions are

not scaled to the global EDGAR v4.3.2 inventory due to the large reported uncertainties in this inventory (ranging from 46.8% to 153.2% (Crippa et al., 2018)).

To examine the impact of the new BC and OC emissions calibration, in addition to the updated IEA energy consumption data, Fig. 5 and Fig. S2-S3 show time series of global BC and OC emissions from CEDS$_{GBD-MAPS}$ compared to emissions from the CEDS$_{Hoesly}$ inventory. In 2014, respective global annual emissions of BC and OC are 21% and 28% lower

than the CEDS$_{Hoesly}$ inventory and have total global annual emissions in 2017 of 6 and 13 TgC yr$_{-1}$ for BC and OC, respectively. These reductions in global emissions are largely due to the added calibration of emissions from China, Africa, Japan, and other countries in Asia included in the REAS inventory (Fig. S2-S3). Figures 5 and S2-S3 additionally compare CEDS$_{GBD-MAPS}$ emissions to those from the GAINS (ECLIPSE v5a) and EDGAR (v4.3.2) inventories, which generally show improved agreement in BC and OC emissions with the GAINS inventory. CEDS$_{GBD-MAPS}$ emissions between 1990 and 2015 are now 7-

14% lower than GAINS BC emissions, while CEDS$_{GBD-MAPS}$ emissions of OC remain 12-25% higher than GAINS estimates. Further discussion of CEDS$_{GBD-MAPS}$ BC and OC emissions and comparisons to EDGAR and GAINS inventories are below in Sect. 4.1.2. As an additional point of comparison, Bond et al. (2013) report global BC and OC values for the year 2000, derived from averages of energy-related burning emissions from SPEW and GAINS. Reported global estimates of BC and OC are 5 TgC and ~11-14 TgC (16 Tg organic aerosol/ organic mass: organic carbon ratio of 1.1 - 1.4), respectively (Bond et al., 2013).

These also have improved agreement with the CEDS$_{GBD-MAPS}$ estimates of BC and OC in 2000 relative to those in the





CEDS<sub>Hoesly</sub> inventory. Lastly, we note plans for an upcoming update to the core CEDS system to improve historical trends in carbonaceous aerosol by incorporating reported inventory values for total PM2.5 and its ratio with BC and OC emissions.

## 2.4 Fuel Specific Emissions – CEDS<sub>GBD-MAPS</sub> Update Details

Prior to gridding, CEDS<sub>GBD-MAPS</sub> Step 4 combines total country-level emissions for each of the 52 working sectors and nine
fuel groups into 17 aggregate sectors and 4 fuel-groups: total coal (hard coal + brown coal + coal coke), solid biofuel, the sum of liquid fuels (heavy oil + light oil + diesel oil) and natural gas, and all remaining 'process' emissions (Table 2). In contrast, the CEDSv2019-12-23 system aggregates all fuel-specific emissions and reports inventory values as a function of sector only. In CEDS<sub>GBD-MAPS</sub>, country-total emissions from these aggregate sectors and fuel groups are distributed across a $0.5°×0.5°$ global grid using spatial gridding proxies, as discussed in Sect. 2.1 (Table S6). During gridding, the same spatial proxies are
applied to all fuel groups within each sector. In practice, this requires that the gridding procedure be repeated four times for each of the fuel groups. After gridding in CEDS Step 5, both annual country-total and gridded emission fluxes from each fuel group are aggregated to 11 final sectors. Figure S4 demonstrates the level of detail available in the new CEDS<sub>GBD-MAPS</sub> gridded emission inventory by illustrating global BC emissions in 2017 from 1) all source sectors, 2) the residential sector only, 3) residential biofuel-use only, and 4) residential coal-use only. Additional uncertainties associated with the CEDS<sub>GBD-MAPS</sub> fuel-
specific emissions in both the country-total and annual gridded products are discussed further in Sect. 4.2.3

## 3 Results

The new CEDS<sub>GBD-MAPS</sub> inventory provides global emissions of $NO_x$, $SO_2$, NMVOCs, $NH_3$, CO, OC, and BC for 11 anthropogenic sectors (agriculture, energy, industry, on-road, non-road transportation, residential, commercial, other, waste, solvents, international shipping) and four fuel groups (combustion of total coal, solid biofuel, and liquid fuels and natural gas,
and process sources) over the time period between 1970 - 2017. Final country-level emissions are provided as annual time series in units of kilotons per year (kt yr-1) for each sector and fuel-type and include $NO_x$ as emissions of $NO_2$. Final global gridded ($0.5° × 0.5°$) emissions for each compound, sector, and fuel group have been converted to emission fluxes (kg m-2 s-2), distributed over 12 months, and represent $NO_x$ as NO to facilitate their use in earth system models. Total NMVOCs in gridded products are additionally separated into 25 sub-VOC classes. Using a combination of updated energy consumption
data and calibration procedures, CEDS<sub>GBD-MAPS</sub> provides the most contemporary bottom-up global emission inventory to-date, and is the first inventory to report global emissions of multiple atmospheric pollutants from multiple fuel groups and sectors using consistent methodology. The following results section presents an overview of the CEDS<sub>GBD-MAPS</sub> emission inventory, with particular focus on emissions in 2017 and historical trends as a function of compound, sector, fuel type, and world region. Section 4 compares these results to other global emission inventories and discusses the magnitudes and sources of inventory
uncertainties. Known issues in the inventory data at the time of submission are detailed in Sect. S5.



## 3.1 Global Annual Total Emissions in 2017

Figures 6 and 7 show time series from 1970 – 2017 of global annual CEDS_{GBD-MAPS} emissions for each emitted compound. Global CEDS_{GBD-MAPS} emissions for reactive gases in 2017 are 122 Tg for NO_x (as NO_2), 538 Tg for CO, 79 Tg for SO_2, 175 TgC for total NMVOCs, and 61 Tg for NH_3. Global 2017 emissions of carbonaceous aerosol are 13 and 6 TgC for OC and

BC, respectively. The time series in Fig. 6 and 7 additionally show the contributions to global emissions from each of the 11 source sectors (Fig. 6) and four fuel groups (Fig. 7). Each panel in Fig. 6 additionally shows a pie chart with the fractional contribution of each sector to total global emissions in 2017 (outside), while the inner pie chart shows the fractional contributions from each of the fuel groups to each source sector. Numerical values for these fractional contributions are in Table S7. Global totals for 2017 are provided in the center of each pie chart. Global emissions from each compound are

additionally split into contributions from 9 world regions (defined in Table S8) in Fig. 8 to aid in the interpretation of global trends below.

For global 2017 emissions of NO_x, Fig. 6 and Table S7 show that 60% of NO_x emissions are associated with the energy generation (22%), industry (15%), and on-road transportation (23%) sectors. These sectors have the largest contributions from emissions from coal combustion (> 46% for the energy and industry emissions) and the combined

combustion of liquid fuels (oil) and natural gas (with these two fuels accounting for 100% of NO_x on-road emissions). Time series of regional contributions to global emissions in Fig. 8 additionally show 59% of global 2017 NO_x emissions are from China (24 Tg), the combined Other Asia/Pacific/Middle East region (Table S8) (23 Tg), and International Shipping (25 Tg). For global 2017 emissions of remaining gas-phase pollutants, 67% of CO emissions are from the on-road (100%: oil + gas) and residential (86%: biofuel) sectors, 78% of SO_2 emissions are from the energy generation (63%: coal) and industry (38%

coal, 36% process, 25% oil + gas) sectors, 89% of NH_3 emissions are from the agriculture (100%: process) and waste (100%: process) sectors, and emissions of NMVOCs have the largest single contribution (36%) from the energy sector, 99% of which are associated with CEDS_{GBD-MAPS} process sources (Table 2). For carbonaceous aerosol in 2017, 58% of global BC emissions are from the residential (70%: biofuel) and on-road (100%: oil + gas) sectors, while 67% of global OC emissions are from the residential (92%: biofuel) and waste (100%: process) sectors. Fig. 8 shows that in 2017, China is the dominant source of global

CO (144 Tg, 27% of global total), NH_3 (12 Tg, 20% of global total), and BC (1 TgC 24% of global total). By 2017, the combined Other Asia/Pacific/Middle East region surpasses China to become the largest source of global SO_2 (17 Tg, 22% of global total) and OC (3 TgC, 22% of global total). In contrast, Africa is the dominant source of global NMVOCs (48 TgC) in 2017, followed by the Other Asia/Pacific/Middle East region (42 TgC) and China (29 TgC).

As discussed above in Sect. 2 and below in Sect. 4.2.3, the distinction between CEDS combustion and process-level

source categories for all species may result in the underrepresentation of emissions from combustion sources relative to those from CEDS process-level sectors. As shown in Table 2, for example, some combustion emissions from the energy, industry, and waste sectors, such as fossil fuel fires and waste incineration are categorized as CEDS 'process-level' source categories





(Table 2). These emissions are allocated to the final CEDS process category rather than the CEDS total coal, biofuel, or oil and gas categories.

**3.2 Historical Trends in Annual Global Emissions**

Historical emission trends between 1970 and 2017 in Fig. 6 and 7 indicate that global emissions of each compound generally follow three patterns: (1) global CO and $SO_2$ emissions peak prior to 1990 and generally decrease until 2017, (2) global emissions of $NO_x$, BC, and OC peak much later around 2010 and then decrease until 2017, and (3) global emissions of $NH_3$ and NMVOCs continuously increase throughout the entire time period. These trends generally reflect the sector-specific

regulations implemented in dominant source regions around the world. For example, global emissions of CO generally decrease after the incorporation of catalytic converters in North America and Europe around 1990 (Fig. S6 and S7). Despite, however, continued reductions in these regions, global emissions of CO slightly increase between 2002 and 2012 due to simultaneous increases in energy, industry, and residential combustion CO emissions in China, India, Africa, and the Other Asia region (Fig. S8-S11). Global CO emissions then decrease by 9% between 2012 and 2017, largely due to reductions in

industrial coal, residential biofuel, and process energy sector emissions in China (S5, S8, S16-S17) that are associated with the implementation of emission control strategies (reviewed in Zheng et al., 2018), as well as continued reductions in on-road transport emissions in North America and Europe (Fig. S6-S7). Similarly, global $SO_2$ emissions decrease after peaking in 1979, largely due to emission control policies in the energy and industry sectors in North America and Europe (Fig. S6-S7). While simultaneous increases in emissions from coal use in the energy and industry sectors in China result in a brief increase

in global $SO_2$ emissions between 1999 and 2004 (Fig. 6, S8), global $SO_2$ emissions decline by 32% between 2004 and 2017 due to the implementation of stricter emission standards for the energy and industry sectors after 2010 in China (Zheng et al., 2018), as well as continued reductions in North America and Europe (Fig. S6- S8). Regional $SO_2$ emission trends are particularly large with a factor of 9.5 decrease in total $SO_2$ emissions in North America between 1973 and 2017, a factor of 6.9 decrease in Europe between 1979 and 2017, and a factor of 5.9 increase in China between 1970 and 2004, followed by a

factor of 2.6 decrease after 2011 (Fig. 8). While China is the largest global contributor to $SO_2$ emissions between 1994 and 2015, these large regional reductions, coupled with increasing $SO_2$ emissions in the Other Asia region and India (Fig. 8), indicate that future global $SO_2$ emissions will increasingly reflect activities in these other rapidly growing regions.

In contrast to historical emissions of $SO_2$ and CO, global emissions of $NO_x$, BC, and OC peak later between 2011 and 2013. Global emissions then decrease by 7%, 9%, and 7%, respectively by 2017 (Fig. 6). These trends also reflect the sector-

specific regulations implemented in dominant source regions. For $NO_x$ for example, global emissions between 1970 and 2017 are dominated by the combustion of coal, oil, and gas in the on-road transportation, energy generation, industry, and international shipping sectors (Fig. 6, 8). Global on-road transportation emissions are generally flat between 1988 and 2013 due to competing trends across world regions. While more stringent vehicle emission standards result in more than a factor of 2 decrease in on-road transportation $NO_x$ emissions in North America and Europe between 1992 and 2017 (Fig. S6-S7),

transport emissions in China, India, and the Other Asia region simultaneously experience between a factor of 1.3 to 2.8 increase





(Fig. S8-S10). Subsequent reductions between 2013 and 2017 in global on-road emissions correspond to a 12% reduction in on-road transportation emissions in China due to the phase in of stricter emission standards (Zheng et al., 2018), coupled with a continued decrease in emissions from North America and Europe. Global $NO_x$ emissions from the energy and industry sectors increase by up to a factor of 6 between 1970 and 2011 due to regional increases in China, India, Other Asia, and Africa, with

reductions between 2011 and 2017, again largely from reductions in China from stricter emissions control policies for coal fired power plants and coal use in industrial processes (Zheng et al., 2018;Liu et al., 2015). Global emissions of $NO_x$ from waste processing and agricultural activities also increased by a factor of 2 and 65%, respectively, between 1970 and 2017, also contributing to the offset of recent reductions in emissions from regulated combustion sources (Fig. 6). Similar to global $NO_x$ emissions, trends in historical BC and OC emissions reflect a balance between emission trends in North America, Europe and

other world regions, with reduction between 2010 and 2017 largely driven by reductions in emissions from China (Fig. 8, S8). In contrast to $NO_x$ emissions, however, BC and OC emissions are dominated by contributions from biofuel combustion in the residential sector, as well as on-road transportation, industry, and energy sectors for BC and the waste sector for global OC (Fig. 6). Though emissions of BC and OC have a higher level of uncertainty relative to other compounds (Sect. 4), emissions from Africa and the Other Asia region experience steady growth in BC and OC emissions from these sectors. The exceptions

are in China and India, both of which experience a plateau or reduction in BC and OC emissions from the residential, energy (China only), industry, and on-road transportation sectors between 2010 and 2017. In India, reductions in BC and OC emissions from the residential and informal industry sectors are expected to continue under policies to switch to cleaner residential fuels and energy sources, while BC emissions from on-road transport may increase due to increased transport demand (Venkataraman et al., 2018). Similar to trends in $SO_2$ emissions, increasing trends in total OC and BC emissions from Africa,

India, and the Other Asia region, coupled with large decreases in emissions from China, North America, and Europe indicate that global emissions will increasingly reflect activities in these rapidly growing regions.

Trends in historical emissions of NMVOCs and $NH_3$ differ from other pollutants in that they continuously increase between 1970 and 2017. Global emissions of $NH_3$ increase by 81% between 1970 and 2017 and are largely associated with emissions from agricultural practices (75% in 2017) and waste disposal and handling (14% in 2017) (Fig. 6, Table S7). Unlike

emissions from combustion sources, there are no largescale regulations outside of Europe targeting $NH_3$ emissions from agricultural activities, such as livestock manure management. As a result, global agricultural emissions of $NH_3$ increase between 1970 and 2017 by 82%, driven by increases in all regions other than Europe (Fig. 6, S5-S11). Similarly, global $NH_3$ emissions from the waste sector increase by 77% between 1970 and 2017, driven by increases in Latin America, the Other Asia region, Africa, and India (Fig. S5-S11). Global emissions of NMVOCs increase by 40% between 1970 and 2017 and are

largely associated with emissions from the on-road transport, residential, energy, industry, and solvent use sectors (Fig. 6). In contrast to other emitted pollutants, Africa is the largest global source of NMVOC emissions between 2010 and 2017, largely due to continued increases in emissions from the residential (factor of 2.7) and energy (factor of 4) sectors (Fig. S11). Emissions from the Other Asia region are the second largest global source in 2017 and the dominant source between 1989 to 2010 (Fig. 8). China is the third largest source of global NMVOCs between 1996 and 2017. Total NMVOCs in China increase by a factor



of 3.4 between 1970 and 2017 due to activity increases in the solvent, energy, and industry sectors (Zheng et al., 2018), while targeted emission controls for the residential and on-road transport sectors result in their reduced contributions to NMVOC emissions between 2012 and 2017 (Fig. S8). Total emissions of NMVOCs in Europe and North America decrease by up to a factor of 2.4 between 1970 and 2017, due to reductions in all source sectors, except for energy emissions in North America, which increase between 2007 and 2011 and remain flat through 2017 (Fig. S6).

To provide a fuel-centric perspective of global historical emissions trends, Fig. 7 illustrates the contributions from the combustion of coal, solid biofuel, the sum of liquid fuel and natural gas, as well as all remaining CEDS 'process-level' sources (Table 2) to total global emissions between 1970 and 2017. Reductions discussed above between 2010 and 2017 for global emissions of $NO_x$, CO, $SO_2$, BC and OC, are largely associated with reductions in coal combustion from the energy, industry, and residential sectors associated with emission control policies and residential fuel replacement in China, as well as

coal-fired power plant reductions in North America and Europe (Fig. 7, S12, S16). Despite large reductions in emissions, China is still the single largest source of global emissions from coal combustion in 2017 (23-64% for each compound except $NH_3$). Figure S16, however, also shows that emissions from coal combustion are simultaneously increasing in India, the Other Asia/Pacific/Middle East region, and Africa. Specifically, $SO_2$ emissions from coal combustion in India are set to surpass those from China by 2018 if recent CEDS$_{GBD-MAPS}$ trends hold. For biofuel combustion, global emissions of all compounds are

primarily associated with the residential sector (Fig. S13), with recent reductions in biofuel CO, $SO_2$, BC, and OC emissions largely from reductions in China (Fig. S17). In contrast, biofuel emissions from all other regions remain relatively flat or increase between 1970 and 2017, though biofuel emissions of CO, $SO_2$, and OC in India, as well as $SO_2$ emissions in North America and Europe both decrease between 2010 and 2017 (Fig. S17). In 2017, biofuel emissions of all compounds are dominated by emissions from either Africa, India, or the Other Asia/Pacific/Middle East region. For oil and gas combustion,

global emissions of all compounds are primarily associated with on-road transportation, international shipping, and energy and industry ($SO_2$ only) sectors, with general decreases in associated emissions in North America and Europe between 1970 and 2017 and increases in other regions (Fig. S18). Similar to other combustion sectors and fuels, emissions of $NO_x$, CO, NMVOCs, BC, and OC from the combustion of liquid fuels and natural gas in China decrease between 2010 and 2017. Dominant global regions vary by compound (Fig. S18), but are generally either the Other Asia/Pacific/Middle East region, Africa, or

International Shipping. Global CEDS process source emissions, which include contributions from some fuel combustion processes (Table 2), decrease between 2010 and 2017 for CO, $SO_2$, BC, and OC. These trends are primarily associated with reductions in emissions from the energy and industry sectors. In contrast, process source contributions to $NO_x$, $NH_3$, and NMVOCs increase over this same time period due to increases in non-combustion agricultural and solvent use emissions, as well as emissions from waste disposal and energy generation and transformation. Increases in emissions from these sectors

between 1970 – 2017 drive the continuous increases in $NH_3$ and NMVOCs, discussed above. China, the combined Other Asia/Pacific/Middle East region, and Africa are all dominant sources of process level emissions in 2017 (Fig. S19).





## 4 Discussion

### 4.1 Comparison to Global Inventories

#### 4.1.1 Comparison to CEDS$_{Hoesly}$ Inventory

As a result of the similar methodologies, Fig. 6 shows that CEDS$_{GBD-MAPS}$ and CEDS$_{Hoesly}$ emission inventories predict similar magnitudes and historical trends in global emissions of each compound between 1970 and 2014. The two inventories, however, diverge in recent years due to the incorporation of updated activity data and both updated and new calibration emission inventories included in the CEDS$_{GBD-MAPS}$ system. For global emissions of NO$_x$, CO, and SO$_2$, the CEDS$_{GBD-MAPS}$ emissions are smaller than the CEDS$_{Hoesly}$ emissions after 2006 and show a faster decreasing trend. By 2014, global emissions of these

compounds are between 7 and 21% lower than previous CEDS$_{Hoesly}$ estimates. These differences are largely associated with large emission reductions in China as a result of the updated national-level calibration inventory from Zheng et al. (2018), along with the added DICE-Africa (Marais and Wiedinmyer, 2016) and SMoG-India (Venkataraman et al., 2018) calibration inventories. Differences in emissions from India and Africa in the two CEDS inventories are discussed in Sect. 2 (Fig. 3) and combined, account for ~60% of the reduction in global NO$_x$ emissions, 23% of the reduction in global CO, and 14% of the

reduction in global SO$_2$. The largest differences between these two inventories in India and Africa are the reduced NO$_x$ emissions from the transport sector, as well as reduced energy emissions of SO$_2$ in India. Remaining differences between NO$_x$ and SO$_2$ emissions in the two CEDS inventories are largely associated with the updated China emission inventory from Zheng et al. (2018), which reports lower emissions in 2010 and 2012 than a previous version of the MEIC inventory that was used to calibrate China emissions in the CEDS$_{Hoesly}$ inventory (Li et al., 2017c). These emission reductions are largely associated with

the industrial and residential sectors in China and are partially offset by a simultaneous increase in transportation emissions of all compounds relative to CEDS$_{Hoesly}$.

For global emissions of NH$_3$ and NMVOCs, these species remain relatively unchanged between the CEDS$_{Hoesly}$ and CEDS$_{GBD-MAPS}$ inventories. In 2014 CEDS$_{GBD-MAPS}$ emissions are 5% higher than CEDS$_{Hoesly}$ emissions for NMVOCs and 2% lower than CEDS$_{Hoesly}$ global NH$_3$ emissions. Emissions of NH$_3$ remain relatively unchanged (within <2%) from dominant

source regions, including India, Africa (Fig. 3), and China. In contrast, emissions of NMVOCs from Africa and China in the DICE-Africa and Zheng et al. (2018) calibration inventories are larger than those in the CEDS$_{Hoesly}$ inventory. Global emissions of NMVOCs are also higher in EDGARv4.3.2 inventory relative to the previous version used in the CEDS$_{Hoesly}$ inventory.

Global emissions of OC and BC have the largest differences between the two CEDS inventories, with CEDS$_{GBD-MAPS}$ emissions consistently smaller than CEDS$_{Hoesly}$ emissions between 1970 and 2014. By 2014, CEDS$_{GBD-MAPS}$ emissions of BC

and OC are 24 and 33% smaller than corresponding CEDS$_{Hoesly}$ emissions. In the CEDS$_{Hoesly}$ inventory, default emissions of BC and OC are not calibrated and therefore these differences are largely associated with the added calibration inventories, discussed in Sect. 2 and shown in Table 3. As shown in Fig. S2-S3, the added calibration of BC and OC emissions leads to a reduction in global CEDS$_{GBD-MAPS}$ emissions of OC in all calibrated regions, and a reduction in BC emissions in all regions other than India. In India, increases in industry and residential BC emissions from the SMoG-India calibration inventory result



in a slight increase in BC emissions relative to the CEDS$_{Hoesly}$ inventory (Fig. 3). Waste emissions of OC and BC are also reduced in the CEDS$_{GBD-MAPS}$ inventory due to updated assumptions for the fraction of waste burned (Sect. S1.1). As discussed in Hoesly et al. (2018) and further below, BC and OC emissions typically have the largest uncertainties of all the emitted species and their recent changes in the residential and waste sectors are particularly uncertain.

The relative contributions of each source sector to emissions in the two CEDS versions are additionally shown in Fig.
S20. This comparison shows that the fractional sectoral contributions to global emissions in 2014 are the same to within 10% in the two CEDS inventories. The largest differences are a 9% increase in the relative contribution of on-road transportation emissions of CO and reductions in the relative contribution of waste emissions across all compounds. These trends reflect the large update to default waste emissions described above as well as changes associated with the DICE-Africa and national China calibration inventories.

**4.1.2 Comparison to Other Global Inventories (EDGAR & GAINS)**

Figure 6 additionally provides a comparison of the CEDS$_{GBD-MAPS}$ global emissions to those from two widely used inventories: EDGAR v4.3.2 (Crippa et al., 2018;EC-JRC, 2018) and ECLIPSE v5a (GAINS) (IIASA, 2015;Klimont et al., 2017). For a comparison of global emissions across similar emission sectors, the EDGAR v4.3.2 inventory in Fig. 6 includes emissions from all reported sectors (including international shipping), except for those from agricultural waste burning and domestic and
international aviation. Similarly, the GAINS ECLIPSE v5a baseline scenario inventory in Fig. 6 includes all reported emissions, other than those from agricultural waste burning. These include contributions from aggregate residential and commercial combustion sources ('dom'), energy generation ('ene'), industrial combustion processes ('ind'), road and non-road transportation ('tra'), agricultural practices ('agr'), and waste disposal ('wst'). GAINS ECLIPSE v5a baseline estimates for international shipping emissions are also included in Fig. 6. A table with sectoral mappings of the CEDS$_{GBD-MAPS}$, EDGAR
v4.3.2, and GAINS inventories in provided in Table S9.

The comparison in Fig. 6 shows that global emissions of all compounds in the CEDS$_{GBD-MAPS}$ inventory are consistently larger than in the EDGAR v4.3.2 inventory (Crippa et al., 2018). Global CEDS$_{GBD-MAPS}$ emissions of NO$_x$, SO$_2$, CO, and NMVOCs are at least 27% larger, while global emissions of NH$_3$, BC, and OC are within 52%. Figure S21 indicates that differences in global BC and OC emissions are largely due to higher waste and residential and commercial emissions in
the CEDS$_{GBD-MAPS}$ inventory. Figure 6, however also shows that the trends in global emissions are similar between EDGAR v4.3.2 and CEDS$_{GBD-MAPS}$ for most compounds. For example, between 1970 and 2012, global emissions of SO$_2$, NH$_3$, NMVOCs, and BC peak in the same years. Global CO and NO$_x$ emissions both peak one year earlier in the CEDS$_{GBD-MAPS}$ inventory, but otherwise follow similar historical trends. Trends in OC emissions are the most different between the two inventories with a peak in emissions in 1988 in the EDGAR inventory, compared to 2012 in the CEDS$_{GBD-MAPS}$ inventory. A
comparison of relative sectoral contributions in Fig. S21 shows that these differences in OC emissions are largely due to the residential and commercial sectors, which may be underestimated in the EDGAR v4.3.2 inventory relative to GAINS (Crippa et al., 2018) and CEDS$_{GBD-MAPS}$. Both inventories also show a net increase in global emissions of all compounds other than



SO$_2$ between 1970 and 2012. Global SO$_2$ emissions follow a similar trend until 2007, after which, the emissions in CEDS$_{GBD-}$

$_{MAPS}$ decrease at a faster rate than in EDGAR v4.3.2. These differences are largely due to the energy sector, which increase

between 2006 and 2012 in EDGAR, and decrease as a result of emission reductions in China in the CEDS$_{GBD-MAPS}$ inventory

(Fig. S21). For all other compounds, the rate of increase in emissions between 1970 and 2012 is also slightly different between

the two inventories. For example, NH$_3$ emissions in the CEDS$_{GBD-MAPS}$ inventory increase by 74% compared to a 139% increase

in EDGAR. In contrast, BC and OC emissions increase at a faster rate in the CEDS$_{GBD-MAPS}$ inventory. Due to similar sources

of uncertainty and the additional calibration of CEDS$_{GBD-MAPS}$ emissions to EDGAR (except for BC and OC), levels of

uncertainty between the two inventories are expected to be similar, as discussed further in Sect. 4.2.

Similar to the comparison with EDGAR emissions, Fig. 6 also shows that global emissions in the CEDS$_{GBD-MAPS}$

inventory are generally larger than emission estimates from the GAINS model, published as part of the ECLIPSE v5a inventory

(referred to here as GAINS) (Klimont et al., 2017). Two exceptions are for SO$_2$ emissions, which are up to 6% lower than

GAINS in select years, and BC emissions, which are consistently 5-15% lower than GAINS for all years. While the sectoral

definitions may slightly differ between these inventories, Fig. S22 shows that these differences are largely due to different

trends in energy and industry SO$_2$ emissions between 2005 and 2015 and consistently lower BC emissions from the residential

and commercial sector in the CEDS$_{GBD-MAPS}$ inventory. For all years with overlapping data between 1990 and 2015, the

absolute magnitude of global emissions are within ±15% for NO$_x$, SO$_2$, NH$_3$, and BC, within 22% for CO and OC, and within

50% for NMVOCs. Historical trends in each inventory are also similar for all compounds other than CO and NMVOCs (Fig.

6). Peak global emissions occur between 2010 and 2012 for NO$_x$, BC, and OC, while both inventories show a net decrease in

emissions in SO$_2$ and a net increase in emissions of NH$_3$. In contrast, GAINS emissions of CO peak in 2010, while CEDS$_{GBD-}$

$_{MAPS}$ emissions peak in 1990. The largest differences in historical trends are for global NMVOC emissions with GAINS

showing a 3% decrease between 1990 and 2010, while CEDS$_{GBD-MAPS}$ NMVOC emissions increase by 13% over this same

time period (Fig. 6). Sectoral contributions between the two inventories in Fig. S22 indicates that these differences are largely

due differences in the energy, industry, and agricultural emissions of NMVOCs. Uncertainties in the GAINS model have been

previously estimated to fall between 10% and 30% in Europe for gas-phase species (Schöpp et al., 2005) and within the

uncertainty estimates for BC and OC of other global bottom-up inventories (Klimont et al., 2017;Bond et al., 2004), as

discussed in the following section.

## 4.2 Uncertainties

The level and sources of uncertainty in the CEDS$_{GBD-MAPS}$ inventory are similar to those in the CEDS$_{Hoesly}$ inventory, which

are largely a function of uncertainty in the activity data, emission factors, and country-level inventories. As these uncertainties

have been previously discussed in Hoesly et al. (2018), we have not performed a formal uncertainty analysis here, but rather

provide a brief summary of the sources of uncertainty associated with this work. We note plans for a robust uncertainty analysis

in an upcoming release of the CEDS core system. While this section highlights many of the challenges associated with

estimating comprehensive and accurate global bottom-up emission inventories, such inventories remain vital for their use in chemistry and climate models and for the development and evaluation of future control and mitigation strategies.

### 4.2.1 Uncertainties in Global Bottom-Up Inventories

Uncertainties in bottom-up emission inventories vary as a function of space, time, and compound, making total uncertainties difficult to quantify. Default emission estimates in the CEDS system will be subject to uncertainties in underlying activity
data, such as IEA energy consumption data, as well as activity drivers for process-level emissions. Knowledge of accurate emission factors also drive inventory uncertainties as these are not often available for all sectors in countries with emerging economies, and are heavily dependent on the use, performance, and enforcement of control technologies within each sector and country (e.g., Zhang et al., 2009;Wang et al., 2015). In general, these uncertainties are expected to decrease in recent years with the improvement of data collection and reporting standards. The most recent years in CEDS$_{GBD-MAPS}$, however, are still
subject to increased levels of uncertainty as the degree of compliance with control measures are often variable or unknown (e.g., Wang et al., 2015;Zheng et al., 2018) and recent activity and regional emissions data are often updated as new information becomes available in each version release. In addition, default CEDS emissions after 2010 currently rely on the projection of emission factors from the GAINS EMF30 model for sectors and countries where contemporary regional scaling inventories are not available.

As the CEDS system uses a "mosaic" approach and incorporates information from other global and national-level inventories, the final CEDS$_{GBD-MAPS}$ emissions will also be subject to the same sources and levels of uncertainty as these external inventories. For example, as discussed in Sect. 2.1, default process-level emissions in CEDS$_{GBD-MAPS}$ are derived using emissions from the EDGAR v4.3.2 inventory, with many countries additionally calibrated to this inventory during Step 2. As reported and discussed in Crippa et al. (2018), EDGAR v4.3.2 emissions for 2012 at the regional level are estimated to have
the smallest uncertainties for $SO_2$, between 14.4% and 47.6%, with uncertainties of $NO_x$ between 17.2% and 69.4% (up to 123.5% for Brazil), CO between 25.9% and 123.4% (lower for industrialized countries), and NMVOCs between 32.7% and 147.5% (lower for industrialized countries). Emissions of $NH_3$ are highly uncertain in all inventories (186% to 294.4% in EDGAR) due to uncertainties in the reporting of agricultural statistics and emission factors that will depend on individual farming practices, biological processes, and environmental conditions (e.g., Paulot et al., 2014). As noted in Crippa et al.
(2018) and Klimont et al. (2017), EDGAR v4.3.2 and GAINS uncertainty estimates for BC and OC fall within the factor of two range that has been previously estimated by the seminal work of Bond et al. (2004). While CEDS$_{GBD-MAPS}$ emissions are not calibrated to EDGAR v4.3.2 BC and OC emissions, estimates are derived from similar sources and are therefore expected to be consistent with uncertainties in both EDGAR and other global bottom-up inventories. It should also be noted that these reported uncertainty estimates from EDGAR only reflect the uncertainties associated with the emission estimation process and
do not account for the potential of missing emissions sources or super-emitters within a given sector (Crippa et al., 2018).

To evaluate and improve the accuracy of these bottom-up emission estimates, inventories are increasingly using information from high-resolution satellite retrievals, particularly for major cities, large area and natural sources, and large point



sources (e.g., Li et al., 2017a;McLinden et al., 2016;Streets et al., 2013;van der Werf et al., 2017;Beirle et al., 2011;McLinden et al., 2012;Lamsal et al., 2011;Zheng et al., 2019). For example, both the CEDS$_{Hoesly}$ and CEDS$_{GBD-MAPS}$ inventories

incorporate SO$_2$ emission estimates derived using satellite retrievals in McLinden et al. (2016) to account for previously missing SO$_2$ point sources in the CEDS 1B2_Fugitive-petr-and-gas sector (described further in the supplement of Hoesly et al. (2018)), with additional use of satellite data planned for a future CEDS core release. With the continued advancement of satellite-retrievals, the development of source and sector-specific inventories, such as CEDS$_{GBD-MAPS}$, will continue to provide new opportunities for the application of new satellite-based inventories, which will aid in the quantification of spatial and

temporal emissions from distinct sources associated with specific sectors and fuel-types that may not be accurately estimated using conventional-bottom up approaches.

### 4.2.2. Uncertainties in Regional-Level Calibration Inventories

Similar to the CEDS$_{Hoesly}$ inventory, the CEDS$_{GBD-MAPS}$ emissions will also reflect the uncertainties associated with the inventories used for the calibration procedure. The inventories with the largest impact on the CEDS$_{GBD-MAPS}$ emissions will be

those from China from Zheng et al. (2018), the DICE-Africa emission inventory from Marais and Wiedinmyer (2016), and the SMoG-India inventory from Venkataraman et al. (2018). While formal uncertainty analyses were not performed for all of these inventories, similar bottom-up methods used in these studies will result in similar sources of uncertainties (activity and emission factors) as the global inventories. For example, Zheng et al. (2018) state that the largest sources of uncertainties are the accuracy and availability of underlying data (reviewed in Li et al. (2017b)) and that the levels of uncertainty for China

emissions between 2010 and 2017 are expected to be similar to previous national-level bottom-up inventories derived using similar data sources and methodology, such as Zhao et al. (2011), Lu et al. (2011), and Zhang et al. (2009). Similar to global inventories, these previous regional studies estimate much lower levels of uncertainty for SO$_2$ and NO$_x$ ( ±16% and -13 to +37% respectively) than for CO (70%) and OC and BC emissions (-43 to +258% and -43 to +208%, respectively). Some sectors in China and other regions are particularly uncertain, as discussed further below.

Regional and national inventories, however, have the added benefit of using local knowledge to reduce potential uncertainties in emission factors and missing emission sources. For example, Marais and Wiedinmyer (2016) note that the DICE-Africa emissions are uncertain due to gaps in fuel consumption data, however, this inventory also includes sources frequently missing in global inventories such as widespread diesel/petrol generator use, kerosene use, and ad-hoc oil refining, and have used emission factors for on-road car and natural gas flaring that are more representative of the inefficient fuel

combustion conditions in Africa (Marais and Wiedinmyer, 2016;Marais et al., 2019). As discussed in Sect. 2, the CEDS$_{GBD-MAPS}$ inventory may still underestimate total emissions from some of these sources (up to 11% in 2013; Sect. 2.2.3), but otherwise will have uncertainties for total Africa emissions similar to the DICE-Africa inventory. For emissions in India, uncertainties also arise from missing fuel consumption data and the application of non-local or uncertain emission factors. Venkataraman et al. (2018), however, is one of the few studies to present a detailed uncertainty analysis of their inventory and

use the propagation of source-specific activity data and emission factors to estimate that total emission uncertainties are smaller



for $SO_2$ (-20 to 24%), than for $NO_x$ (-65 to 125%) and NMVOCs (-44 to +66%). While uncertainties are not explicitly reported for OC and BC emissions, Fig. 1 in Venkataraman et al. (2018) indicates that uncertainties in these emissions are between -60% to + 95%, consistent with BC and OC uncertainties reported in other bottom-up inventories. We also note the ongoing work to improve the accuracy of highly uncertain emission sectors in a future release of the SMoG-India inventory, through

the CarbOnaceous AerosoL Emissions, Source apportionment and ClimatE impacts (COALESCE) project (Venkataraman et al., 2020).

Though the inclusion of these regional inventories can improve the accuracy of the global CEDS system, Hoesly et al. (2018) note that large uncertainties may persist, even in developed countries with stringent reporting standards. In the US for example, it has been shown that compared to the US National Emissions Inventory (US NEI), total $NO_x$ emissions from

on-road and industrial sources in some regions may be overestimated by up to a factor of two (e.g., Travis et al., 2016). In addition, $NH_3$ emissions in agricultural regions in winter may be underestimated by a factor of 1.6 to 4.4 (Moravek et al., 2019), and national and regional emissions of NMVOCs from oil and gas extraction regions, solvents, and the use of personal care products may also be underestimated by up to a factor of 2 (McDonald et al., 2018;Ahmadov et al., 2015).

### 4.2.3 Uncertainties in Sectoral and Fuel Contributions

Emissions reported as a function of individual source sectors are typically considered to have higher levels of uncertainty than those reported as country totals, due to the cancelation of compounding errors (Schöpp et al., 2005). Source sectors with the largest levels of uncertainty in CEDS$_{GBD-MAPS}$ estimates are generally consistent with other inventories, which include waste burning, residential emissions, and agricultural processes (Hoesly et al., 2018). This higher level of sectoral uncertainty is reflected in the relatively larger uncertainties discussed above in global emissions of OC, BC, and $NH_3$ relative to other gas-

phase species. In general, uncertainties from these sources are larger due the difficulty in accurately tracking energy consumption statistics and uncertainties in the variability of source-specific emission factors, which will depend on local operational and environmental conditions. For example, residential emission factors from heating and cooking vary depending on technology-used and operational conditions (e.g, Venkataraman et al., 2018;Carter et al., 2014;Jayarathne et al., 2018), while soil $NO_x$ emissions and $NH_3$ from wastewater and agriculture result from biological processes that depend on local

practices and environmental conditions (e.g., Chen et al., 2012;Paulot et al., 2014). While uncertainties are not always reported at the sectoral level, Venkataraman et al. (2018) do report that industry emissions of $NO_x$ and NMVOCs in the SMoG-India inventory actually have larger uncertainties than those from the transportation, agriculture, and residential (NMVOCs only) sectors, while the relative uncertainties for $SO_2$ emissions follow the opposite trend. For total fine particulate matter emissions, Venkataraman et al. (2018) estimate that the sectors with the largest uncertainties are the residential and industry emissions.

Similarly, Lei et al. (2011) estimate that BC and OC emissions from the residential sector in China have the largest inventory uncertainties, while Zhang et al. (2009) and Zheng et al. (2018) also report relatively smaller uncertainties from power plants and heavy industry in China due to known activity data, local emission factors, pollution control technologies, and direct



emissions monitoring. Overall, the mosaic calibration procedure in the CEDS system will result in similar levels of uncertainties as these regional calibration inventories.

With the release of fuel-specific information in the CEDS$_{GBD-MAPS}$ inventory, additional uncertainties in the allocation of fuel types is expected. In this work, activity data at the detailed sector and fuel level are taken from the IEA World Energy statistics (IEA, 2019) and are subject to the same sources of uncertainty. Emission factors for CEDS working sectors and fuels (Table S2) are derived from GAINS. In general, emissions from solid biofuel combustion are considered to be less certain than fossil fuel consumption due to large uncertainties in both fuel consumption and EFs, particularly in the residential and

commercial sectors. For example, by combining information from EDGAR v4.3.2 (Crippa et al., 2018) and a recent TNO-RWC (Netherland Organization for Applied Scientific Research, Residential Wood Combustion) inventory from Denier van der Gon et al. (2015), Crippa et al. (2019) estimated that uncertainties in emissions from wood combustion in the residential sector in Europe are between 200 to 300% for OC, BC, and NH$_3$. Crippa et al. (2019) also report that these uncertainties are largely driven by uncertainties in regional emission factors, as uncertainties in biofuel consumption are estimated to be between

38.9 and 59.5%. These uncertainties, however, are still larger than those estimated for fossil fuel consumption in many countries. As noted in Hoesly et al. (2018), increased levels of uncertainty in fossil fuel emissions are also expected in some countries, including the consumption and emission factors related to coal combustion in China (e.g., Liu et al., 2015;Guan et al., 2012;Hong et al., 2017), which will have the largest impacts on CEDS$_{GBD-MAPS}$ emissions of NO$_x$, SO$_2$, and BC. Specific to the CEDS$_{GBD-MAPS}$ fuel inventory, additional uncertainties may arise from the potential underestimation of total coal, oil and

gas, and biofuel emissions associated with fugitive emissions and gas flaring in the energy sector, as well as waste incineration in the waste sector. As discussed above and in Hoesly et al. (2018), fugitive emissions are highly uncertain. The degree of underestimation in combustion-fuel contributions will be dependent on the fractional contribution of process level emissions in these sectors relative to those from coal, biofuel, and oil and gas combustion (Table S7). Additional uncertainties in the gridded fuel-specific products are discussed in the following section.

## 760   4.2.4 Uncertainties and limitations in gridded emission fluxes

As noted in Sect. 2.1, global gridded CEDS$_{GBD-MAPS}$ emission fluxes are provided to facilitate their use in earth system models. Relative to the reported country-total emission files, additional uncertainties are introduced in the 0.5°×0.5° global gridded CEDS$_{GBD-MAPS}$ emission fluxes through the use of source-specific spatial gridding proxies in CEDS Step 5. As noted in Sect. 2.1, historical spatial distributions within each country are largely based on normalized gridded emissions from the EDGAR

v4.3.2 inventory. These spatial proxies are held constant after 2012, which serve to increase the uncertainties in spatial allocation in large countries in recent years. The magnitude of this uncertainty will depend on the specific compound and sector. For example, gridded emissions from the energy sector will not reflect the closure or fuel-switching of individual coal-fired power stations after 2012. Changes in total country-level emissions from this sector and fuel-type, however, will be accurately reflected in the total country-level emission files. This source of uncertainty is also present in the CEDS$_{Hoesly}$

inventory. An additional source of uncertainty in the gridded emissions is that the same spatial allocations are applied uniformly



across emissions of all three fuel-types within each source sector. This may lead to additional uncertainties if, for example, emissions from the use of coal, biofuel, and 'other' fuels within each sector are spatially distinct. These uncertainties, however, do not impact the final country-level CEDS_GBD-MAPS products because they are not gridded.

Lastly, while CEDS_GBD-MAPS emissions provide a global inventory of key atmospheric pollutants, this inventory does not include a complete set of sources or species required for GCM or CTM simulations of atmospheric chemical processes. As noted in Sect. 2, neither CEDS_Hoesly nor CEDS_GBD-MAPS estimates include emissions from large or small open fires, which must be supplemented with additional open-burning inventories, such as the Global Fire Emissions Database (GFED, 2019;van der Werf et al., 2017) or Fire INventory from NCAR (FINN, 2018;Wiedinmyer et al., 2011). In addition, simulations of atmospheric chemistry require emissions from biogenic sources, typically supplied from inventories, such as the Model of

Emissions of Gases and Aerosols from Nature (MEGAN, 2019;Guenther et al., 2012). Other sources to consider in atmospheric simulations include volcanic emissions, sea spray, and windblown dust. In addition, the CEDS system does not include dust emissions from windblown and anthropogenic sources such as roads, combustion, or industrial process. Anthropogenic dust sources may contribute up to ~10% of total fine dust emissions in recent years and are important to consider when simulating concentrations of total atmospheric particulate matter (Philip et al., 2017). Lastly, the CEDS_GBD-MAPS inventory also excludes

emissions of greenhouse gases such as methane and carbon dioxide ($CH_4$, $CO_2$). These compounds are included in the CEDS_Hoesly inventory.

## 5 Data availability

The source code for the CEDS_GBD-MAPS system is available on GitHub ([https://github.com/emcduffie/CEDS/tree/CEDS_GBD-MAPS](https://github.com/emcduffie/CEDS/tree/CEDS_GBD-MAPS) and [https://doi.org/10.5281/zenodo.3865670](https://doi.org/10.5281/zenodo.3865670) (McDuffie et al., 2020a)). Final products from the CEDS_GBD-MAPS system

include total annual emissions for each country as well as annual global gridded (0.5°×0.5°) emission fluxes for the years 1970 – 2017. Both products are available on Zenodo ([https://doi.org/10.5281/zenodo.3754964](https://doi.org/10.5281/zenodo.3754964)) (McDuffie et al., 2020c) and report total emissions and gridded fluxes as a function of 11 final source sectors and four fuel categories (total coal, solid biofuel, oil + gas, process). Time series of annual country-total emissions from 1970 – 2017 are provided in units of kt yr$^{-1}$ and provide $NO_x$ emissions as $NO_2$. These data do not speciate total NMVOCs into sub-VOC classes. In these .csv files, total anthropogenic

emissions for each country are calculated as the sum of all sectors and fuel-types within each country. For the global gridded products, emission fluxes of each compound as a function of 11 sectors and four fuel types are available for each year in individual netCDF files. These data are in units of kg m$^{-2}$ s$^{-1}$ and provide $NO_x$ emissions as NO. Total NMVOCs are speciated into 25 sub-VOC classes as described in Sect. 2. For consistency with the CEDS data released for CMIP6 (CEDS, 2017a, b), gridded anthropogenic fluxes for 1970-2017 are additionally available in the CMIP6 format. Note that $NO_x$ is in units of $NO_2$

in this format. Additional file format details are in the README.txt file in the Zenodo repository ([https://doi.org/10.5281/zenodo.3754964](https://doi.org/10.5281/zenodo.3754964)).



To provide an example of the products and file formats available for download from the full CEDS$_{GBD-MAPS}$ repository, we have also prepared an additional data 'snapshot' inventory that provides emissions in all three file formats described above, for the 2014 – 2015 time period (McDuffie et al., 2020b). The gridded data are provided as monthly averages for the Dec 2014

– Feb 2015 time period, while the annual data include total emissions from both 2014 and 2015. These data can be downloaded from https://doi.org/10.5281/zenodo.3833935, and are further described in the associated README.txt file.

**6 Summary and Conclusions**

We describe the new CEDS$_{GBD-MAPS}$ global emission inventory for key atmospheric reactive gases and carbonaceous aerosol from 11 anthropogenic emission sectors and four fuel types (total coal, solid biofuel, and liquid fuel and natural gas combustion

and remaining process-level emissions) over the time period from 1970 – 2017. The CEDS$_{GBD-MAPS}$ inventory was derived from an updated version of the Community Emissions Data System, which incorporates updated activity data for combustion and process-level emission sources, updated calibration inventories, the added calibration of BC and OC emissions, and adjustments to the aggregation and gridding procedures to enable the extension of emission estimates to 2017 while retaining sectoral and fuel-type information. By incorporating new regional calibration inventories for India and Africa, default

CEDS$_{GBD-MAPS}$ emissions are now lower than previous CEDS$_{Hoesly}$ estimates for all compounds in these regions other than NMVOCs in Africa and BC in India. These updates improve the agreement of CEDS$_{GBD-MAPS}$ Africa emissions with those from EDGAR v4.3.2, as well as the agreement of all India emissions other than BC with both the EDGAR (2012) and GAINS (2010) inventories. The added calibration of default BC and OC estimates also reduces these global emissions by up to 21% and 28%, respectively, relative to the CEDS$_{Hoesly}$ inventory. This reduction improves CEDS$_{GBD-MAPS}$ agreement with both

GAINS and EDGAR global estimates of BC and OC, particularly in recent years. The resulting CEDS$_{GBD-MAPS}$ inventory provides the most contemporary global emission inventory to-date for these key atmospheric pollutants and is the first to provide their global emissions as a function of both detailed source sector and fuel type.

Global 2017 emissions from the CEDS$_{GBD-MAPS}$ inventory suggest that the combustion of coal and oil and gas in the energy and industry sectors are the largest global sources of SO$_2$ emissions, while CO is primarily from on-road transportation

and biofuel combustion in the residential sector. Global emissions of both compounds peak by 1990 and decrease until 2017 as a result of continuous reductions in on-road transport emissions in Europe, North America as well as reductions in coal combustion emissions from the energy and industry sectors across these regions and in China. In contrast, global NO$_x$, BC, and OC emissions peak later between 2010 and 2012, but also decrease until 2017 due to reductions in North America, Africa, and China. Dominant sources of NO$_x$ in 2017 are from international shipping energy, industry and on-road transportation

sectors. Major sources of BC emissions are from residential biofuel combustion and on-road transportation, while dominant OC sources are from the residential biofuel and the waste sector. Besides international shipping, China is the largest regional source of global NO$_x$ and BC emissions in 2017, while countries across Asia and Middle East contribute more than other regions to global OC emissions. As emissions in North America, Europe, and China continue to decrease, global emissions of



NOx, CO, SO2, BC, and OC will increasingly reflect emissions in rapidly growing regions such as India, Africa, and countries

in Asia and the Middle East. Lastly, in contrast to other compounds, global emissions of NMVOCs and NH3 continuously increase over the entire time period. These increases are predominantly due to increases in agriculture emissions in nearly all world regions, as well as NMVOCs from increased waste, energy sector, and solvent use emissions. In 2017, global emissions of these compounds have the largest regional contributions from Africa, India, China, and countries in Asia and the Middle East.

Historical global emission trends in the CEDS<sub>GBD-MAPS</sub> inventory are generally similar to those in three other global inventories, CEDS<sub>Hoesly</sub>, EDGAR v4.3.2, and ECLIPSE v5a (GAINS). Relative to the CEDS<sub>Hoesly</sub> inventory, however, CEDS<sub>GBD-MAPS</sub> emissions diverge in recent years, particularly for NOx, CO, SO2, BC, and OC emissions. In addition to the use of updated underlying activity in the CEDS<sub>GBD-MAPS</sub> inventory, emissions of these compounds were most impacted by the updated CEDS calibration inventories, including those for China, India, and Africa. These same updates also contribute to the

different trends in global NOx, CO, and SO2 emissions after 2010 between CEDS<sub>GBD-MAPS</sub> and the GAINS and EDGAR inventories. Global emissions between 1970 and 2017 from the CEDS<sub>GBD-MAPS</sub> inventory are larger than the CEDS<sub>Hoesly</sub> emissions for all compounds other than NMVOCs and are consistently higher than all emissions from EDGAR v4.3.2. Global CEDS<sub>GBD-MAPS</sub> emissions are also larger than GAINS emissions, except for BC and select years of SO2 emissions.

    Due to similar bottom-up methodologies and the use of EDGAR v4.3.2 data in the CEDS system, country-level

CEDS<sub>GBD-MAPS</sub> emissions are expected to have similar sources and magnitudes of uncertainty as those in the CEDS<sub>Hoesly</sub>, EDGAR v4.3.2, GAINS, and calibration emission inventories. These inventories consistently predict the smallest uncertainties in emissions of SO2 and the largest for emissions of NH3, OC, and BC. The latter three compounds largely depend on accurate knowledge of activity data and emission factors for small scattered sources that vary by location, combustion technologies used, and environmental conditions. Uncertainties in the sectoral and fuel allocations in CEDS<sub>GBD-MAPS</sub> emissions will also

generally follow the uncertainties in the CEDSv2019-12-23 system and will largely depend on the accuracy of the fuel allocations for combustion sources in the underlying IEA activity data. Gridded CEDS<sub>GBD-MAPS</sub> emissions also have uncertainties associated with the accuracy of the normalized spatial emission distributions from EDGAR v4.3.2, which are equally applied to all the four fuel categories and are held constant after 2012.

    Contemporary global emission estimates with detailed sector and fuel-specific information are vital for quantifying

the anthropogenic sources of air pollution and mitigating the resulting impacts on human health, the environment, and society. While bottom-up methods can provide sectoral-specific emission estimates, previous global inventories of multiple compounds and sources have lagged in time and do not provide fuel-specific emissions for multiple compounds at the global scale. To address this community need, the CEDS<sub>GBD-MAPS</sub> inventory utilizes the CEDS system (v2019-12-23) to provide emissions of seven key atmospheric pollutants with detailed sectoral and fuel-type information, extended to the year 2017. Due to the direct

and secondary contribution of these reactive gases and carbonaceous aerosol to ambient air pollution, contemporary gridded and country-level emissions with both sector and fuel-type information can provide new insights necessary to motivate and develop effective fuel abatement and air pollution mitigation strategies around the world. The CEDS<sub>GBD-MAPS</sub> source code is



publicly available (https://github.com/emcduffie/CEDS/tree/CEDS_GBD-MAPS and https://doi.org/10.5281/zenodo.3865670) and both country total and global gridded emissions from the 2020_v1 version of this dataset are publicly available

at Zenodo with the following doi: https://doi.org/10.5281/zenodo.3754964.

**Information about the Supplement**

The supplement for this article describes a list of known inventory issues at the time of submission, as well as a number of additional CEDS_GBD-MAPS details, tables and figures, and data sources, including the following: (Boden et al., 2016, 2017;BP, 2015;Doxsey-Whitfield et al., 2015;EC-JRC/PBL, 2012, 2016;EIA, 2019;IEA, 2015;Klein Goldewijk et al., 2011;Sharma et

al., 2019;Stohl et al., 2015;The World Bank, 2016;UN, 2014, 2015;Wiedinmyer et al., 2014;Commoner et al., 2000;Reyna-Bensusan et al., 2018;Nagpure et al., 2015;Meidiana and Gamse, 2010;US EPA, 2006).

**Author Contributions**

EEM prepared the manuscript with contributions from all co-authors. RVM, MB, and SSJ supervised the scientific content of this publication. EEM led the development of the CEDS_GBD-MAPS source code and CEDS_GBD-MAPS dataset, with significant

contributions from SSJ and PO, as well as supplemental data from KT, CV, and EAM.

**Competing Interests**

The authors declare that they have no conflict of interest.

**Acknowledgements**

This work was supported by the Health Effects Institute (HEI), Global Burden of Disease – Major Air Pollution Sources
project. We thank Christine Wiedinmyer and Qiang Zhang for their respective contributions to the DICE-Africa and updated China nation-level inventories, used here for calibration of CEDS_GBD-MAPS emissions. CEDS utilizes many sources of input data and we are grateful for these contributions from a large number of research teams.



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




**Tables**

**Table 1.** Comparison of three historical, gridded, source-specific emission inventories of atmospheric pollutants (NO$_x$, SO$_2$, CO, NMVOCs, NH$_3$, BC, OC).

| Inventory Name (version) | Temporal Coverage | Number of Reported Gridded Sectors | Detailed Fuels | Spatial Resolution | Reference |
|---|---|---|---|---|---|
| CEDS (v2016_07_26) | 1750 – 2014 | 9 | Total only | 0.5°×0.5° | (Hoesly et al., 2018) |
| EDGAR (v4.3.2) | 1970 – 2012 | 26 | Biofuel (Europe only)[b] | 0.1°×0.1° | (Crippa et al., 2018) |
| ECLIPSE (v5a) | 1990, 1995, 2000, 2005, 2010 (projections to 2050)[a] | 8 | Total only | 0.5°×0.5° | (Klimont et al., 2017;Amann et al., 2011) |

[a]Projections assume current air pollution legislation (CLE) in the GAINS model
[b]Described in Crippa et al. (2019)





**Table 2.** CEDS sector and fuel-type definitions. Aggregate sectors and fuel-types in the **CEDS_Hoesly** and *CEDS_GBD-MAPS* inventories, as well as the system's *intermediate gridding sectors*, and detailed working sectors/fuel-types (consistent between CEDS_Hoesly and CEDS_GBD-MAPS inventories). CEDS working sectors are methodologically treated as two different categories: combustion sectors (c) and 'process' sectors (p). As described in text, combustion sector emissions are calculated as a function of CEDS working fuels while process emissions assigned to the single 'process' fuel-type.

**CEDS Emission Sectors**

**Energy Production (ENE)**
 *Energy Production (ENE)*
  *Electricity and heat production*
   1A1a_Electricity-public (c)
   1A1a_Electricity-autoproducer (c)
   1A1a_Heat-production (c)
  *Fuel Production and Transformation*
   1A1bc_Other-transformation (p)
   1B1_Fugitive-solid-fuels (p)
  *Oil and Gas Fugitive/Flaring*
   1B2_Fugitive-petr-and-gas (p)
  *Fuel Production and Transformation*
   1B2d_Fugitive-other-energy (p)
  *Fossil Fuel Fires*
   7A_Fossil-fuel-fires (p)

**Industry (IND)**
 *Industry (IND)*
  *Industrial combustion*
   1A2a_Ind-Comb-Iron-steel (c)
   1A2b_Ind-Comb-Non-ferrous-metals (c)
   1A2c_Ind-Comb-Chemicals (c)
   1A2d_Ind-Comb-Pulp-paper (c)
   1A2e_Ind-Comb-Food-tobacco (c)
   1A2f_Ind-Comb-Non-metalic-minerals (c)
   1A2g_Ind-Comb-Construction (c)
   1A2g_Ind-Comb-transpequip (c)
   1A2g_Ind-Comb-machinery (c)
   1A2g_Ind-Comb-mining-quarrying (c)
   1A2g_Ind-Comb-wood-products (c)
   1A2g_Ind-Comb-textile-leather (c)
   1A2g_Ind-Comb-other (c)
   1A5_Other-unspecified (c)
  *Industrial process and product use*
   2A1_Cement-production (p)
   2A2_Lime-production (p)
   2A6_Other-minerals (p)
   2B_Chemical-industry (p)
   2C_Metal-production (p)
   2H_Pulp-and-paper-food-beverage-wood (p)
   2L_Other-process-emissions (p)
   6A_Other-in-total (p)

**Transportation (TRA)**
 *Road Transportation (ROAD)*
  *Road transportation*
   1A3b_Road (c)

**Residential, Commercial, Other (RCO)**
 *Residential (RCOR)*
  *Res., Comm., Other - Residential*
   1A4b_Residential (c)
 *Commercial (RCOC)*
  *Res., Comm., Other - Commercial*
   1A4a_Commercial-institutional (c)
 *Other (RCOO)*
  *Res., Comm., Other - Other*
   1A4c_Agriculture-forestry-fishing (c)

**Solvents (SLV)**
 *Solvents (SLV)*
  *Solvents production and application*
   2D_Degreasing-Cleaning (p)
   2D3_Other-product-use (p)
   2D_Paint-application (p)
   2D3_Chemical-products-manufacture-processing (p)

**Agriculture (AGR)**
 *Agriculture (AGR)*
  *Agriculture*
   3B_Manure-management (p)
   3D_Soil-emissions (p)
   3I_Agriculture-other (p)
   3D_Rice-Cultivation (p)
   3E_Enteric-fermentation (p)

**Waste (WST)**
 *Waste (WST)*
  *Waste*
   5A_Solid-waste-disposal (p)
   5E_Other-waste-handling (p)
   5C_Waste-incineration (p)
   5D_Wastewater-handling (p)

**Shipping (SHP)**
 *Shipping (SHP)*
  *International shipping*
   1A3di_International-shipping (c)
  *Tanker Loading*
   1A3di_Oil_Tanker_Loading (p)

**Transportation Cont. (TRA)**
 *Non-Road Transportation (NRTR)*
  *Non-road Transportation*
   1A3c_Rail (c)
   1A3dii_Domestic-navigation (c)
   1A3eii_Other-transp (c)

**CEDS Fuels**

**Total**
 *Coal*
  Brown coal
  Coal coke
  Hard coal
 *Biofuel*
  Biofuel

 *Liquid Fuel & Natural Gas*
  Heavy oil
  Diesel oil
  Light oil
  Natural Gas
 *Process*
  Process






**Table 3. Calibration Inventories**

| Inventory Name | Scaled Inventory Years | Scaled Species | Reference |
|---|---|---|---|
| EDGAR v4.3.2 | 1992 – 2012 | CO, NH$_3$, NMVOCs, NO$_x$ | (EC-JRC, 2018) |
| EMEP NFR14 | 1990 – 2017 | CO, NH$_3$, NMVOCs, NO$_x$, SO$_2$, BC | (EMEP, 2019) |
| UNFCCC | 1990 – 2017 | CO, NMVOCs, NO$_x$, SO$_2$ | (UNFCCC, 2019) |
| REAS 2.1[a] | 2000 – 2008 | CO, NH$_3$, NMVOCs, NO$_x$, SO$_2$, BC | (Kurokawa et al., 2013) |
| APEI (Canada) | 1990 – 2017 | CO, NH$_3$, NMVOCs, NO$_x$, SO$_2$ | (ECCC, 2019) |
| US EPA | 1970, 1975, 1980, 1985, 1990 – 2017 | CO, NH$_3$, NMVOCs, NO$_x$, SO$_2$ | (US EPA, 2019) |
| MEIC (China) | 2008, 2010 – 2017 | CO, NH$_3$, NMVOCs, NO$_x$, SO$_2$, BC, OC | (Zheng et al., 2018;Li et al., 2017c) |
| Argentina[a] | 1990 – 1999, 2011 – 2009, 2011 | CO, NMVOCs, NO$_x$, SO$_2$ | (Argentina UNFCCC Submission, 2016) |
| Japan[a] | 1960 – 2010 | CO, NH$_3$, NMVOCs, NO$_x$, SO$_2$, BC, OC | (preliminary update from Kurokawa et al., 2013)[a] |
| NEIR (South Korea)[a] | 1999 –2012 | CO, NMVOCs, NO$_x$, SO$_2$ | (South Korea National Institute of Environmental Research, 2016) |
| Taiwan[a] | 2003, 2006, 2010 | CO, NMVOCs, NO$_x$, SO$_2$ | (TEPA, 2016) |
| NPI (Australia) | 2000 – 2017 | CO, NMVOCs, NO$_x$, SO$_2$ | (ADE, 2019) |
| DICE-Africa[b] | 2006, 2013 | CO, NMVOCs, NO$_x$, SO$_2$, BC, OC | (Marais and Wiedinmyer, 2016) |
| SMoG-India[b] | 2015 | CO, NMVOCs, NO$_x$, SO$_2$, BC, OC | (Venkataraman et al., 2018) |

[a]Not updated from CEDS v2019-12-23, details in Hoesly et al. (2018).

[b]Emissions calibrated as a function of sector and fuel-type



**Figures**

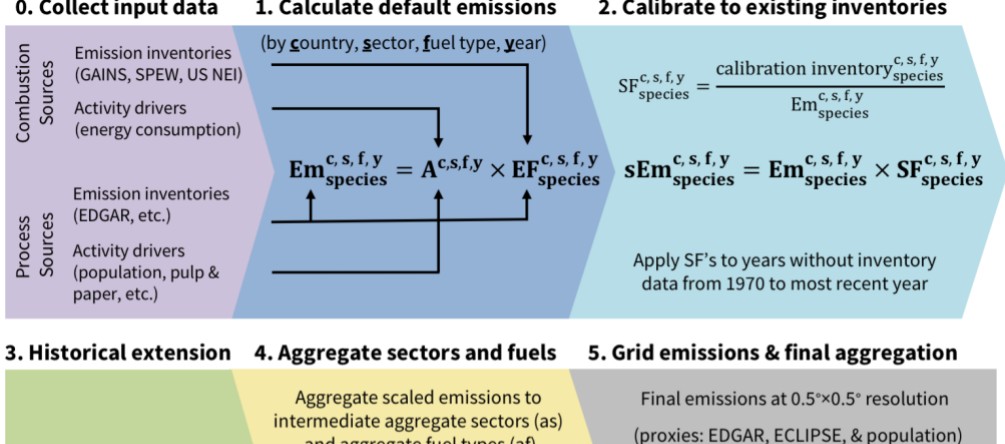

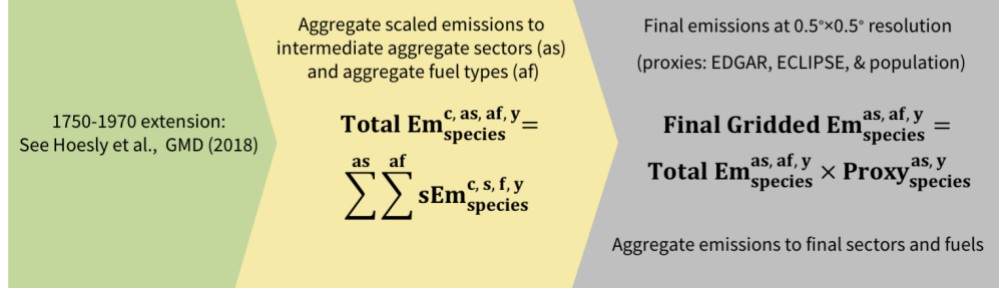

**Figure 1: Default CEDS System Summary, adapted from Fig. 1 in Hoesly et al. (2018). Key steps include: (0) collecting activity driver (A) and emission factor (EF) input data for non-combustion and combustion emission sources, (1) calculating default emissions (Em) as a function of chemical species, country, emission sector, fuel-type, and year, (2) calculating scaling factors (SFs) for overlapping years with existing inventories in order to calibrate default estimates (sEm) and extending SFs for non-overlapping years between 1970 – 2017 (for earlier emissions, see Hoesly et al. (2018)), (4) aggregating scaled emissions to intermediate sectors and fuel-types, and (5) using source and compound-specific spatial proxies to calculate final gridded emissions and aggregating them to the final sectors and fuels. A list of intermediate and final sectors and fuels are in Table 2.**

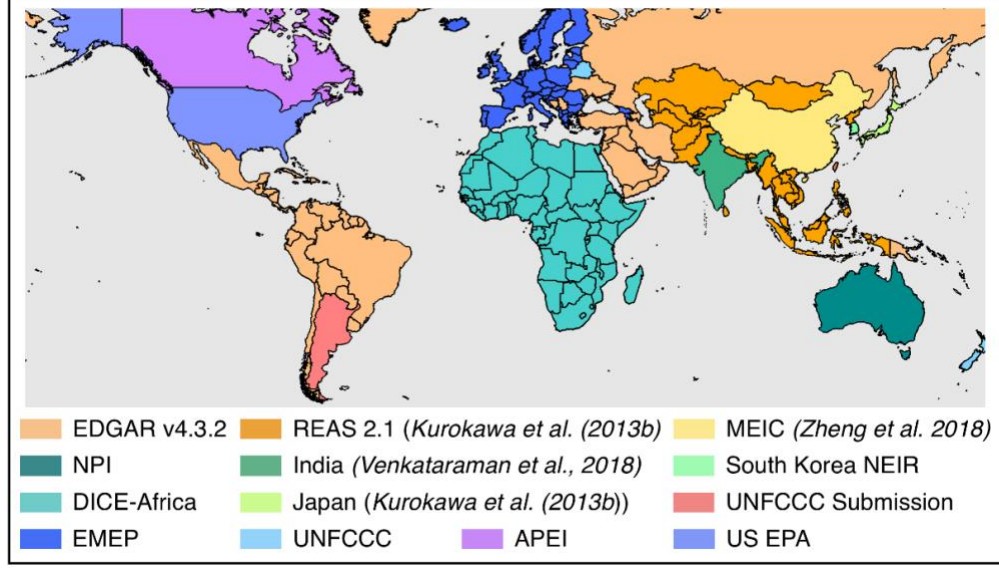

**Figure 2: Final calibration inventories used for CEDS_GBD-MAPS NO_x emissions, inventory details in Table 3.**

Earth System
Science
Data

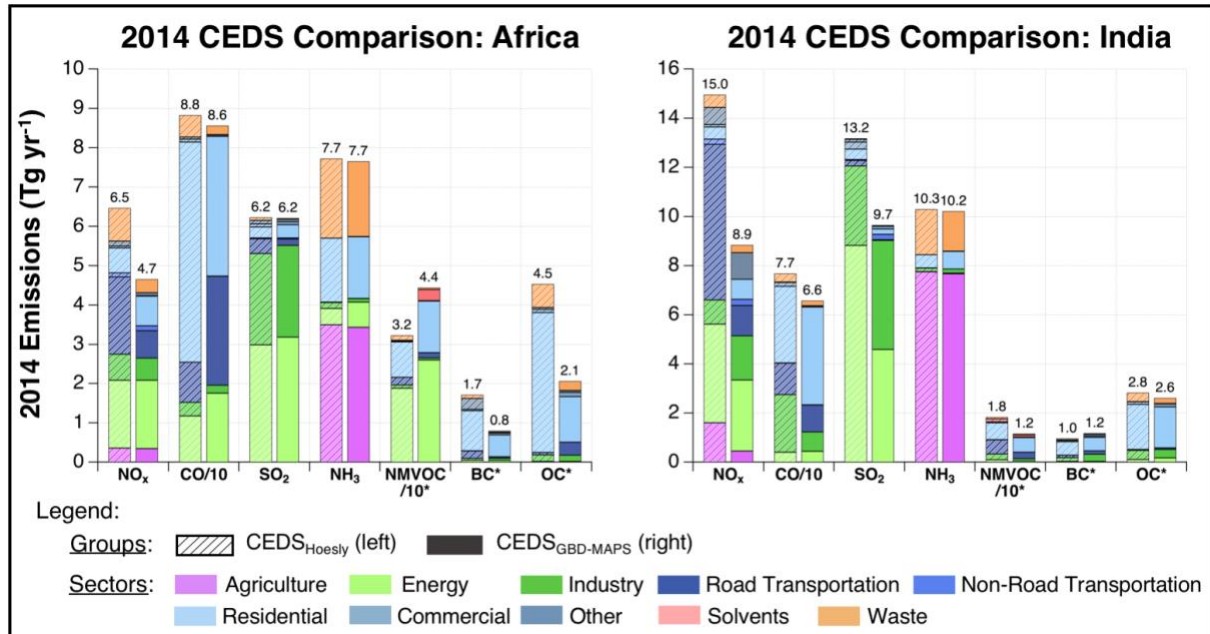

**Figure 3: Sectoral contributions to total annual emissions for 2014 of CEDS**Hoesly **(left) and CEDS**GBD-MAPS **(right) emissions after calibration to DICE-Africa and SMoG-India regional inventories. The total annual emissions are given by the values above each bar, bar colors represent absolute sectoral contributions to emissions of each chemical compound. CO and NMVOC emissions are divided by 10 for clarity. Stars indicate that NMVOCs, BC, and OC emissions are in units of TgC yr-1. NOx is in units of Tg NO2 yr-1**

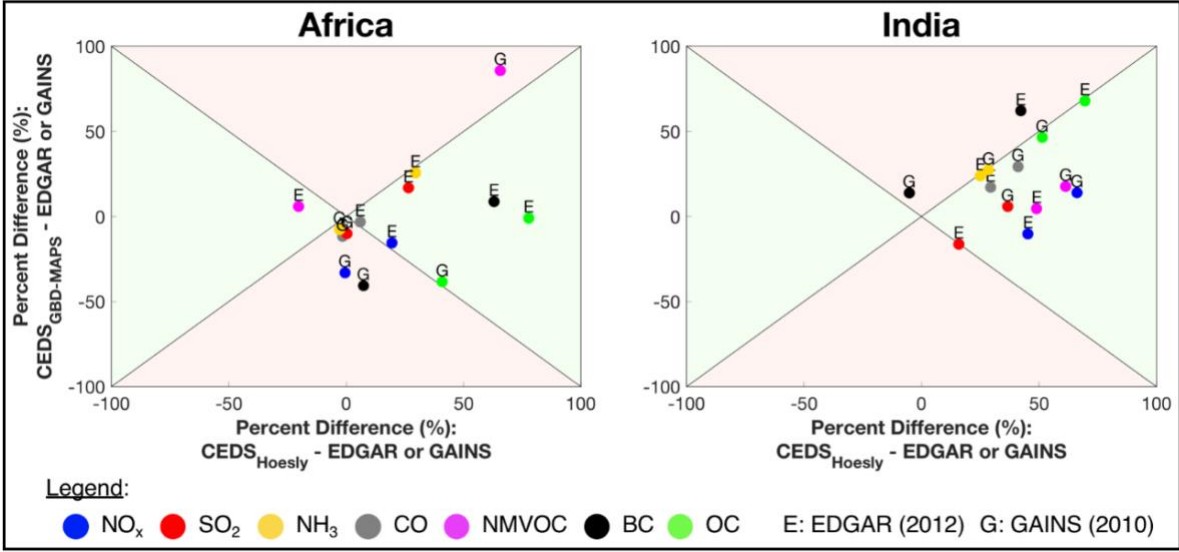

**Figure 4: Comparison of the percent difference between CEDS**GBD_MAPS**. X and Y-axes show the percent difference between the CEDS emission inventories (y-axis: CEDS**GBD-MAPS**, x-axis: CEDS**Hoesly**) for each compound and the GAINS (ECLIPSE v5a) or EDGARv4.3.2 inventories from Africa and India (i.e., 100*(CEDS – EDGAR)/(CEDS – EDGAR)/2)). Comparisons are conducted with the most recent available year, 2010 for the comparison with GAINS and 2012 for the comparison with EDGAR. Green regions indicate areas where the CEDS**GBD-MAPS **emissions have improved agreement with EDGAR and GAINS relative to the CEDS**Hoesly **inventory. Red areas indicate regions where CEDS**GBD-MAPS **emissions have worse agreement with EDGAR or GAINS relative to the**

**CEDSHoesly inventory. The color of each point represents the chemical compound and each point is labeled with an 'E' or 'G' indicating that the percent difference was calculated using EDGAR or GAINS, respectively.**

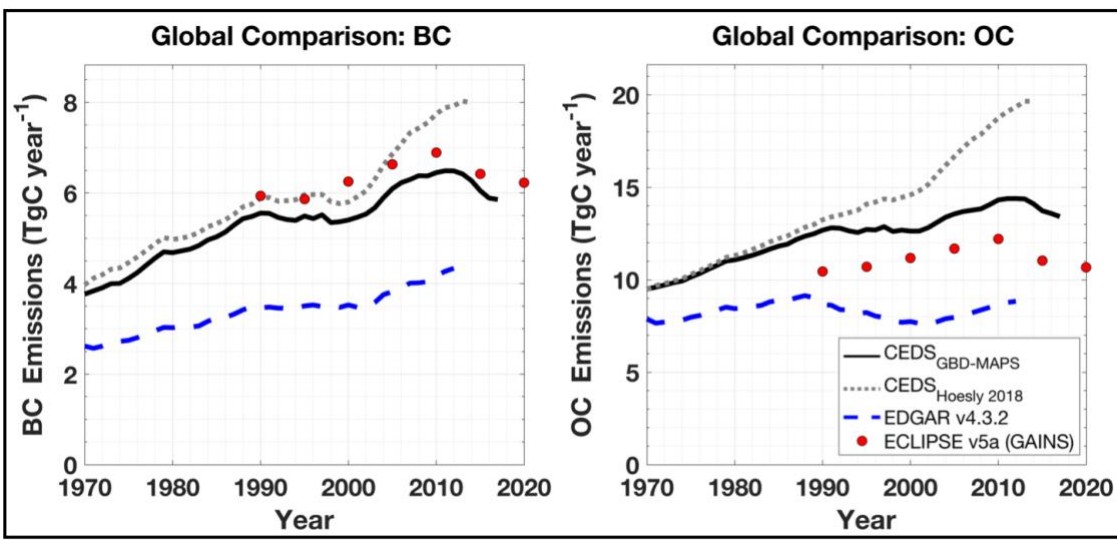

**Figure 5: Comparison of global inventories of BC and OC emissions. Total EDGARv4.3.2 and GAINS (ECLIPSE v5a) emission inventories shown without agricultural waste burning and aviation emissions. CEDSGBD-MAPS emissions of BC and OC are not calibrated to EDGAR or GAINS estimates.**





**Figure 6.** Time series of global annual emissions of NO$_x$ (as NO$_2$), CO, SO$_2$, NMVOCs, NH$_3$, BC, and OC for all sectors and fuel types. Solid black lines are the CEDS$_{GBD-MAPS}$ inventory, with fractional sector contributions indicated by colors. Dashed gray lines are the CEDS$_{Hoesly}$ inventory. Dashed blue lines are the EDGAR v4.3.2 global inventory. Red markers are ECLIPSE v5a baseline 'current legislation' (CLE) emissions (from the GAINS model) with data in 2015 and 2020 from GAINS CLE projections. All inventories include international shipping but exclude aircraft emissions. Pie chart inserts show fractional contributions of emission sectors to total 2017 emissions (outer) and fuel type contributions to each sector (inner). Emission totals for 2017 (units: Tg yr$^{-1}$, TgC yr$^{-1}$ for NMVOCs, OC, BC) are given inside each pie chart.

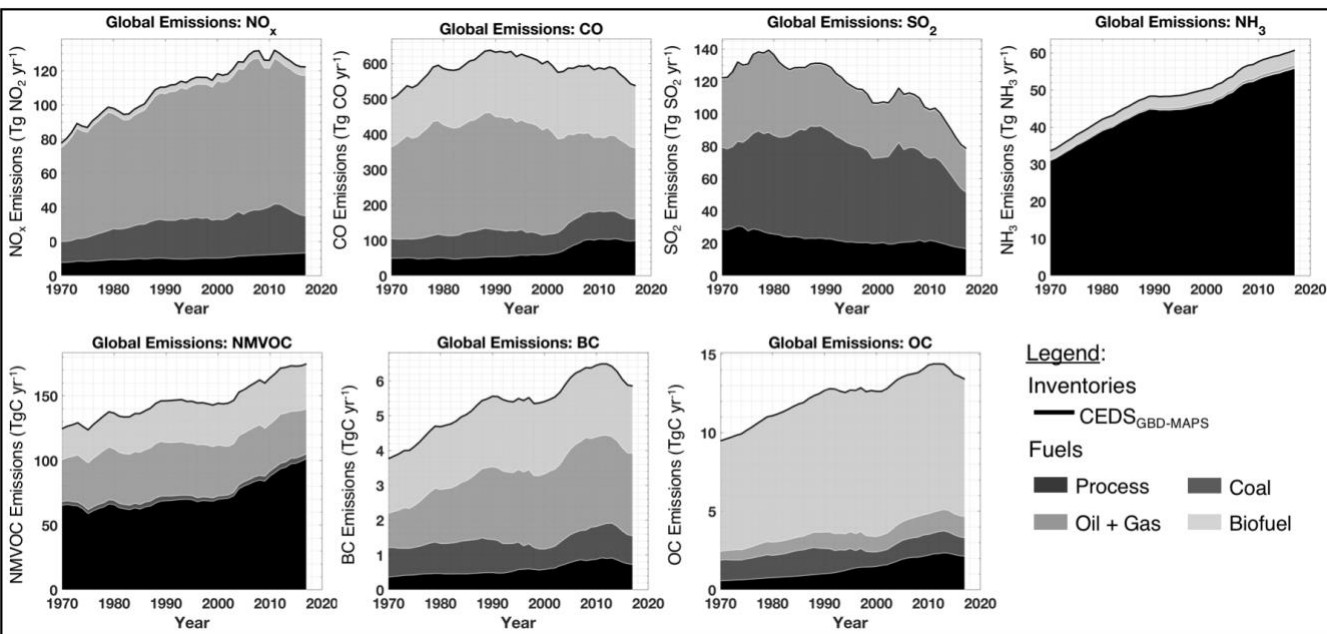

**Figure 7. Time series of global annual emissions of NOₓ, CO, SO₂, NH₃, NMVOCs, BC, and OC for all sectors, colored by fuel group.**


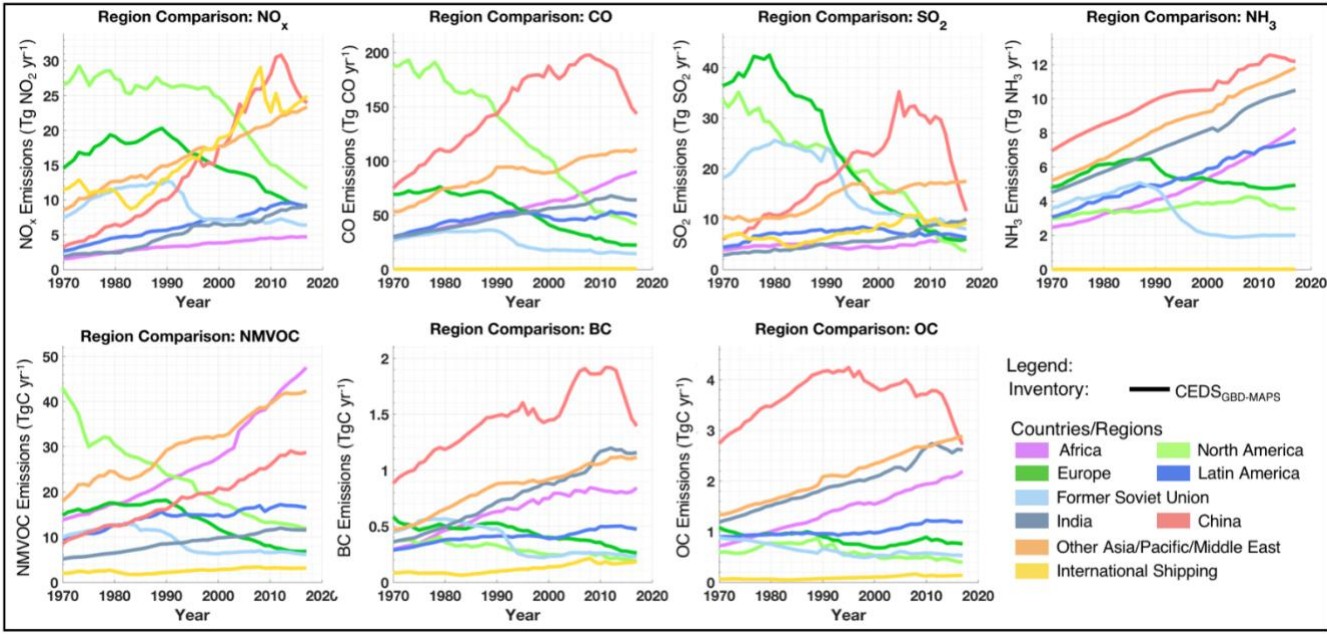

**Figure 8. Time series of global annual CEDS_GBD-MAPS emissions of NOₓ, CO, SO₂, NH₃, NMVOCs, BC, and OC for all sectors and fuel types (excluding aircraft emissions), split into nine countries/regions.**
