# Peer review of "A global anthropogenic emission inventory of atmospheric pollutants from sector- and fuel-specific sources (1970-2017): An application of the Community Emissions Data System (CEDS)"

_Earth System Science Data, 2020_

## Referee Comment (RC1) · Anonymous Referee #1 · 3 Aug 2020

McDuffie et al. Describe an update to the global emission inventory of atmospheric pollutants from the Community Emissions data System (CEDS). The updated dataset improves upon the earlier release of the CEDS inventory by adding additional emission sectors, separating emissions by fuel type, extending the timeseries to 2017, and updating the regional inventories used to "calibrate" the emissions corresponding regional sections of the global domain. The new CEDS inventory is currently the most up to date global emission inventory available to the community that is based on reported data. This fact, along with the new features provided with this data release, should

make the inventory very attractive to global atmospheric chemistry modellers.

The manuscript describing the dataset is generally organised and written well. A lot of detail goes into such emission inventories, and the authors have found a good balance between including information in the manuscript, the supplement, and as references to other work. As well as describing the methodology of constructing the inventory, the resulting dataset itself is also presented and described, as well as compared to other global inventories, including the previous version of the CEDS inventory. Uncertainties are also discussed.

My only major comment on the manuscript concerns the calibration procedure. It is obviously a strength of the CEDS approach that regional emissions are scaled using detailed regional inventories where they are available. In this way, more detailed local information can be incorporated than would typically be the case for completely globally consistent inventories such as EDGAR or GAINS. What is not clear to this reviewer is the necessity of also scaling the "default emission estimates" calculated in "Step 1" of the CEDS workflow to "existing, authoritative" global inventories (such as EDGAR and GAINS). Given the general uncertainties in emission inventories, would it not be valuable to have an additional semi-independent global inventory in addition to these two established inventories? Of course, a lot of the information used in constructing the CEDS inventory is shared with, or derived from the other global inventories, so a completely independent emissions inventory would be very difficult to compile. This reviewer would however like to see some more discussion of why it is necessary to calibrate the total CEDS emissions using other global inventories. Related to this point, it would also be very interesting to know the size of the "scaling factors" which are applied in "Step 2" to calibrate the CEDS default emissions with the other global inventories. These numbers to not appear to be presented in the manuscript or the supplement.

I also have one minor, and one extremely minor comment.

The minor comment relates to data availability. It is great that the CEDS inventory as

well as the code is made available to the public. But what about the input data which are necessary for the CEDS code to run? While the data sources do all appear to be well referenced, it would be nice to see some comment in Section 5 on how freely available the input data sets are. This would of course influence the feasibility of other groups being able to reproduce the CEDS emissions using the CEDS code.

The extremely minor comment relates to the presence or absence of seasonal cycles in the gridded emission data. While it seems clear that the gridded CEDS data do include a seasonal cycle, in two places (lines 250 and 790), these data are referred to as "annual" fluxes, implying strongly that they are annual averages. Perhaps this should be corrected to something like "seasonal cycles of annual . . . fluxes".

---

## Referee Comment (RC2) · Hugo Denier van der Gon (Referee) · 27 Aug 2020

Review of essd-2020-103 "A global anthropogenic emission inventory of atmospheric pollutants from sector- and fuel-specific sources (1970–2017): An application of the Community Emissions Data System (CEDS)" by McDuffie et al.

The paper describes an interesting new global emission Inventory (CEDSGBD-MAPS )for atmospheric pollutants (1970 – 2017) based on the so-called mosaic approach based on the Community Emissions Data System (CEDS). The paper is generally wellwritten and deserves to be published but I do have several concerns where I would ask for adjustment or further explanation. A problem with emission inventory papers is that one tries to describe a complete set for all pollutants, all source sectors, all countries and many years. It is impossible to write a paper on this that documents, explains & discusses all and is still readable. Choices have to be made. The intention of my review is not to be a dictate. Part of my comments will relate to choices made and I do not demand that all answers to my comments find their way into the paper. If the authors have good reasons for not adjusting something, they can explain themselves.

The mosaic approach is not new and was previously successfully applied for example in the framework of HTAP by Janssens-maenhout et al (2015). This is an often used mosaic inventory. The approach by Janssens-maenhout et al differs from the approach taken in this paper and I think this should be briefly discussed in the introduction. Also to make clear that mosaic inventories are becoming a more frequently followed approach.

A more fundamental problem is the term "calibration inventory" that is coined in the paper. Calibration is the comparison of measurement values delivered by a device under test (or a system) with those of a calibration standard of known accuracy. However, I think it is fair to say that the authors don't know the accuracy of their calibration inventories. They motivate that regional or national inventories may include more national/regional knowledge and are therefore more accurate. This was also the motivation for e.g. the earlier HTAP_v2.2 mosaic inventory. It may well be true (and this reviewer firmly believes in the usefulness of mosaic inventories) but a) we don't know for sure if the regional inventory is really better and b) we don't know how accurate exactly. Good enough for calibration? In my opinion the term calibration adds too much certainty to a more empirical and intuitive solution for an operational problem. It reads well but in reality it is more fitting or scaling than calibrating. In e.g. line 213 is also stated that scaling factors are calculated in the calibration procedure . Apparently scaling is seen as calibrating. Should the authors really think that calibration is still the best
terminology some additional clarification/disclaimer is needed to avoid "whitewashing" of something still uncertain (scaling) by calling it certain (calibrating). [see also the confusion created in line 360-365 between scaling and calibration and the remark that BC / OC are not scaled due to large uncertainties in EDGAR – but how well do you know that other inventories are much less uncertain? ]

An advantage of the mosaic approach is the inclusion of more locally / nationally representative inventories in the global emission map. A disadvantage is that the emissions from different regions become apples and oranges. Obviously still the same species but the underlying choices are no longer necessary the same. It would be interesting for some of the more uncertain species like CO, NMVOC or BC to show a plot comparing some implied emission factors for certain source sectors for e.g. Africa, India, China, Former Soviet Union. What is the range in these implied EFs and based on expert judgement of the authors do these ranges seem plausible? This may be used to flag some of the pollutant / source sector / region combinations that may deserve further investigation in the future? It could also be connected to the paragraph starting at line 300.

From the methods section it was not clear to me where the shipping emissions come from. Are these based on AIS data or taken from EDGAR? Or another approach? Like with the regional inventories there may be ways to "scale/calibrate" these in recent years by using AIS based inventories. Was this considered?

The region "Other Asia/Pacific/Middle East region". This I find non-informative and I invite the authors to think of a solution possibly by breaking it up. The mix of countries (see Table S8 - e.g. Australia, Mongolia, Yemen, Saudi Arabia, Korea, New Zealand, Pakistan, Indonesia etc. ) is such that any discussion of the trends for this group in the paper is pointless. Also graphs of such a group in my opinion do not add any information.

Compliments to the authors for all the line plots, they are generally really good to read

and intercompare and thereby also reveal some issues that appear unlikely to be correct. That does not mean they have to (or even can be) solved in the current paper. There are a few individual cases that draw attention and possibly merit more comments. I like to share them but it is also up to the authors to think about what they feel is justified. I am not advocating to make the paper very anecdotical by discussing every detail. Like the drop in OC emissions for Industry in Figure S6; the CO peak from road transport and SO2 peak for energy in fig S8 (the latter is discussed in the text) - My suspicion is that what such abrupt peaks or drops have in common is most likely a change in legislation or methodology that "on paper" has almost immediate effect but in reality is smeared out over a longer time. For example the car fleet cannot be changed in 1-2 years, cleaner fuels (like low sulphur) generally take years to be completely adopted. NMVOCs from the Energy sector appear a special case (Fig S5) with a very large contribution but little explanation is given other than that these are process emissions. NMVOCs in general draw some attention – e.g. in line 630 there is a discrepancy of possibly missing NMVOC emissions as CEDS has no agricultural NMVOC emission? And, for example Fig3 India NOx emissions – almost a factor 2 difference between 2 CEDS versions. It is commented on in the text but would it also imply it is better not to use the previous CEDS version because of these large deviations? The difference is too large for both to be equally plausible. This also applies to the discussion in line 568 and onward. As both inventories come from the CEDS team it seems logical to express some advice on to what extend you believe the new inventory replaces the old one. (Like the EDGAR team would advise to use v5 and v3 or v4. )

Line 648 – "decreasing uncertainties": Here I do not by definition agree. If for example the (more uncertain) emissions from Africa and India become dominant and e.g. the more certain emissions from the US & EU go down, than the overall uncertainty might also increase in future years.

A good assessment of uncertainty from a mosaic inventory is very challenging and

simply stating that the uncertainty is similar to the other inventories (e.g. line 655) is an unsatisfactory answer. Moreover, there may also be considerable uncertainty in the spatial distribution. The authors, however, announce that in the near future a more robust uncertainty analysis is planned. And a much longer paper would not be helpful for the community. So separating this is an acceptable solution.

Additional suggestions for final discussion: Recently Huneeus et al. (2020) published an evaluation of emission inventories for South America which included EDGSAR, ECLIPSE and CEDS. It would be interesting to comment on how the new inventory presented here would have an impact on SA estimates and compares to the CEDS version used in that paper? Elguindi et al. (2020) recently published a paper on inter-comparison of bottom-up inventories and top-down emissions. This may well be the way forward to build more confidence in mosaic inventories and justify certain choices.

Small editorial remarks

Line 42 – from "waste" combustion (otherwise strange to have carb aerosol from waste.)

Line 78 – as inputs to solve for? Not clear to me, maybe reformulate slightly?

Line 108 – "emission" reduction of coal-fired etc.

Line 181 – explain the term "working sector"

Line 410 – you mean Section S4.

Line 481 Global emissions of NOx from waste "combustion".

Line 680 I don't see how satellites will aid in fuel-type recognition.

Line 684 "emissions" – should be "uncertainties"?

Line 786 – but not for the latest years? And these will not be scaled ("calibrated") so not consistent?

[Figure]

Line 791 – it seems the reference of (McDuffie et al., 2020c) here and in the ref list is redundant because this sis the dataset connected to the present paper? So won't the reference to that data not be simply this paper instead of (McDuffie et al., 2020c)

Line 836 in agricultural "NH3" emissions

Line 867 – what is fuel abatement?

There is an error in Table S8 – Other Asia includes Montenegro and I assume Chinese Taipei is Taiwan?

There is an error in Table S9 – in the column for EDGAR "solvent use" and "waste" are swapped.

References

Janssens-Maenhout, G., Crippa, M., Guizzardi, D., Dentener, F., Muntean, M., Pouliot, G., Keating, T., Zhang, Q., Kurokawa, J., Wankmüller, R., Denier van der Gon, H., Kuenen, J. J. P., Klimont, Z., Frost, G., Darras, S., Koffi, B., and Li, M.: HTAP_v2.2: a mosaic of regional and global emission grid maps for 2008 and 2010 to study hemispheric transport of air pollution, Atmos. Chem. Phys., 15, 11411–11432, https://doi.org/10.5194/acp-15-11411-2015, 2015.

Huneeus, N., Denier van der Gon, H., Castesana, P., Menares, C., Granier, C., Granier, L., ... & Gomez, D., Evaluation of anthropogenic air pollutant emission inventories for South America at national and city scale, Atmospheric Environment 235 (2020) https://doi.org/10.1016/j.atmosenv.2020.117606

Elguindi, N., Granier, C., Stavrakou, T., Darras, S., Bauwens, M., Cao, H., et al. ( 2020). Intercomparison of magnitudes and trends in anthropogenic surface emissions from bottom‐up inventories, top‐down estimates and emission scenarios. Earth's Future, 8, e2020EF001520. https://doi.org/10.1029/2020EF001520

---

## Author Response (AR1)

**Review Responses for: A global anthropogenic emission inventory of atmospheric pollutants from sector- and fuel-specific sources (1970–2017): An application of the Community Emissions Data System (CEDS)"** *by* **Erin E. McDuffie et al.**

We thank both Reviewers for their comments, which have helped improve the quality and clarity of our manuscript describing the CEDS$_{GBD-MAPS}$ dataset. We have responded to each comment below. The original comments are in black, our responses are in blue and the changes to the manuscript text are in *blue italics*. Overall, the dataset remains unchanged, but we have added two additional supplemental figures and 1 supplemental table to address reviewer-specific concerns. All other manuscript changes are related to clarifying the CEDS methodology or descriptions of the final dataset. Changes were made to maintain a similar manuscript length, while providing improved clarity, context, and interpretation of major features. Line numbers in our responses below correspond to the re-submitted (non-tracked) version of the manuscript.

**Anonymous Referee #1**

McDuffie et al. Describe an update to the global emission inventory of atmospheric pollutants from the Community Emissions data System (CEDS). The updated dataset improves upon the earlier release of the CEDS inventory by adding additional emission sectors, separating emissions by fuel type, extending the timeseries to 2017, and updating the regional inventories used to "calibrate" the emissions corresponding regional sections of the global domain. The new CEDS inventory is currently the most up to date global emission inventory available to the community that is based on reported data. This fact, along with the new features provided with this data release, should make the inventory very attractive to global atmospheric chemistry modellers.

The manuscript describing the dataset is generally organised and written well. A lot of detail goes into such emission inventories, and the authors have found a good balance between including information in the manuscript, the supplement, and as references to other work. As well as describing the methodology of constructing the inventory, the resulting dataset itself is also presented and described, as well as compared to other global inventories, including the previous version of the CEDS inventory. Uncertainties are also discussed.

My only major comment on the manuscript concerns the calibration procedure. It is obviously a strength of the CEDS approach that regional emissions are scaled using detailed regional inventories where they are available. In this way, more detailed local information can be incorporated than would typically be the case for completely globally consistent inventories such as EDGAR or GAINS.

We thank this reviewer for their thoughtful comments and concerns. First, we note that per the suggestion of Reviewer #2, we have changed the terminology throughout the manuscript to now describe the original 'calibration' procedure as the 'scaling' procedure. We agree with Reviewer #2 that the term 'calibration' implies too great a level of certainty and accuracy in the regional and global inventories.

What is not clear to this reviewer is the necessity of also scaling the "default emission estimates" calculated in "Step 1" of the CEDS workflow to "existing, authoritative" global inventories (such as EDGAR and GAINS). Given the general uncertainties in emission inventories, would it not be valuable to have an additional semi-independent global inventory in addition to these two established inventories? Of course, a lot of the information used in constructing the CEDS inventory is shared with, or derived from the other global inventories, so a completely independent emissions inventory would be very difficult to compile.

The Reviewer is correct that regardless of default emission scaling, information from both EDGAR and GAINS global inventories are used to develop the CEDS$_{GBD-MAPS}$ (and core CEDS) inventory. The original aim of CEDS was not to generate a completely new independent inventory, but to use a consistent and reproducible methodology, while also leveraging information from regional and country-specific inventories to generate historical emission time series with consistent sectoral and fuel-type definitions across all years and all world countries. EDGAR already exists as a consistent, global inventory developed using consistent assumptions, so we do not need to re-invent that work.

This reviewer would however like to see some more discussion of why it is necessary to calibrate the total CEDS emissions using other global inventories.

Section 2.1 describes how the default CEDS combustions source emissions are estimated using a combination of energy consumption data from the International Energy Agency and Emission Factors from other inventories such as GAINS and the US NEI. The GAINS EFs are only available for more aggregate regions, whereas the EDGAR inventory has country-specific emission estimates for all countries. Scaling to the EDGAR inventory is therefore meant to better account for country-specific information in the CEDS combustion emission estimates in locations where country-specific information was not available for the default estimates. We do note however that for countries that are later scaled to other regional inventories (shown in Fig. 2), the initial scaling to the EDGAR inventory should have a limited effect on the final emission values. We have added the following text to the main manuscript to clarify this point.

Line 310– *For example, global CEDS$_{GBD-MAPS}$ combustion source emissions of NO$_x$, total NMVOCs, CO, and NH$_3$ are first scaled to EDGAR v4.3.2 country-level emissions as a means to incorporate additional country-specific information relative to default estimates derived using more regionally-aggregate EFs from GAINS.*

Related to this point, it would also be very interesting to know the size of the "scaling factors" which are applied in "Step 2" to calibrate the CEDS default emissions with the other global inventories. These numbers to not appear to be presented in the manuscript or the supplement.

Scaling factor limits are set following the original CEDS protocol so that these factors do not exceed the range of 1/100 to 100. For a select number of sectors and countries, these limits are extended to 1/1000 to 1000 to ensure better agreement between the final CESD$_{GBD-MAPS}$ emission totals and the regional inventories, as was described in Section S2.3.2. To clarify, we have added the following sentence to the Supplement.

Supplement - Line 170 - *Following original CEDS protocols, scaling factors are limited to values between 0.01 and 100, with select inventories and sectors expanded to a range of 0.001 and 1000, as described in Supplemental Section S2.3.2. As discussed in Hoesly et al. (2018), particularly small or large scaling factors may result for multiple reasons, including default CEDS estimates that are drastically different than regional emissions or imprecise mapping between CEDS and regional emission sectors.*

At the Reviewer's request, we have additionally provided a supplemental table of example scaling factors for a sub-set of African countries and years (Table S4). As stated in the main text, small and large scaling factors may result from largely different emission estimates in the default inventory relative to the regional inventories and/or imperfect mapping between the CEDS and regional inventory sectors.

Supplement - Line 169: *Example scaling factors for select years and countries in Africa, as a function of scaling sector are provided in Table S4. Data are included for illustrative purposes only.*

*Table S4. Example BC scaling factors for select DICE-Africa countries and years.*

| Country (ISO) | Scaling Sector | Scaling Fuel | 2006 | 2007 | 2008 | 2009 | 2010 | 2011 | 2012 | 2013 | 2014 | 2015 |
|---|---|---|---|---|---|---|---|---|---|---|---|---|
| ago | residential | biomass | 0.332 | 0.338 | 0.344 | 0.350 | 0.355 | 0.361 | 0.367 | 0.373 | 0.373 | 0.373 |
| ago | residential | light oil | 0.340 | 0.311 | 0.282 | 0.252 | 0.223 | 0.194 | 0.165 | 0.136 | 0.136 | 0.136 |
| ago | road_ transport | gas_ diesel | 0.307 | 0.293 | 0.278 | 0.264 | 0.250 | 0.235 | 0.221 | 0.207 | 0.207 | 0.207 |
| nam | residential | biomass | 0.297 | 0.320 | 0.342 | 0.364 | 0.386 | 0.409 | 0.431 | 0.453 | 0.453 | 0.453 |
| nam | residential | light oil | 44.71 | 44.72 | 44.72 | 44.72 | 44.73 | 44.73 | 44.73 | 44.74 | 44.74 | 44.74 |
| nam | road_ transport | gas_ diesel | 0.274 | 0.260 | 0.247 | 0.234 | 0.220 | 0.207 | 0.194 | 0.180 | 0.180 | 0.180 |

I also have one minor, and one extremely minor comment.
The minor comment relates to data availability. It is great that the CEDS inventory as well as the code is made available to the public. But what about the input data which are necessary for the CEDS code to run? While the data sources do all appear to be well referenced, it would be nice to see some comment in Section 5 on how freely available the input data sets are. This would of course influence the feasibility of other groups being able to reproduce the CEDS emissions using the CEDS code.

We have added the following sentence to section 5.
Line 858: *To run the CEDS system, users are required to first purchase the proprietary energy consumption data from the IEA (World Energy Statistics; https://www.iea.org/subscribe-to-data-services/world-energy-balances-and-statistics). The IEA is updated annually and provides the most comprehensive global energy statistics available to-date. All additional input data are available on the CEDS GitHub repository.*

The extremely minor comment relates to the presence or absence of seasonal cycles in the gridded emission data. While it seems clear that the gridded CEDS data do include a seasonal cycle, in two places (lines 250 and 790), these data are referred to as "annual" fluxes, implying strongly that they are annual averages. Perhaps this should be corrected to something like "seasonal cycles of annual . . . fluxes".

We have replaced the incorrect use of 'annual fluxes' on original lines 250 and 790 with 'monthly fluxes'. The seasonal profiles are primarily from the ECLIPSE project as mentioned in Section 2.1.

Line 257 - *Final products from the CEDS$_{GBD-MAPS}$ system include total annual emissions from 1970 - 2017 for each country, as well as monthly global gridded (0.5°×0.5°) emission fluxes…*

Line 862 - *Final products from the CEDS$_{GBD-MAPS}$ system include total annual emissions for each country as well as monthly global gridded (0.5°×0.5°) emission fluxes for the years 1970 – 2017.*

**Reviewer #2 – Hugo Denier van der Gon**

The paper describes an interesting new global emission Inventory (CEDSGBD-MAPS) for atmospheric pollutants (1970 – 2017) based on the so-called mosaic approach based on the Community Emissions Data System (CEDS). The paper is generally well-written and deserves to be published but I do have several concerns where I would ask for adjustment or further explanation. A problem with emission inventory papers is that one tries to describe a complete set for all pollutants, all source sectors, all countries and many years. It is impossible to write a paper on this that documents, explains & discusses all and is still readable. Choices have to be made. The intention of my review is not to be a dictate. Part of my comments will relate to choices made and I do not demand that all answers to my comments find their way into the paper. If the authors have good reasons for not adjusting something, they can explain themselves.

We thank the Reviewer for their thoughtful and detailed comments. We recognize the Reviewer's expertise and experience in this field and have addressed their comments accordingly. The changes and additions described below have greatly improved the quality and clarity of this manuscript.

The mosaic approach is not new and was previously successfully applied for example in the framework of HTAP by Janssens-maenhout et al (2015). This is an often used mosaic inventory. The approach by Janssens-maenhout et al differs from the approach taken in this paper and I think this should be briefly discussed in the introduction. Also to make clear that mosaic inventories are becoming a more frequently followed approach.

We have made adjustments to sentences in the Introduction and Methods sections to provide further context on the use of the 'mosaic' development strategy.

Line 95 - *In contrast to EDGAR and GAINS, the CEDS system implements an increasingly utilized mosaic approach, which, in this case, incorporates activity and emission input data from other sources such as EDGAR, GAINS, and regional/national-level inventories to produce global emissions that are both historically consistent and reflective of contemporary country-level estimates (Hoesly et al., 2018).*

Line 202 - *As described in Hoesly et al. (2018), CEDS uses a "mosaic" scaling approach to retain detailed fuel- and sector-specific information across different inventories, while maintaining consistent methodology over space and time. The development and use of mosaic inventories has been recently increasing as they provide a means to utilize detailed local emissions, while harmonizing this information across large regional or global scales (Li et al., 2017;Janssens-Maenhout et al., 2015). The CEDS approach, however, differs from previous mosaic inventories, such as that developed for the HTAP project (Janssens-Maenhout et al., 2015), as local and regional inventories in CEDS$_{GBD-MAPS}$ are used to scale sectoral emissions at the national-level, rather than merging together spatially distributed gridded estimates.*

A more fundamental problem is the term "calibration inventory" that is coined in the paper. Calibration is the comparison of measurement values delivered by a device under test (or a system) with those of a calibration standard of known accuracy. However, I think it is fair to say that the authors don't know the accuracy of their calibration inventories. They motivate that regional or national inventories may include more national/regional knowledge and are therefore more accurate. This was also the motivation for e.g. the earlier HTAP_v2.2 mosaic inventory. It may well be true (and this reviewer firmly believes in the usefulness of mosaic inventories) but a) we don't know for sure if the regional inventory is really better and b) we don't know how accurate exactly. Good enough for calibration? In my opinion the term calibration adds too much certainty to a more empirical and intuitive solution for an operational problem. It reads well but in reality it is more fitting or scaling than calibrating. In e.g. line 213 is also stated that scaling factors are calculated in the calibration procedure . Apparently scaling is seen as calibrating. Should the authors really think that calibration is still the best terminology some additional clarification/disclaimer is needed to avoid "whitewashing" of something still uncertain (scaling) by calling it certain (calibrating). [see also the confusion created in line 360-365 between scaling and calibration and the remark that BC / OC are not scaled due to large uncertainties in EDGAR – but how well do you know that other inventories are much less uncertain? ]

We appreciate this comment and in retrospect, agree that the term 'calibration' provides too much weight to the accuracy of the regional inventories. While the accuracies of some regional inventories have been quantified (e.g., Venkataraman et al., 2018), we agree that the term 'calibration' is not appropriate in this context. We have changed all instances and variations of the term 'calibration' to 'scaling' throughout the main text, supplement, and figure and table captions. Similarly, the 'calibration inventories' are changed to 'scaling inventories'. These changes are now also in better alignment with the original CEDS description in Hoesly et al. (2018).

An advantage of the mosaic approach is the inclusion of more locally / nationally representative inventories in the global emission map. A disadvantage is that the emissions from different regions become apples and oranges. Obviously still the same species but the underlying choices are no longer necessary the same. It would be interesting for some of the more uncertain species like CO, NMVOC or BC to show a plot comparing some implied emission factors for certain source sectors for e.g. Africa, India, China, Former Soviet Union. What is the range in these implied EFs and based on expert judgement of the authors do these ranges seem plausible? This may be used to flag some of the pollutant / source sector / region combinations that may deserve further investigation in the future? It could also be connected to the paragraph starting at line 300.

We have added Figure S2 to the Supplement to provide an illustration of the time series of implied emission factors for each compound for the top 15 emitting countries. The panels in this figure show the implied emission factors (units of g g$^{-1}$) for the fuel and sector combinations that

dominantly contribute to emissions of each compound. For example, $NO_x$ emissions are predominately from the combustion of oil and natural gas in the road-transport sector. We have used this figure to support several sections of our manuscript including the Methods and uncertainties discussion and have added the following sections to the main text.

Line 229 - *Figure S2 provides a time series of implied emission factors after the scaling procedure for select sector- and fuel- combinations that dominant emissions of each compound in the top 15 emitting countries.*

Line 336 - *Figure S2 shows that after scaling, the implied emission factors of CO from oil and gas combustion in the on-road transport sector for four African countries range from 0.19-0.28 g $g^{-1}$, slightly smaller than the range of 0.029 - 0.380 g $g^{-1}$ used in the DICE-Africa inventory.*

Line 345 - *After scaling, the implied EFs for residential biofuel emissions of OC are ~0.001-0.002 g $g^{-1}$ in three African countries (Figure S2), within the range of EFs of 0.0007 – 0.003 g $g^{-1}$ implemented in the DICE-Africa inventory.*

Line 354 - *Figure S2 shows that the implied emission factor for $NO_x$ emissions from oil & gas combustion in the on-road transport sector in India is ~0.015 g $g^{-1}$ in 2015, which falls within the range of values of 0.0026 – 0.046 g $g^{-1}$ used for various vehicles and fuel type in Venkataraman et al. (2018).*

Line 359 - *For $SO_2$, Figure S2 shows that the implied EF for coal combustion in the energy sector is ~0.004 g $g^{-1}$, slightly lower than the range of 0.0049 – 0.0073 g $g^{-1}$ used for the SMoG-India inventory.*

Line 761 - *In addition to uncertainties in the scaling inventory emissions, uncertainties are also introduced by the CEDS$_{GBD-MAPS}$ scaling procedure. Uncertainties arise when mapping sectoral and fuel (when available) specific emissions between inventories (as discussed previously), as well as in the application of the calculated scaling factors outside the range of available scaling inventory years. For example, the implied CO EFs in Figure S2 highlight one case in China where the EFs for oil and gas combustion in the on-road transport sector peak in 1999 at a value over three times larger than EFs in all other top emitting countries. For China specifically, the calculated scaling factors for the year 2010 (earliest scaling inventory year) are applied to emissions from all years prior, which was calculated as a value of ~1.58 for the on-road transport sector. The implied EF of ~1.8 g $g^{-1}$ for this sector in 2003 (Figure S2) suggests that the SF from 2010 may not be representative of emissions during this earlier time period. We do note, however, that the 1999 peak in total CO emissions in China (Figure S9) is driven by the IEA energy data and is consistent with the CEDS$_{Hoesly}$ inventory (Hoesly et al., 2018). In contrast, EFs from this sector in China after the year 2010 agree with the magnitude and trends found in other countries, further indicating that the scaling factors*

*are most appropriate for years with overlapping inventory data. Other similar examples include coal energy emissions of $SO_2$ in Thailand (Figure S2). In this case, the REAS scaling inventory spans the years 2000 – 2008. The default EFs for the energy sector, however, independently decrease between 1997 and 2001. As a result, when the implied EF of 3.3 for the year 200 is applied to all historical energy emissions, the implied EFs prior to 1997 become an order of magnitude larger than those in nearly all other top emitting countries (Figure S2). Overall, the applicability of the scaling factors to emissions in years outside the available scaling inventory years remain uncertain due to real historical changes in activity, fuel-use, and emissions mitigation strategies. These uncertainties, however, vary by compound and sector as, for example, there are no similar peaks in on-road emissions for compounds other than CO in China.*

Supplemental Line 202 - *To illustrate the outcome of the scaling procedure, implied emission factors for the top 15 emitting countries are additionally shown in Figure S2 for the select fuel-types and sectors that dominantly contribute to global emission of each compound. Various anomalies in the implied EFs can arise from multiple sources of uncertainty, including the underlying activity data or application of scaling factors outside the available scaling inventory years, as is the case with the on-road CO emission factor for China in 1999. These uncertainties are discussed further in Section 4.2 in the main text.*

[Figure]

*Figure S2. Time-series of implied (post-scaling) emission factors for select fuel and sector combinations that dominantly contribute to global emissions of each compounds. NOx, CO, and BC: oil & natural gas combustion in the on-road transport sector, SO₂: coal combustion in the energy sector, NH₃: agricultural emissions, NMVOCs: process-level energy sources, and OC: residential biofuel combustion. Time series are shown for the top 15 emitting countries, listed by their ISO codes to the right of each panel. Time series are colored by the region of each country.*

From the methods section it was not clear to me where the shipping emissions come from. Are these based on AIS data or taken from EDGAR? Or another approach? Like with the regional inventories there may be ways to "scale/calibrate" these in recent years by using AIS based inventories. Was this considered?

We had not originally included an explicit description for international shipping emissions as these were not changed relative to the CEDS$_{Hoesly}$ inventory. To clarify the source of these emissions, we have added the following description to the Methods section.

Line 192 - *For International Shipping, IEA activity data is supplemented with consumption data and EFs from the International Maritime Organization (IMO), as described in Hoesly et al. (2018) and its supplement.*

The region "Other Asia/Pacific/Middle East region". This I find non-informative and I invite the authors to think of a solution possibly by breaking it up. The mix of countries (see Table S8 - e.g. Australia, Mongolia, Yemen, Saudi Arabia, Korea, New Zealand, Pakistan, Indonesia etc. ) is such that any discussion of the trends for this group in the paper is pointless. Also graphs of such a group in my opinion do not add any information.

In an attempt to simplify the line plots and maintain consistency with Hoesly et al. (2018), we chose to aggregate many of the countries that were not central to the discussion in our manuscript (i.e., those countries that were not largely updated relative to the CEDS$_{Hoesly}$ inventory). To provide a more meaningful discussion, we have now broken the 'Other Asia / Pacific/ Middle East region into the Australasia, Middle East, and Other Asia / Pacific regions. The new definitions are in Table S9. We have also updated Fig 8 accordingly, as well as SI Fig's S10 and S17-S20. The following sections of text have been updated as well.

Line 441 - *Time series of regional contributions to global emissions in Fig. 8 additionally show that 50% of global 2017 NO$_x$ emissions are from the combined Other Asia/Pacific region (Table S9) (13 Tg), China (24 Tg), International Shipping (25 Tg).*

Line 450 - *Fig. 8 shows that in 2017, China is the dominant source of global CO (144 Tg, 27% of global total), SO$_2$ (12 Tg, 15% of global total), NH$_3$ (12 Tg, 20% of global total), OC (2.7 TgC, 20% of global total), and BC (1.4 TgC, 24% of global total). In contrast, Africa is the dominant source of global NMVOCs in 2017 (48 TgC, 27% of global total) and International Shipping is the dominant source of global NO$_x$ emissions (25 Tg, 20% of global total).*

Line 466 - *Despite, however, continued reductions in these regions, global emissions of CO slightly increase between 2002 and 2012 due to simultaneous increases among the energy, industry, and residential sectors in China, India, Africa, and the Other Asia/Pacific region (Fig. S9-S12).*

Line 480 - *While China is the largest global contributor to $SO_2$ emissions between 1994 and 2017, these large regional reductions, coupled with increasing $SO_2$ emissions in the Other Asia/Pacific region, African countries, and India (Fig. 8), indicate that future global $SO_2$ emissions will increasingly reflect activities in these other rapidly growing regions.*

Line 488 - *While more stringent vehicle emission standards result in more than a factor of 2 decrease in on-road transportation $NO_x$ emissions in North America and Europe between 1992 and 2017 (Fig. S7-S8), on-road transport emissions in China, India, and the Other Asia/Pacific region simultaneously experience between a factor of 1.3 to 2.8 increase (Fig. S9-S11).*

Line 493 - *Global $NO_x$ emissions from the energy and industry sectors increase by up to a factor of 6 between 1970 and 2011 due to regional increases in China, India, the Other Asia/Pacific region, and African countries, with reductions between 2011 and 2017, again largely from reductions in China from stricter emissions control policies for coal fired power plants and coal use in industrial processes (Zheng et al., 2018;Liu et al., 2015).*

Line 503 - *Though emissions of BC and OC have a higher level of uncertainty relative to other compounds (Sect. 4), emissions from African countries and the Other Asia/Pacific region experience growth in BC and OC emissions from these sectors.*

Line 509 – *Similar to trends in $SO_2$ emissions, increasing trends in total OC and BC emissions from Africa, India, Latin America, the Middle East, and the Other Asia/Pacific region, coupled with large decreases in emissions from China, North America, and Europe (Fig. 8) indicate that global emissions will increasingly reflect activities in these rapidly growing regions.*

Line 518- *Similarly, global $NH_3$ emissions from the waste sector increase by 77% between 1970 and 2017, driven by increases in Latin America, the Other Asia/Pacific region, Africa, and India (Fig. S6-S12).*

Line 525 - *Emissions from China are the second largest global NMVOC source between 1996 and 2017 (Fig. 8), while the Other Asia/Pacific region is the third largest source between 1999 and 2017.*

Line 540 - *Figure S17, however, also shows that emissions from coal combustion are simultaneously increasing in India, the Other Asia/Pacific region, and Africa.*

Line 544 - *In contrast, biofuel emissions from all other regions remain relatively flat or increase between 1970 and 2017, though biofuel emissions of NMVOCs, CO, $SO_2$, and OC in India, as well as $SO_2$ emissions in North America both decrease between 2010 and 2017 (Fig. S18). In 2017, biofuel*

*emissions of all compounds are dominated by emissions from either Africa (NO$_x$, SO$_2$, NH$_3$, NMVOC, BC) or India (OC).*

*Line 550 - In contrast to other combustion sectors and fuels, emissions of NO$_x$, CO, NMVOCs, BC, and OC from the combustion of liquid fuels and natural gas in China remain relatively flat or slightly decrease between 2010 and 2017. Dominant global regions vary by compound (Fig. S19) and include International Shipping (NOx, SO$_2$), Africa (OC), India (BC), North America (CO, NH$_3$), and the Other Asia/Pacific region (NMVOCs).*

*Line 559 - Dominant source regions in 2017 of these process level emissions include China (NO$_x$, CO, NH$_3$, BC, OC), India (SO$_2$), and African countries (NMVOCs) (Fig. S20).*

*Line 903 - Outside of international shipping, China is the largest regional source of global emissions of all compounds other than NMVOCs. As emissions in North America, Europe, and China continue to decrease, global emissions of NO$_x$, CO, SO$_2$, BC, and OC will increasingly reflect emissions in rapidly growing regions such as Africa, India, and countries throughout Asia, Latin America, and the Middle East. Lastly, in contrast to other compounds, global emissions of NMVOCs and NH$_3$ continuously increase over the entire time period. These increases are predominantly due to increases in agricultural NH$_3$ emissions in nearly all world regions, as well as NMVOCs from increased waste, energy sector, and solvent use emissions. In 2017, global emissions of these compounds have the largest regional contributions from India, China, and countries throughout Africa, Asia, and the Pacific.*

Compliments to the authors for all the line plots, they are generally really good to read and intercompare and thereby also reveal some issues that appear unlikely to be correct. That does not mean they have to (or even can be) solved in the current paper. There are a few individual cases that draw attention and possibly merit more comments. I like to share them but it is also up to the authors to think about what they feel is justified. I am not advocating to make the paper very anecdotical by discussing every detail. Like the drop in OC emissions for Industry in Figure S6; the CO peak from road transport and SO2 peak for energy in fig S8 (the latter is discussed in the text) - My suspicion is that what such abrupt peaks or drops have in common is most likely a change in legislation or methodology that "on paper" has almost immediate effect but in reality is smeared out over a longer time. For example the car fleet cannot be changed in 1-2 years, cleaner fuels (like low sulphur) generally take years to be completely adopted. NMVOCs from the Energy sector appear a special case (Fig S5) with a very large contribution but little explanation is given other than that these are process emissions. NMVOCs in general draw some attention – e.g. in line 630 there is a discrepancy of possibly missing NMVOC emissions as CEDS has no agricultural NMVOC emission? And, for example Fig3 India NOx emissions – almost a factor 2 difference between 2 CEDS versions. It is commented on in the text but would it also imply it is better not to use the previous CEDS version because of these large deviations? The difference is too large for both to be equally

plausible. This also applies to the discussion in line 568 and onward. As both inventories come from the CEDS team it seems logical to express some advice on to what extend you believe the new inventory replaces the old one. (Like the EDGAR team would advise to use v5 and v3 or v4. )

The Reviewer has highlighted some of the notable features in the region, fuel, and sector-specific line plots in the main text and supplement. These features highlight additional uncertainties in the CEDS system that were not explicitly discussed in the original text:  1) uncertainties and discontinuities in the IEA energy consumption data, and 2) uncertainties in the methodology of the CEDS scaling procedure. To provide a further analysis of some of these features, we have added two paragraphs to the main text that discuss these additional sources of uncertainty. We have chosen to discuss the specific features that the Review has noted as these are some of the most striking and provide a means to illustrate how these sources of uncertainty can impact the final emission estimates. We have added/edited the following text.

Line 676 - ***4.2.1 Uncertainties in Activity Data***
*As discussed in Section 2.1, CEDS default emissions from combustion sources are largely informed by fuel consumption data from the IEA 2019 World Energy Statistics Product (IEA, 2019). While this database provides energy consumption data as a function of detailed source sector and fuel-type for most countries, the IEA data is uncertain and includes breaks in time-series data that can lead to abrupt changes in the CEDS$_{GBD-MAPS}$ emissions for select sectors, fuels, and countries. For example, Fig. S6 shows an order of magnitude decrease (0.1 TgC) in OC industrial emissions from North America between 1992 and 1993, which is driven by a break in IEA biofuel consumption data for the non-specified manufacturing industry sector (CEDS sector: 1A2g_Ind-Comb-other) in the United States. While the magnitude of this particular change is negligible on the global scale, this is not the case for all sectors. For example, as noted in Section S4, a known issue in the IEA data in China in the energy sector causes peaks in the associated $NO_x$ and $SO_2$ CEDS$_{GBD-MAPS}$ emissions in 2004. These peak emissions may be over-estimated by up to 4 and 10 Tg, respectively, which is large enough to impact historical trends in both regional (Figure 8: $NO_x$ and $SO_2$) and global (Figures6-7: $SO_2$) emissions. These point to areas where improvements could be made to the underlying driver data in future work.*

Line 761 - *In addition to uncertainties in the scaling inventory emissions, uncertainties are also introduced by the CEDS$_{GBD-MAPS}$ scaling procedure. Uncertainties arise when mapping sectoral and fuel (when available) specific emissions between inventories (as discussed previously), as well as in the application of the calculated scaling factors outside the range of available scaling inventory years. For example, the implied CO EFs in Figure S2 highlight one case in China where the EFs for oil and gas combustion in the on-road transport sector peak in 1999 at a value over three times larger than EFs in all other top emitting countries. For China specifically, the calculated scaling factors for the year 2010 (earliest scaling inventory year) are applied to emissions from all years prior, which was calculated as a value of ~1.58 for the on-road transport sector. The implied EF of ~1.8 g g$^{-1}$ for*

*this sector in 2003 (Figure S2) suggests that the SF from 2010 may not be representative of emissions during this earlier time period. We do note, however, that the 1999 peak in total CO emissions in China (Figure S9) is driven by the IEA energy data and is consistent with the CEDS$_{Hoesly}$ inventory (Hoesly et al., 2018). In contrast, EFs from this sector in China after the year 2010 agree with the magnitude and trends found in other countries, further indicating that the scaling factors are most appropriate for years with overlapping inventory data. Other similar examples include coal energy emissions of SO$_2$ in Thailand (Figure S2). In this case, the REAS scaling inventory spans the years 2000 – 2008. The default EFs for the energy sector, however, independently decrease between 1997 and 2001. As a result, when the implied EF of 3.3 for the year 2000 is applied to all historical energy emissions, the implied EFs prior to 1997 become an order of magnitude larger than those in nearly all other top emitting countries (Figure S2). Overall, the applicability of the scaling factors to emissions in years outside the available scaling inventory years remain uncertain due to real historical changes in activity, fuel-use, and emissions mitigation strategies. These uncertainties, however, vary by compound and sector as, for example, there are no similar peaks in on-road emissions for compounds other than CO in China.*

To answer the Reviewers other specific comments/questions, the CEDS$_{GBD-MAPS}$ inventory does not include agricultural emissions of NMVOCs, nor do the EDGAR or GAINS inventories. In the original Fig. S22, it appeared that the GAINS inventory included AGR NMVOC emissions and that the CEDS inventory did not. There was an error in the color scale of this figure that was displaying GAINS NMVOC solvent emissions as those from the AGR sector. This color scale has been corrected in the new Fig. S24.

[Figure]

Figure S24.

The increase in energy NMVOC emissions noted by the Reviewer is largely associated with large increases in the Fugitive solid fuels sector in Africa between 2003 and 2017 in the EDGAR v4.3.2 inventory. As these emissions are assigned to the 'process' fuel-type in CEDS$_{GBD-MAPS}$, these emissions are taken directly from the EDGAR inventory. For instance, NMVOC emissions from this sector in EDGAR increased by nearly a factor of 5 in Nigeria during this time period. The source of the increase in this sector has been clarified on the following lines.

Line 347 - *Total CEDS$_{GBD-MAPS}$ emissions of NMVOCs are larger, primarily due to increased contributions from solvent use and the energy sector associated with changes in the EDGAR v4.3.2 inventory, while total emissions of CO, SO$_2$, and NH$_3$ are relatively consistent between the two CEDS versions.*

Line 524 - *Increases in energy sector emissions after 2003 are largely driven by increases in fugitive emissions from select African countries, including Nigeria, Kenya, and Angola, and Mozambique.*

For NO$_x$ emissions in India, the original Fig. 3 shows that CEDS$_{GBD-MAPS}$ emissions are reduced by ~40%, due largely to road emissions. This sector is particularly uncertain in India due to uncertainties in the vehicle fleet. As discussed in Section 2, however, the road emissions in India are scaled to match the recent SMoG-India inventory. As previously discussed in Section 2.2, major difference between the two CEDS inventories for this sector are due to differences in the employed emission factors. We have added additional discussion that note that the scaled implied emission factors are within the range used for the road transport sector in the SMoG-India inventory, further suggesting that the large decrease in road emissions in India are resulting from the smaller NOx emissions factor. As the CEDS$_{GBD-MAPS}$ inventory presented in this manuscript was developed for the GBD-MAPS project (and not an updated release of the core CEDS system) we have discussed the differences between the two inventories in this manuscript, but refrain from making more specific recommendations. We have clarified this point in the Methods section.

Line 159 - *The CEDS$_{GBD-MAPS}$ inventory is developed for the GBD-MAPS project and is not an updated release of the core CEDS emissions inventory.*

Line 648 – "decreasing uncertainties": Here I do not by definition agree. If for example the (more uncertain) emissions from Africa and India become dominant and e.g. the more certain emissions from the US & EU go down, than the overall uncertainty might also increase in future years.

The reviewer raises a good point and we have updated the relevant text accordingly.

Line 695- *While improvements in data collection and reporting standards may decrease the uncertainty in some underlying sources overtime, the most recent years of CEDS$_{GBD-MAPS}$ emissions are still subject to high levels of uncertainty. For instance, the degree of local and national*

*compliance with control measures is often variable or unknown (e.g., Wang et al., 2015;Zheng et al., 2018), recent activity and regional emissions data are often updated as new information becomes available, and emissions in generally more uncertain regions, including India and Africa are becoming an increasingly large fraction of global totals. Additionally, from a methodological standpoint, default CEDS emissions after 2010 also currently rely on the projection of emission factors from the GAINS EMF30 data release for sectors and countries where contemporary regional scaling inventories are not available.*

A good assessment of uncertainty from a mosaic inventory is very challenging and simply stating that the uncertainty is similar to the other inventories (e.g. line 655) is an unsatisfactory answer. Moreover, there may also be considerable uncertainty in the spatial distribution. The authors, however, announce that in the near future a more robust uncertainty analysis is planned. And a much longer paper would not be helpful for the community. So separating this is an acceptable solution.

As this Reviewer points out, it is challenging to quantify uncertainties in emission estimates, especially for those derived using a mosaic approach. Core CEDS system uncertainties have been previously described in Hoesly et al. (2018), and a more robust uncertainty analysis is planned for an upcoming release of the core CEDS system, as mentioned on line 788. We have provided a summary of the sources of uncertainties in this inventory, including uncertainties in global bottom up inventories (Section 4.2.2), regional-level inventories (including the reported uncertainties in the few studies where they were reported) (Section 4.2.3), sectoral and fuel contributions (Section 4.2.4), as well as those in the gridded emission files (Section 4.2.5). We have also added two additional paragraphs to the uncertainties section as well as an addition supplemental figure of implied emission factors in order to provide a further discussion and analysis of inventory uncertainties.

Additional suggestions for final discussion: Recently Huneeus et al. (2020) published an evaluation of emission inventories for South America which included EDGSAR, ECLIPSE and CEDS. It would be interesting to comment on how the new inventory presented here would have an impact on SA estimates and compares to the CEDS version used in that paper?

In Section 2, we highlight the major updates implemented in this work relative to the previous CEDS_Hoesly inventory. These updates will result in differences at the global level (discussed explicitly in Section 4.1.1), regional level (discussed for China, India, and Africa in Sections 2.2-2.3 and throughout Section 4), and in the gridded emission products. From a global perspective, we discuss on line 615 about how differences reflect the different activity data and scaling inventories. Per the Reviewer's question/comment, the updated CEDS emissions in Latin American countries will therefore largely reflect the changes between the EDGARv4.3.1 (used to derived CEDS_Hoesly) and EDGARv4.3.2 (used for CEDS_GBD-MAPS), except for in Argentina (see Fig. 2). The Reviewer, however,

has highlighted an important point that we had not provided a comparison to the previous CEDS_Hoesly inventory across all regions. Therefore, we have added Figure S22 to the supplement to highlight the regional inventory differences. Based on this figure, the CEDS_GBD-MAPS emissions in Latin America are slightly lower than CEDS_Hoesly, which would indicate a slight improvement in the agreement with other inventories shown for select Latin American countries in Figure 4 of Huneeus et al. (2020).

We have added the following text to the end of Section 4.1.1: Comparison to CEDS_Hoesly Inventory.

Line 609 - *Similar to the total global emissions, changes between the two CEDS versions for the national-level and 0.5°×0.5° gridded products will also result from updates to the energy consumption data, scaling inventories (Section 2.2-2.3) and spatial distribution proxies from EDGARv4.3.2 (Section 2.1). Time series of differences between the CEDS_Hoesly and CEDS_GBD-MAPS inventories for 11 world regions are shown for each compound in Fig. S22. In recent years, Figure S22 shows that CEDS_GBD-MAPS emissions are generally lower in each region, with the greatest differences in Africa, India and China. The relative changes in Africa and India are discussed previously in Section 2. For China, the CEDS_GBD-MAPS emissions are generally lower than the CEDS_Hoesly estimates after the year 2010 as a result of the updated scaling inventory. Regional differences between inventories are also greater for OC and BC emissions relative to other compounds due to the added scaling procedure discussed in Section 2. Differences in spatial distributions are not discussed here as changes represent differences in the spatial proxies, which are largely from updates to the EDGAR inventory.*

We have added Figure S22.

[Figure]

**Figure S22.** *Comparison of CEDS_Hoesly and CEDS_GBD-MAPS emissions as a function of 11 world regions.*

SI Line 385 - *Figures S21 and S22 compare CEDS_GBD-MAPS and CEDS_Hoesly emissions.*

Small editorial remarks

Line 42 – from "waste" combustion (otherwise strange to have carb aerosol from waste.)

*To keep consistent terminology throughout the text, we have changed to 'the waste sector'.*
*Line 39 - Dominant sources of global CO emissions in 2017 include on-road transportation and residential biofuel combustion. Dominant global sources of carbonaceous aerosol in 2017 include residential biofuel combustion, on-road transportation (BC only), as well as emissions from the waste sector.*

Line 78 – as inputs to solve for? Not clear to me, maybe reformulate slightly?

*Updated as follows.*
*Line 76 - For example, spatially gridded emission inventories are used as inputs in general circulation/climate (GCM) and chemical transport models (CTM), which are used to predict the evolution of atmospheric constituents over space and time.*

 Line 108 – "emission" reduction of coal-fired etc.

*Changed*

Line 181 – explain the term "working sector"
*This terminology is from Hoesly et al. (2018) and refers to the sub-sectors that are carried through the CEDS system calculations and later aggregated for the final reported CEDS sectors. This term was first used 4 lines prior and is now further defined there.*

*Line 176- In Eq. (1), emissions are calculated using relevant activity (A) and emission factor (EF) data for each country (c) and year (y), as a function of 52 detailed working sectors (s) (sub-sectors used for intermediate steps in the CEDS system) and nine working fuel-types (f) (Table 2).*

Line 410 – you mean Section S4.

*Changed*

Line 481 Global emissions of NOx from waste "combustion".

*Changed*

Line 680 I don't see how satellites will aid in fuel-type recognition.

In this case, we were considering examples such as from McLinden et al. (2016), where top-down estimates from specific point sources could be identified and/or better quantified by incorporating satellite-derived estimates. We did not mean to say that satellite-retrievals will aid in identifying the fuel used by various point sources, but rather that they can aid in reducing the uncertainties for point sources where fuel-type is already known.

Line 684 "emissions" – should be "uncertainties"?

We have clarified this sentence.
Line 732- *The inventories with the largest impact on the CEDS$_{GBD-MAPS}$ emission uncertainties relative to the CEDS$_{Hoesly}$ inventory will be those from China from Zheng et al. (2018), the DICE-Africa emission inventory from Marais and Wiedinmyer (2016), and the SMoG-India inventory from Venkataraman et al. (2018).*

Line 786 – but not for the latest years? And these will not be scaled ("calibrated") so not consistent?

We clarified this sentence.
Line 854 - *These compounds were previously included through 2014 in the CEDS$_{Hoesly}$ inventory.*

Line 791 – it seems the reference of (McDuffie et al., 2020c) here and in the ref list is redundant because this sis the dataset connected to the present paper? So won't the reference to that data not be simply this paper instead of (McDuffie et al., 2020c).

Yes, the dataset is connected to the paper. However, the dataset is also publicly available at Zenodo, which has a unique data doi and is not directly tied to this manuscript. Therefore, the data doi (McDuffie t al., 2020c) is for the dataset. The description of the dataset is provided in this manuscript, which will have a unique doi. Having a data-specific doi will also allow for the creation of new doi's on Zenodo when there are future updates to the CEDS_GBD-MAPS dataset.

Line 836 in agricultural "NH3" emissions

Changed

Line 867 – what is fuel abatement?

Changed.

*Line 936 - Due to the direct and secondary contribution of these reactive gases and carbonaceous aerosol to ambient air pollution, contemporary gridded and country-level emissions with both sector and fuel-type information can provide new insights necessary to motivate and develop effective strategies for emission reductions and air pollution mitigation around the world.*

There is an error in Table S8 – Other Asia includes Montenegro and I assume Chinese Taipei is Taiwan?

Changed both (Table S8 now Table S9)

There is an error in Table S9 – in the column for EDGAR "solvent use" and "waste" are swapped.

Changed. (Table S9 now Table S10)

[revised manuscript text omitted]

$$Em_{species}^{country, \ sector, \ fuel, \ year} = A^{c, \ s, \ f, \ y} \times EF_{species}^{c, \ s, \ f, \ y} \tag{1}$$

For emissions from CEDS combustion sources, annual activity drivers in Eq. (1) primarily include country-, fuel-, and sector-specific energy consumption data from the International Energy Agency (IEA, 2019). Sector- and compound-specific emission factors are typically derived from energy use and total emissions reported from other inventories, including
215  from the GAINS model (Klimont et al., 2017;IIASA, 2014;Amann et al., 2015), Speciated Pollutant Emission Wizard (SPEW) (Bond et al., 2007), and the US National Emissions Inventory (NEI) (NEI, 2013). For International Shipping, IEA activity data is supplemented with consumption data and EFs from the International Maritime Organization (IMO), as described in Hoesly et al. (2018) and its supplement. In contrast, default emissions (Em) for CEDS process sources are directly taken from other inventories, including from the EDGAR v4.3.2 global emission inventory (EC-JRC, 2018;Crippa et al., 2018). "Implied
220  emission factors" are then calculated for these process sources in Eq. (1) using global population data (UN, 2019, 2018) or pulp and paper consumption (FAOSTAT, 2015) as the primary activity drivers. For years without available emissions, default estimates for CEDS process sources are calculated in Eq. (1) from a linear interpolation of the "implied emission factors" and available activity data (A) for that year. Supplemental Sect. S2.1 and S2.2 provide additional details regarding the input datasets for activity drivers and emission factors used for both CEDS combustion and process source categories.

225  While CEDS Step 1 is designed to provide a complete set of historical emission estimates, CEDS Step 2 scales these total default emission estimates to existing, authoritative global, regional, and national-level inventories. As described in Hoesly et al. (2018), CEDS uses a "mosaic" scaling approach to retain detailed fuel- and sector-specific information across different inventories, while maintaining consistent methodology over space and time. The development and use of mosaic inventories has been recently increasing as they provide a means to utilize detailed local emissions, while harmonizing this
230  information across large regional or global scales (Li et al., 2017c;Janssens-Maenhout et al., 2015). The CEDS approach, however, differs from previous mosaic inventories, such as that developed for the HTAP project (Janssens-Maenhout et al., 2015), as local and regional inventories in CEDS$_{GBD-MAPS}$ are used to scale sectoral emissions at the national-level, rather than merging together spatially distributed gridded estimates.

[revised manuscript text omitted]

445    on-road transport sector in India is ~0.015 g g$^{-1}$ in 2015, which falls within the range of values of 0.0026 – 0.046 g g$^{-1}$ used for various vehicles and fuel type in Venkataraman et al. (2018). Similarly, NO$_x$ transport emissions are also lower in CEDS$_{GBD-MAPS}$ relative to the EDGAR and GAINS inventories. Causes of other reductions are mixed. For example, lower emissions of SO$_2$ and NMVOCs are largely associated with the energy sector, while reductions in the industry sector contribute to reduced CO emissions. For SO$_2$, Figure S2 shows that the implied EF for coal combustion in the energy sector is ~0.004 g g$^{-1}$, slightly

450    lower than the range of 0.0049 – 0.0073 g g$^{-1}$ used for the SMoG-India inventory.

[revised manuscript text omitted]
 that 50% of global 2017 NO$_x$ emissions are from the combined Other Asia/Pacific region (Table S9) (13 Tg), China (24 Tg), International Shipping (25 Tg). For global

2017 emissions of remaining gas-phase pollutants, 67% of CO emissions are from the on-road (100%: oil + gas) and residential (86%: biofuel) sectors, 78% of $SO_2$ emissions are from the energy generation (63%: coal) and industry (38% coal, 36% process, 25% oil + gas) sectors, 89% of $NH_3$ emissions are from the agriculture (100%: process) and waste (100%: process) sectors, and emissions of NMVOCs have the largest single contribution (36%) from the energy sector, 99% of which are associated with CEDS$_{GBD-MAPS}$ process sources (Table 2). For carbonaceous aerosol in 2017, 58% of global BC emissions are from the residential (70%: biofuel) and on-road (100%: oil + gas) sectors, while 67% of global OC emissions are from the residential (92%: biofuel) and waste (100%: process) sectors. Fig. 8 shows that in 2017, China is the dominant source of global CO (144 Tg, 27% of global total), $SO_2$ (12 Tg, 15% of global total), $NH_3$ (12 Tg, 20% of global total), OC (2.7 TgC, 20% of global total), and BC (1.4 TgC, 24% of global total). In contrast, Africa is the dominant source of global NMVOCs in 2017 (48 TgC, 27% of global total) and International Shipping is the dominant source of global $NO_x$ emissions (25 Tg, 20% of global total).

[revised manuscript text omitted]
. S12). Increases in energy sector emissions after 2003 are largely driven by increases in fugitive emissions from select African countries, including Nigeria, Kenya, and Angola, and Mozambique. Emissions from China are the second largest global NMVOC source between 1996 and 2017 (Fig. 8), while the Other Asia/Pacific region is the third largest source between 1999 and 2017. Total NMVOCs in China increase by a factor of 3.4 between 1970 and 2017 due to activity increases in the solvent, energy, and industry sectors (Zheng et al., 2018), while targeted emission controls for the residential and on-road transport sectors result in their reduced contributions to NMVOC emissions between 2012 and 2017 (Fig. S9). Total emissions of NMVOCs in Europe and North America decrease by up to a factor of 2.4 between 1970 and 2017, due to reductions in all source sectors, except for energy emissions in North America, which increase between 2007 and 2011 and remain flat through 2017 (Fig. S7).

To provide a fuel-centric perspective of global historical emissions trends, Fig. 7 illustrates the contributions from the combustion of coal, solid biofuel, the sum of liquid fuel and natural gas, as well as all remaining CEDS 'process-level' sources (Table 2) to total global emissions between 1970 and 2017. Reductions discussed above between 2010 and 2017 for global emissions of $NO_x$, CO, $SO_2$, BC and OC, are largely associated with reductions in coal combustion from the energy, industry, and residential sectors associated with emission control policies and residential fuel replacement in China, as well as coal-fired power plant reductions in North America and Europe (Fig. 7, S13, S17). Despite large reductions in emissions, China is still the single largest source of global emissions from coal combustion in 2017 (23-64% for each compound except $NH_3$). Figure S17, however, also shows that emissions from coal combustion are simultaneously increasing in India, the Other Asia/Pacific region, and Africa. Specifically, $SO_2$ emissions from coal combustion in India are set to surpass those from China by 2018 if recent $CEDS_{GBD-MAPS}$ trends hold. For biofuel combustion, global emissions of all compounds are primarily associated with the residential sector (Fig. S14), with recent reductions in biofuel CO, $SO_2$, BC, and OC emissions largely from reductions in China (Fig. S18). In contrast, biofuel emissions from all other regions remain relatively flat or increase

between 1970 and 2017, though biofuel emissions of NMVOCs, CO, SO$_2$, and OC in India, as well as SO$_2$ emissions in North America both decrease between 2010 and 2017 (Fig. S18). In 2017, biofuel emissions of all compounds are dominated by emissions from either Africa (NO$_x$, SO$_2$, NH$_3$, NMVOC, BC) or India (OC). For oil and gas combustion, global emissions of all compounds are primarily associated with on-road transportation, international shipping, and energy and industry (SO$_2$ only) sectors, with general decreases in associated emissions in North America and Europe between 1970 and 2017 and increases in other regions (Fig. S19). In contrast to other combustion sectors and fuels, emissions of NO$_x$, CO, NMVOCs, BC, and OC from the combustion of liquid fuels and natural gas in China remain relatively flat or slightly decrease between 2010 and 2017. Dominant global regions vary by compound (Fig. S19) and include International Shipping (NO$_x$, SO$_2$), Africa (OC), India (BC), North America (CO, NH$_3$), and the Other Asia/Pacific region (NMVOCs). Global CEDS process source emissions, which include contributions from some fuel combustion processes (Table 2), decrease between 2010 and 2017 for CO, SO$_2$, BC, and OC. These trends are primarily associated with reductions in emissions from the energy and industry sectors. In contrast, process source contributions to NO$_x$, NH$_3$, and NMVOCs increase over this same time period due to increases in non-combustion agricultural and solvent use emissions, as well as emissions from waste disposal and energy generation and transformation. Increases in emissions from these sectors between 1970 – 2017 drive the continuous increases in NH$_3$ and NMVOCs, discussed above. Dominant source regions in 2017 of these process level emissions include China (NO$_x$, CO, NH$_3$, BC, OC), India (SO$_2$), and African countries (NMVOCs) (Fig. S20).

**4 Discussion**

**4.1 Comparison to Global Inventories**

**4.1.1 Comparison to CEDS$_{Hoesly}$ Inventory**

As a result of the similar methodologies, Fig. 6 shows that CEDS$_{GBD-MAPS}$ and CEDS$_{Hoesly}$ emission inventories predict similar magnitudes and historical trends in global emissions of each compound between 1970 and 2014. The two inventories, however, diverge in recent years due to the incorporation of updated activity data and both updated and new scaling emission inventories included in the CEDS$_{GBD-MAPS}$ system. For global emissions of NO$_x$, CO, and SO$_2$, the CEDS$_{GBD-MAPS}$ emissions are smaller than the CEDS$_{Hoesly}$ emissions after 2006 and show a faster decreasing trend. By 2014, global emissions of these compounds are between 7 and 21% lower than previous CEDS$_{Hoesly}$ estimates. These differences are largely associated with large emission reductions in China as a result of the updated national-level scaling inventory from Zheng et al. (2018), along with the added DICE-Africa (Marais and Wiedinmyer, 2016) and SMoG-India (Venkataraman et al., 2018) scaling inventories. Differences in emissions from India and Africa in the two CEDS inventories are discussed in Sect. 2 (Fig. 3) and combined, account for ~60% of the reduction in global NO$_x$ emissions, 23% of the reduction in global CO, and 14% of the reduction in global SO$_2$. The largest differences between these two inventories in India and Africa are the reduced NO$_x$ emissions from the transport sector, as well as reduced energy emissions of SO$_2$ in India. Remaining differences between NO$_x$ and SO$_2$ emissions in the

two CEDS inventories are largely associated with the updated China emission inventory from Zheng et al. (2018), which reports lower emissions in 2010 and 2012 than a previous version of the MEIC inventory that was used to scale China emissions in the CEDS_Hoesly inventory (Li et al., 2017c). These emission reductions are largely associated with the industrial and residential sectors in China and are partially offset by a simultaneous increase in transportation emissions of all compounds

775 relative to CEDS_Hoesly.

For global emissions of $NH_3$ and NMVOCs, these species remain relatively unchanged between the CEDS_Hoesly and CEDS_GBD-MAPS inventories. In 2014 CEDS_GBD-MAPS emissions are 5% higher than CEDS_Hoesly emissions for NMVOCs and 2% lower than CEDS_Hoesly global $NH_3$ emissions. Emissions of $NH_3$ remain relatively unchanged (within <2%) from dominant source regions, including India, Africa (Fig. 3), and China. In contrast, emissions of NMVOCs from Africa and China in the

780 DICE-Africa and Zheng et al. (2018) scaling inventories are larger than those in the CEDS_Hoesly inventory. Global emissions of NMVOCs are also higher in EDGARv4.3.2 inventory relative to the previous version used in the CEDS_Hoesly inventory. NMVOCs are particularly large from the process energy sector emissions in Africa (Figure S12), which primarily include fugitive emissions from oil and gas operations (Table 2). Default energy sector emissions from 'non-combustion' processes are taken from the EDGAR inventory and are not scaled to DICE-Africa inventory. Therefore, the large increase in these

785 emissions in Africa relative to CEDS_Hoesly are largely driven by changes in the EDGAR v4.3.2 inventory, with emissions from the 1B2_Fugitive_Fossil fuels sector, increasing for example by a factor of 5 in Nigeria between 2003 and 2017.

Global emissions of OC and BC have the largest differences between the two CEDS inventories, with CEDS_GBD-MAPS emissions consistently smaller than CEDS_Hoesly emissions between 1970 and 2014. By 2014, CEDS_GBD-MAPS emissions of BC and OC are 24 and 33% smaller than corresponding CEDS_Hoesly emissions. In the CEDS_Hoesly inventory, default emissions of

790 BC and OC are not scaled and therefore these differences are largely associated with the added scaling inventories, discussed in Sect. 2 and shown in Table 3. As shown in Fig. S3-S4, the added scaling of BC and OC emissions leads to a reduction in global CEDS_GBD-MAPS emissions of OC in all scaled regions, and a reduction in BC emissions in all regions other than India. In India, increases in industry and residential BC emissions from the SMoG-India scaling inventory result in a slight increase in BC emissions relative to the CEDS_Hoesly inventory (Fig. 3). Waste emissions of OC and BC are also reduced in the CEDS_GBD-

795 MAPS inventory due to updated assumptions for the fraction of waste burned (Sect. S1.1). As discussed in Hoesly et al. (2018) and further below, BC and OC emissions typically have the largest uncertainties of all the emitted species and their recent changes in the residential and waste sectors are particularly uncertain.

The relative contributions of each source sector to emissions in the two CEDS versions are additionally shown in Fig. S21. This comparison shows that the fractional sectoral contributions to global emissions in 2014 are the same to within 10%

800 in the two CEDS inventories. The largest differences are a 9% increase in the relative contribution of on-road transportation emissions of CO and reductions in the relative contribution of waste emissions across all compounds. These trends reflect the large update to default waste emissions described above as well as changes associated with the DICE-Africa and national China scaling inventories.

815       Similar to the total global emissions, changes between the two CEDS versions for the national-level and 0.5°×0.5° gridded products will also result from updates to the energy consumption data, scaling inventories (Section 2.2-2.3) and spatial distribution proxies from EDGARv4.3.2 (Section 2.1). Time series of differences between the CEDS$_{Hoesly}$ and CEDS$_{GBD-MAPS}$ inventories for 11 world regions are shown for each compound in Fig. S22. In recent years, Figure S22 shows that CEDS$_{GBD-MAPS}$ emissions are generally lower in each region, with the greatest differences in Africa, India and China. The relative changes

820 in Africa and India are discussed previously in Section 2. For China, the CEDS$_{GBD-MAPS}$ emissions are generally lower than the CEDS$_{Hoesly}$ estimates after the year 2010 as a result of the updated scaling inventory. Regional differences between inventories are also greater for OC and BC emissions relative to other compounds due to the added scaling procedure discussed in Section 2. Differences in spatial distributions are not discussed here as changes represent differences in the spatial proxies, which are largely from updates to the EDGAR inventory.

[revised manuscript text omitted]

As discussed in Section 2.1, CEDS default emissions from combustion sources are largely informed by fuel consumption data from the IEA 2019 World Energy Statistics Product (IEA, 2019). While this database provides energy consumption data as a function of detailed source sector and fuel-type for most countries, the IEA data is uncertain and includes breaks in time-series data that can lead to abrupt changes in the CEDS$_{GBD-MAPS}$ emissions for select sectors, fuels, and countries. For example, Fig.

895 S7 shows an order of magnitude decrease (0.1 TgC) in OC industrial emissions from North America between 1992 and 1993, which is driven by a break in IEA biofuel consumption data for the non-specified manufacturing industry sector (CEDS sector: 1A2g_Ind-Comb-other) in the United States. While the magnitude of this particular change is negligible on the global scale, this is not the case for all sectors. For example, as noted in Section S4, a known issue in the IEA data in China in the energy sector causes peaks in the associated NO$_x$ and SO$_2$ CEDS$_{GBD-MAPS}$ emissions in 2004. These peak emissions may be over-

900 estimated by up to 4 and 10 Tg, respectively, which is large enough to impact historical trends in both regional (Figure 8: NO$_x$ and SO$_2$) and global (Figures 6-7: SO$_2$) emissions. These point to areas where improvements could be made to the underlying driver data in future work.

**4.2.2 Uncertainties in Global Bottom-Up Inventories**

Uncertainties in bottom-up emission inventories vary as a function of space, time, and compound, making total uncertainties

905 difficult to quantify. Default emission estimates in the CEDS system are subject to uncertainties in underlying activity data, such as IEA energy consumption data, as well as activity drivers for process-level emissions. Knowledge of accurate emission factors also drive inventory uncertainties as these are not often available for all sectors in countries with emerging economies, and are heavily dependent on the use, performance, and enforcement of control technologies within each sector and country (e.g., Zhang et al., 2009;Wang et al., 2015). While improvements in data collection and reporting standards may decrease the

910 uncertainty in some underlying sources overtime, the most recent years of CEDS$_{GBD-MAPS}$ emissions are still subject to considerable uncertainty. For instance, the degree of local and national compliance with control measures is often variable or unknown (e.g., Wang et al., 2015;Zheng et al., 2018), recent activity and regional emissions data are often updated as new information becomes available, and emissions in generally more uncertain regions, including India and Africa are becoming an increasingly large fraction of global totals. Additionally, from a methodological standpoint, default CEDS emissions after

[revised manuscript text omitted]

In addition to uncertainties in the scaling inventory emissions, uncertainties are also introduced by the $CEDS_{GBD-MAPS}$ scaling procedure. Uncertainties arise when mapping sectoral and fuel (when available) specific emissions between inventories (as discussed previously), as well as in the application of the calculated scaling factors outside the range of available scaling inventory years. For example, the implied CO EFs in Figure S2 highlight one case in China where the EFs for oil and gas
000 combustion in the on-road transport sector peak in 1999 at a value over three times larger than EFs in all other top emitting countries. For China specifically, the calculated scaling factors for the year 2010 (earliest scaling inventory year) are applied to emissions from all years prior, which was calculated as a value of ~1.58 for the on-road transport sector. The implied EF of ~1.8 g $g^{-1}$ for this sector in 2003 (Figure S2) suggests that the SF from 2010 may not be representative of emissions during this earlier time period. We do note, however, that the 1999 peak in total CO emissions in China (Figure S9) is driven by the IEA
005 energy data and is consistent with the $CEDS_{Hoesly}$ inventory (Hoesly et al., 2018). In contrast, EFs from this sector in China after the year 2010 agree with the magnitude and trends found in other countries, further indicating that the scaling factors are most appropriate for years with overlapping inventory data. Other similar examples include coal energy emissions of $SO_2$ in

Thailand (Figure S2). In this case, the REAS scaling inventory spans the years 2000 – 2008. The default EFs for the energy sector, however, independently decrease between 1997 and 2001. As a result, when the implied EF of 3.3 for the year 2000 is applied to all historical energy emissions, the implied EFs prior to 1997 become an order of magnitude larger than those in nearly all other top emitting countries (Figure S2). Overall, the applicability of the scaling factors to emissions in years outside the available scaling inventory years remain uncertain due to real historical changes in activity, fuel-use, and emissions mitigation strategies. These uncertainties, however, vary by compound and sector as, for example, there are no similar peaks in on-road emissions for compounds other than CO in China.

[revised manuscript text omitted]

OC sources are from the residential biofuel and the waste sector. Outside of international shipping, China is the largest regional source of global emissions of all compounds other than NMVOCs. As emissions in North America, Europe, and China continue to decrease, global emissions of $NO_x$, CO, $SO_2$, BC, and OC will increasingly reflect emissions in rapidly growing regions such as Africa, India, and countries throughout Asia, Latin America, and the Middle East. Lastly, in contrast to other compounds, global emissions of NMVOCs and $NH_3$ continuously increase over the entire time period. These increases are predominantly due to increases in agricultural $NH_3$ emissions in nearly all world regions, as well as NMVOCs from increased waste, energy sector, and solvent use emissions. In 2017, global emissions of these compounds have the largest regional contributions from India, China, and countries throughout Africa, Asia, and the Pacific.

Historical global emission trends in the $CEDS_{GBD-MAPS}$ inventory are generally similar to those in three other global inventories, $CEDS_{Hoesly}$, EDGAR v4.3.2, and ECLIPSE v5a (GAINS). Relative to the $CEDS_{Hoesly}$ inventory, however, $CEDS_{GBD-MAPS}$ emissions diverge in recent years, particularly for $NO_x$, CO, $SO_2$, 
[revised manuscript text omitted]

| *Energy Production (ENE)* | *Residential (RCOR)* |
| *Electricity and heat production* | *Res., Comm., Other - Residential* |
| 1A1a_Electricity-public (c) | 1A4b_Residential (c) |
| 1A1a_Electricity-autoproducer (c) | *Commercial (RCOC)* |
| 1A1a_Heat-production (c) | *Res., Comm., Other - Commercial* |
| *Fuel Production and Transformation* | 1A4a_Commercial-institutional (c) |
| 1A1bc_Other-transformation (p) | *Other (RCOO)* |
| 1B1_Fugitive-solid-fuels (p) | *Res., Comm., Other - Other* |
| *Oil and Gas Fugitive/Flaring* | 1A4c_Agriculture-forestry-fishing (c) |
| 1B2_Fugitive-petr-and-gas (p) | **Solvents (SLV)** |
| *Fuel Production and Transformation* | *Solvents (SLV)* |
| 1B2d_Fugitive-other-energy (p) | *Solvents production and application* |
| *Fossil Fuel Fires* | 2D_Degreasing-Cleaning (p) |
| 7A_Fossil-fuel-fires (p) | 2D3_Other-product-use (p) |
| **Industry (IND)** | 2D_Paint-application (p) |
| *Industry (IND)* | 2D3_Chemical-products-manufacture-processing (p) |
| *Industrial combustion* | **Agriculture (AGR)** |
| 1A2a_Ind-Comb-Iron-steel (c) | *Agriculture (AGR)* |
| 1A2b_Ind-Comb-Non-ferrous-metals (c) | *Agriculture* |
| 1A2c_Ind-Comb-Chemicals (c) | 3B_Manure-management (p) |
| 1A2d_Ind-Comb-Pulp-paper (c) | 3D_Soil-emissions (p) |
| 1A2e_Ind-Comb-Food-tobacco (c) | 3I_Agriculture-other (p) |
| 1A2f_Ind-Comb-Non-metalic-minerals (c) | 3D_Rice-Cultivation (p) |
| 1A2g_Ind-Comb-Construction (c) | 3E_Enteric-fermentation (p) |
| 1A2g_Ind-Comb-transpequip (c) | **Waste (WST)** |
| 1A2g_Ind-Comb-machinery (c) | *Waste (WST)* |
| 1A2g_Ind-Comb-mining-quarrying (c) | *Waste* |
| 1A2g_Ind-Comb-wood-products (c) | 5A_Solid-waste-disposal (p) |
| 1A2g_Ind-Comb-textile-leather (c) | 5E_Other-waste-handling (p) |
| 1A2g_Ind-Comb-other (c) | 5C_Waste-incineration (p) |
| 1A5_Other-unspecified (c) | 5D_Wastewater-handling (p) |
| *Industrial process and product use* | **Shipping (SHP)** |
| 2A1_Cement-production (p) | *Shipping (SHP)* |
| 2A2_Lime-production (p) | *International shipping* |
| 2A6_Other-minerals (p) | 1A3di_International-shipping (c) |
| 2B_Chemical-industry (p) | *Tanker Loading* |
| 2C_Metal-production (p) | 1A3di_Oil_Tanker_Loading (p) |
| 2H_Pulp-and-paper-food-beverage-wood (p) | |
| 2L_Other-process-emissions (p) | |
| 6A_Other-in-total (p) | |
| **Transportation (TRA)** | **Transportation Cont. (TRA)** |
| *Road Transportation (ROAD)* | *Non-Road Transportation (NRTR)* |
| *Road transportation* | *Non-road Transportation* |
| 1A3b_Road (c) | 1A3c_Rail (c) |
| | 1A3dii_Domestic-navigation (c) |
| | 1A3eii_Other-transp (c) |

| CEDS Fuels | |
|---|---|
| **Total** | |
| *Coal* | *Liquid Fuel & Natural Gas* |
| Brown coal | Heavy oil |
| Coal coke | Diesel oil |
| Hard coal | Light oil |
| *Biofuel* | Natural Gas |
| Biofuel | *Process* |
| | Process |

735

**Table 3. Scaling Inventories**

| Inventory Name | Scaled Inventory Years | Scaled Species | Reference |
|---|---|---|---|
| EDGAR v4.3.2 | 1992 – 2012 | CO, $NH_3$, NMVOCs, $NO_x$ | (EC-JRC, 2018) |
| EMEP NFR14 | 1990 – 2017 | CO, $NH_3$, NMVOCs, $NO_x$, $SO_2$, BC | (EMEP, 2019) |
| UNFCCC | 1990 – 2017 | CO, NMVOCs, $NO_x$, $SO_2$ | (UNFCCC, 2019) |
| REAS 2.1[a] | 2000 – 2008 | CO, $NH_3$, NMVOCs, $NO_x$, $SO_2$, BC | (Kurokawa et al., 2013) |
| APEI (Canada) | 1990 – 2017 | CO, $NH_3$, NMVOCs, $NO_x$, $SO_2$ | (ECCC, 2019) |
| US EPA | 1970, 1975, 1980, 1985, 1990 – 2017 | CO, $NH_3$, NMVOCs, $NO_x$, $SO_2$ | (US EPA, 2019) |
| MEIC (China) | 2008, 2010 – 2017 | CO, $NH_3$, NMVOCs, $NO_x$, $SO_2$, BC, OC | (Zheng et al., 2018;Li et al., 2017c) |
| Argentina[a] | 1990 – 1999, 2011 – 2009, 2011 | CO, NMVOCs, $NO_x$, $SO_2$ | (Argentina UNFCCC Submission, 2016) |
| Japan[a] | 1960 – 2010 | CO, $NH_3$, NMVOCs, $NO_x$, $SO_2$, BC, OC | (preliminary update from Kurokawa et al., 2013)[a] |
| NEIR (South Korea)[a] | 1999 –2012 | CO, NMVOCs, $NO_x$, $SO_2$ | (South Korea National Institute of Environmental Research, 2016) |
| Taiwan[a] | 2003, 2006, 2010 | CO, NMVOCs, $NO_x$, $SO_2$ | (TEPA, 2016) |
| NPI (Australia) | 2000 – 2017 | CO, NMVOCs, $NO_x$, $SO_2$ | (ADE, 2019) |
| DICE-Africa[b] | 2006, 2013 | CO, NMVOCs, $NO_x$, $SO_2$, BC, OC | (Marais and Wiedinmyer, 2016) |
| SMoG-India[b] | 2015 | CO, NMVOCs, $NO_x$, $SO_2$, BC, OC | (Venkataraman et al., 2018) |

[a]Not updated from CEDS v2019-12-23, details in Hoesly et al. (2018).
[b]Emissions scaled as a function of sector and fuel-type

**Figures**

[Figure]

Figure 1: Default CEDS System Summary, adapted from Fig. 1 in Hoesly et al. (2018). Key steps include: (0) collecting activity driver (A) and emission factor (EF) input data for non-combustion and combustion emission sources, (1) calculating default emissions (Em) as a function of chemical species, country, emission sector, fuel-type, and year, (2) calculating scaling factors (SFs) for overlapping years with existing inventories in order to scale default estimates (sEm) and extending SFs for non-overlapping years between 1970 – 2017  (for earlier emissions, see Hoesly et al. (2018)), (4) aggregating scaled emissions to intermediate sectors and fuel-types, and (5) using source and compound-specific spatial proxies to calculate final gridded emissions and aggregating them to the final sectors and fuels. A list of intermediate and final sectors and fuels are in Table 2.

[Figure]

Figure 2: Final scaling inventories used for CEDS$_{GBD-MAPS}$ NO$_x$ emissions, inventory details in Table 3.

[Figure]

1755    **Figure 3: Sectoral contributions to total annual emissions for 2014 of CEDS_Hoesly (left) and CEDS_GBD-MAPS (right) emissions after scaling to DICE-Africa and SMoG-India regional inventories. The total annual emissions are given by the values above each bar, bar colors represent absolute sectoral contributions to emissions of each chemical compound. CO and NMVOC emissions are divided by 10 for clarity. Stars indicate that NMVOCs, BC, and OC emissions are in units of TgC yr$^{-1}$. NO$_x$ is in units of Tg NO$_2$ yr$^{-1}$**

[Figure]

1760    **Figure 4: Comparison of the percent difference between CEDS_GBD_MAPS. X and Y-axes show the percent difference between the CEDS emission inventories (y-axis: CEDS_GBD-MAPS, x-axis: CEDS_Hoesly) for each compound and the GAINS (ECLIPSE v5a) or EDGARv4.3.2 inventories from Africa and India (i.e., 100*(CEDS – EDGAR)/(CEDS – EDGAR)/2)). Comparisons are conducted with the most recent available year, 2010 for the comparison with GAINS and 2012 for the comparison with EDGAR. Green regions indicate areas where the CEDS_GBD-MAPS emissions have improved agreement with EDGAR and GAINS relative to the CEDS_Hoesly**
1765    **inventory. Red areas indicate regions where CEDS_GBD-MAPS emissions have worse agreement with EDGAR or GAINS relative to the CEDS_Hoesly inventory. The color of each point represents the chemical compound and each point is labeled with an 'E' or 'G' indicating that the percent difference was calculated using EDGAR or GAINS, respectively.**

[Figure]

**Figure 5: Comparison of global inventories of BC and OC emissions. Total EDGARv4.3.2 and GAINS (ECLIPSE v5a) emission inventories shown without agricultural waste burning and aviation emissions. CEDS$_{GBD-MAPS}$ emissions of BC and OC are not scaled to EDGAR or GAINS estimates.**

[Figure]

**Figure 6. Time series of global annual emissions of $NO_x$ (as $NO_2$), CO, $SO_2$, NMVOCs, $NH_3$, BC, and OC for all sectors and fuel types. Solid black lines are the CEDS_GBD-MAPS inventory, with fractional sector contributions indicated by colors. Dashed gray lines are the CEDS_Hoesly inventory. Dashed blue lines are the EDGAR v4.3.2 global inventory. Red markers are ECLIPSE v5a baseline 'current legislation' (CLE) emissions (from the GAINS model) with data in 2015 and 2020 from GAINS CLE projections. All inventories include international shipping but exclude aircraft emissions. Pie chart inserts show fractional contributions of emission sectors to total 2017 emissions (outer) and fuel type contributions to each sector (inner). Emission totals for 2017 (units: Tg yr$^{-1}$, TgC yr$^{-1}$ for NMVOCs, OC, BC) are given inside each pie chart.**

[Figure]

**Figure 7. Time series of global annual emissions of NO$_x$, CO, SO$_2$, NH$_3$, NMVOCs, BC, and OC for all sectors, colored by fuel group.**

[Figure]

790

**Figure 8. Time series of global annual CEDS$_{GBD-MAPS}$ emissions of NO$_x$, CO, SO$_2$, NH$_3$, NMVOCs, BC, and OC for all sectors and fuel types (excluding aircraft emissions), split into 11 countries/regions (defined in Table S9).**

---

## Referee Report (RR1)

**Review of revised essd-2020-103_v4** "A global anthropogenic emission inventory of atmospheric pollutants from sector- and fuel-specific sources (1970–2017): An application of the Community Emissions Data System (CEDS)" by McDuffie et al.

I would like to compliment and thank the authors for a very detailed response to this reviewer's comments and suggestions. Personally I think this has increased the clarity of an already well-written manuscript. All points made are well addressed in the revised MS. To mention a few;

- I am glad to read that the authors agree with my reservations on the use of the term "calibration" (inventories) and have adjusted the MS accordingly.
- The new Figure S2 provides interesting information and this is well discussed in the revised manuscript. It is a general fact that using a (scaling) relationship outside its domain (e.g., application on historic years not present in the regional scaling inventory) can rapidly increase uncertainty. By making that visible in Fig S2 the discussion is immediately more clear, shows that it generally goes quite well but some outliers exist and are flagged. I truly like the additions made, partly inspired by this figure, starting on Line 676 (4.2.1 Uncertainties in Activity Data) and further in Line 761.
- The new split in the group 'Other Asia / Pacific/ Middle East region into the Australasia, Middle East, and Other Asia / Pacific regions has no doubt been some work in making new figures but makes the discussion sections in which this country group occurs immediately more focussed on where the changes/ or large contributions are really happening. Even in Fig 8, which is of course more crowded with the additional country groups, it can be seen that the trends in the now broken down groups for some pollutants (e.g. NMVOC) are different.

I will not mirror all changes the authors implemented, just state here that they did an excellent job.. In this reviewer's opinion the description but also discussion of the data in this (well-written) MS has substantially improved and I warmly recommend publication.